# Defective vascular smooth muscle cell tafazzin impairs mitochondrial function and promotes atherosclerosis in preclinical models

Cindy Dong[1], Alison Finigan[1], Nichola Figg[1], Benjamin Jenkins [2], Albert Koulman [2], Suvagata R. Chowdhury[3], Anne-Marie Tricolici[1], Jordi Lambert [1], Sebnem Oc [1], Helle F. Jørgensen [1], Julien Prudent [3], Michael P. Murphy[3,4], Martin Bennett [1] & Emma Yu [1] ✉

Atherosclerotic lesions show significant mitochondrial dysfunction but the underlying mechanisms and consequences remain unknown. Cardiolipin is a phospholipid found exclusively in the mitochondrial inner membrane, the site of oxidative phosphorylation. Tafazzin is a trans-acylase that acylates immature monolysocardiolipin to mature cardiolipin. Tafazzin mutations can result in Barth's Syndrome, which is characterised by dilated cardiomyopathy, skeletal myopathy and impaired growth. However, a role for tafazzin in atherosclerosis development has not been previously identified. Here we show that tafazzin expression is decreased in atherosclerotic lesions and specifically in plaque vascular smooth muscle cells (VSMCs). MicroRNA 125a-5p expression is increased in plaques, downregulates tafazzin expression and is induced by oxidised low-density lipoprotein in a NFκB-dependent manner. Silencing tafazzin or overexpression of mutant tafazzin decreases VSMC cardiolipin content and mitochondrial respiration, and promotes apoptosis and atherosclerosis. In contrast tafazzin overexpression increases respiration, protects against apoptosis and increases features of plaque stability. Tafazzin therefore has important effects on VSMC mitochondrial function and atherosclerosis, and is a potential therapeutic target in atherosclerotic disease.

Atherosclerosis remains the leading cause of death worldwide and is characterised by the formation of fibro-fatty plaques in arteries[1]. Plaques can rupture, leading to thrombus formation, vessel occlusion and clinical sequelae such as myocardial infarction and stroke[2]. With such morbidity and mortality, there is a continuing need to identify underlying disease mechanisms and therapeutic targets.

Vascular smooth muscle cells (VSMCs) show a wide range of phenotypes within plaques and produce extracellular matrix that forms the protective plaque fibrous cap[3]. Both the plaque cap and plaque-derived VSMCs show decreased mitochondrial respiration[4], but what disrupts VSMC mitochondrial function in atherosclerosis and the consequences of impaired VSMC mitochondrial respiration are unknown.

[1]Section of Cardiorespiratory Medicine, Victor Phillip Dahdaleh Heart and Lung Research Institute, University of Cambridge, Cambridge, UK. [2]Core Metabolomics and Lipidomics Laboratory, Wellcome-MRC Institute of Metabolic Science Metabolic Research Laboratories, University of Cambridge, Cambridge, UK. [3]Medical Research Council Mitochondrial Biology Unit, University of Cambridge, Cambridge, UK. [4]Department of Medicine, University of Cambridge, Cambridge, UK. ✉e-mail: epky2@cam.ac.uk

Mitochondrial structure is an important determinant of function, with mitochondria existing in a dynamic network regulated by synthesis, fusion, fission and breakdown (mitophagy)[5]. Mitochondria exist as double-membraned organelles, characterised by a highly folded mitochondrial inner membrane. The inner membrane is the site of the respiratory chain and ATP synthase, which together use substrates from the Krebs cycle to generate ATP, with ROS formed as a by-product[6,7]. Cardiolipin (CL) is an important component of the inner membrane, stabilising protein-lipid interactions and relieving membrane stress induced by protein crowding[8,9]. CL is synthesised in an immature form, deacylated to monolysocardiolipin (MLCL), and subsequently reacylated to mature CL by tafazzin, a nuclear-encoded transacylase[10]. Mature CL can then perform multiple functions, including stabilising respiratory supercomplexes and regulating the release of cytochrome C and hence apoptosis[11,12]. Mutations in *TAFAZZIN* can cause Barth's Syndrome, which is characterised by cardiomyopathy, skeletal myopathy and impaired growth[13]. However, whether tafazzin contributes to the pathogenesis of atherosclerosis is unknown.

We show that tafazzin mRNA and protein are decreased in human atherosclerotic lesions and plaque-derived VSMCs. MicroRNA 125a-5p downregulates VSMC tafazzin expression and is induced by oxidised low-density lipoprotein (oxLDL) via NFκB activation. We show that tafazzin is an important determinant of VSMC cardiolipin content, mitochondrial function, apoptosis and atherogenesis. Finally, we identify that the small peptide SS-31 can partially rescue VSMC mitochondrial function and protect against apoptosis.

## Results

### Human plaques and VSMCs show reduced tafazzin expression

We first examined mRNA expression of *TAFAZZIN* in human atherosclerotic plaques compared with normal aortic tissue. Age and sex of patients was similar for both groups ($n = 9$ with 67% male per group, mean age $69.4 \pm 6.6$ years and $67.3 \pm 3.3$ years for aortas and plaques, respectively). *TAFAZZIN* mRNA expression (Fig. 1A) was lower in human plaques compared to normal aortic tissue. We also found decreased tafazzin protein expression in plaques with no change in citrate synthase protein level, an indicator of total mitochondrial mass (Fig. 1B). Immunofluorescence using alpha-smooth muscle actin (SMA) as a VSMC marker demonstrated that $90.5 \pm 1.6\%$ of SMA-positive cells expressed tafazzin in normal aortic tissue, with similar expression in medial SMA-positive cells of plaques. However, only $7.0 \pm 4.3\%$ of intimal SMA-positive cells expressed tafazzin in plaques (Fig. 1C, D). As plaques comprise a heterogeneous mixture of cells, and VSMC markers can be downregulated in diseased vessels, we cultured VSMCs from normal aortas and plaques ($n = 4$). We confirmed that isolated VSMCs expressed SMA and smooth muscle myosin heavy chain but were negative for the macrophage marker CD68 (Supplementary Fig. 1A). Similar to tissue samples, *TAFAZZIN* mRNA expression was decreased in plaque-derived VSMCs compared with normal aortic VSMCs (Supplementary Fig. 1B). Analysis of combined, published scRNAseq data[14] from human coronary and carotid plaque cells[15–17] also showed that *TAFAZZIN* expression was low in plaque VSMCs and significantly lower in VSMCs compared with macrophages (log2 (fold change) = $-0.598$, $p = 1.15 \times 10^{-45}$) and endothelial cells (log2 (fold change) = $-0.769$, $p = 1.17 \times 10^{-20}$) (Supplementary Fig. 2).

We next examined VSMC-specific tafazzin expression in atherosclerotic mice (Myh11-CreERt2/Rosa26-Confetti/Apoe$^{-/-}$), where VSMC-lineage tracing was induced prior to high fat feeding such that expression of the confetti reporters (red, yellow, green or cyan fluorescent protein) marks VSMCs and VSMC-derived cells[18]. Tafazzin expression was significantly decreased in plaque VSMCs compared with medial VSMCs of atherosclerotic arteries (Supplementary Fig. 3). Together, these data show that tafazzin expression is reduced in plaque VSMCs compared with medial VSMCs from diseased or normal vessels.

To examine the mechanism underlying reduced tafazzin expression, we examined possible upstream regulators with relevance to atherosclerosis. MicroRNAs lead to repression of their targets through decreased translational efficiency and/or destabilising their target mRNA[19]. The TargetScan miRNA database identified *TAFAZZIN* as a potential target of microRNA 125a-5p (miR-125a-5p) with putative binding within the 3'UTR (Supplementary Fig. 4A)[20]. To determine whether miR-125a-5p regulates tafazzin expression in atherosclerosis, we first examined its expression in human plaques. miR-125a-5p expression was increased in plaques compared with normal aortic tissues (Supplementary Fig. 4B). In situ hybridisation for miR-125a-5p was then performed on plaque sections, immediately adjacent to sections stained for SMA and tafazzin (Fig. 1C). miR-125a-5p expression was increased in the plaque intima, where tafazzin expression was decreased (Fig. 1C–E).

Tafazzin and miR-125a-5p expression were further examined in mouse atherosclerosis development. Apolipoprotein E-deficient mice (Apoe$^{-/-}$) were either sacrificed at 6 weeks of age when no plaques are present or after a further 14 weeks of high-fat diet, when atherosclerosis is established[21]. Aortic tafazzin protein abundance was reduced and miR-125a-5p expression significantly increased after high-fat feeding in Apoe$^{-/-}$ mice (Supplementary Fig. 5A, B). In contrast, no aortic root atherosclerotic lesions nor changes in tafazzin abundance were observed in C57BL/6 Apolipoprotein E wild-type mice (Apoe$^{+/+}$) after high-fat feeding (Supplementary Fig. 5C, D). Our mouse and human data, therefore, indicate that reduced tafazzin expression is specifically associated with plaque development and with increased miR-125a-5p expression.

### MicroRNA 125a-5p regulates tafazzin expression

As human plaques showed increased miR-125a-5p expression, we next examined the binding of miR-125a-5p to the 3'UTR of *TAFAZZIN*. We used dual luciferase reporter vectors with either wild-type or mutated *TAFAZZIN* 3'UTR sequences (Fig. 2A) inserted downstream of a firefly luciferase reporter gene. Renilla luciferase was used as a control reporter for normalisation. Human VSMCs (hVSMCs) were co-transfected with the dual luciferase vectors and control or a commercially available miR-125a-5p mimic. Dual-luciferase assays showed decreased firefly/renilla luciferase activity for wild-type but not mutated *TAFAZZIN* 3' UTR after transfection with miR-125a-5p mimic, verifying that *TAFAZZIN* is a direct target of miR-125a-5p (Fig. 2B).

Having confirmed miR-125a-5p binding to *TAFAZZIN* 3'UTR, we then examined whether miR-125a-5p regulates tafazzin expression in VSMCs. hVSMCs were transfected with the miR-125a-5p mimic, and increased miR-125a-5p expression confirmed compared with transfection of a non-targeted control (Fig. 2C). Both tafazzin mRNA and protein abundance were decreased by the miR-125a-5p mimic (Fig. 2D, E), with no change in citrate synthase protein abundance, consistent with a tafazzin-targeted rather than a global mitochondrial reduction (Fig. 2E).

As miR-125a-5p is increased in plaques vs. normal aortas and decreases tafazzin expression, we examined which factors might regulate miR-125a-5p expression. During plaque development, low-density lipoprotein (LDL) becomes trapped in the arterial intima, modified by lipoxygenases or ROS to form oxidised LDL (oxLDL)[22], and taken up by VSMCs[23]. Native (unoxidised) LDL treatment did not affect either miR-125a-5p or *TAFAZZIN* expression (Supplementary Fig. 6A, B); in contrast, oxLDL increased miR-125a-5p expression and decreased tafazzin mRNA and protein abundance, an effect that was blocked by transfection with a miR-125a-5p inhibitor (Fig. 2F–H). Taken together, our data suggest that oxidised LDL increases miR-125a-5p in hVSMCs, resulting in reduced tafazzin expression.

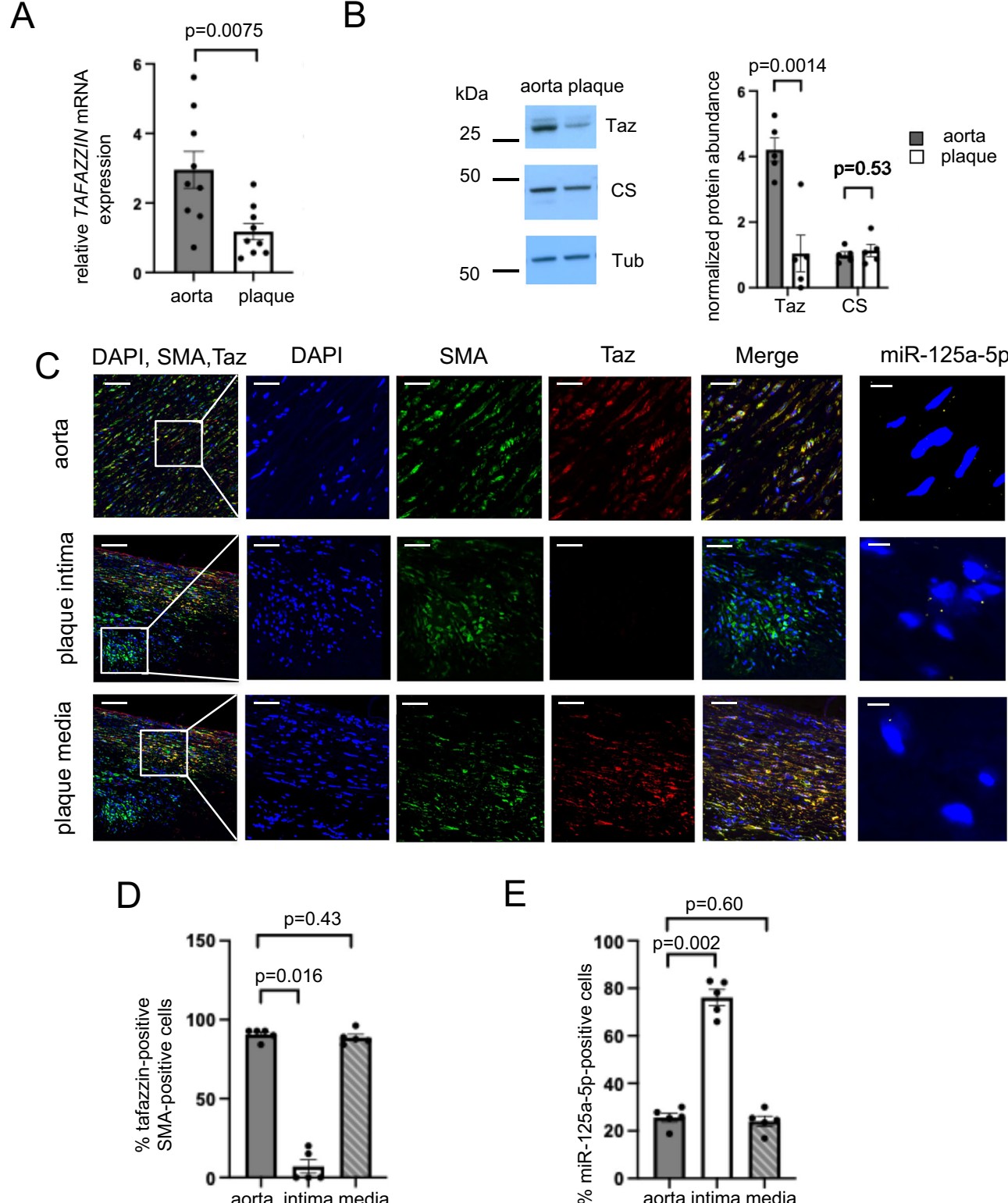

We then examined the mechanism by which oxLDL might increase miR-125a-5p expression. The MotifMap system[24] identified putative NFκB binding sites in the upstream region of the miR-125a-5p transcription start site (Fig. 3A), suggesting that NFκB signalling may regulate oxLDL-induced miR-125a-5p upregulation. OxLDL increased NFκB binding to its consensus DNA motif, determined using the TransAM NFκB p65 Transcription Factor Assay, and increased NFκB phosphorylation and miR-125a-5p expression (Fig. 3B–D). The increased NFκB phosphorylation and miR-125a-5p expression were associated with decreased tafazzin abundance (Fig. 3C). We then treated hVSMCs with Bay 11-7082, which inhibits IκBα phosphorylation and decreases NFκB nuclear translocation[25]. Bay 11-7082 blocked the oxLDL-induced increases in NFκB binding and phosphorylation, and furthermore blocked the increase of miR-125a-5p expression (Fig. 3B–D). The decreased NFκB phosphorylation-dependent change in miR125a-5p was sufficient to abrogate the reduction in tafazzin (Fig. 3C). Similarly siRNA mediated knockdown of NFκB in hVSMCs blocked oxLDL-induced increases in NFκB binding, phosphorylation

**Fig. 1 | Tafazzin expression is reduced in human plaques and plaque vascular smooth muscle cells (VSMCs). A** Quantitative PCR for *TAFAZZIN* mRNA expression in human plaques and aortic tissues. Expression is shown relative to plaque, normalised to beta-2-microglobulin (*B2M*) house-keeping gene (*n* = 9 samples per group, 6 male (M), 3 female (F) donors, two-sided unpaired *t* test). **B** Representative western blot of tafazzin (Taz) and citrate synthase (CS) in aortic and plaque samples with quantification. Tub = tubulin. Data normalised to citrate synthase for tafazzin or tubulin for citrate synthase (*n* = 5 samples per group, 3 M, 2 F, two-sided unpaired *t* test). **C** Left panel: Confocal images ("max projection" from z-stack) of immunofluorescence staining of VSMCs in aorta, plaque intima and plaque media for DAPI (blue), alpha-smooth muscle actin (SMA) (green) and tafazzin (Taz)(red) with overlay. Scale bars = 100 µm in low power views, 25 µm in high power views of outlined areas. Right panel: In situ hybridisation for microRNA 125a-5p (miR-125a-5p)(yellow) with nuclei stained with DAPI (blue). Scale bars = 5 µm. **D** Quantification of % SMA-positive cells expressing tafazzin in aorta, plaque intima (intima) and plaque media (media) (*n* = 5 samples per group, 3 M, 2 F, Kruskal-Wallis test with post-hoc Bonferroni-Holm). **E** Quantification of % cells expressing miR-125a-5p in aorta, plaque intima and media (*n* = 5 samples per group, 3 M, 2 F, 1-way ANOVA, post hoc Bonferroni-Holm). Data are shown as mean ± SEM with nominal or multiplicity adjusted *p*-values. Source data are provided as a Source Data file.

and miR-125a-5p expression, preventing the decrease in tafazzin protein abundance (Supplementary Fig. 7). Overall, our data suggest that oxLDL increases miR-125a-5p expression in a NFκB-dependent manner, with the increased miR-125a-5p then reducing tafazzin expression.

As oxLDL and miR-125a-5p decreased tafazzin expression, we next examined whether there were subsequent effects on mitochondrial respiration or on cellular function. Mitochondrial respiration was assessed using a Seahorse extracellular flux analyser, with maximal oxygen consumption rate (OCR) determined after uncoupling with carbonyl cyanide 4-(trifluoromethoxy)phenylhydrazone (FCCP). hVSMCs transfected with miR-125a-5p mimic showed decreased maximal OCR, decreased proliferation as determined by EDU incorporation, and increased apoptosis, as determined by flow cytometric analysis of annexin V staining (Fig. 3E–G and Supplementary Fig. 8A). Similar effects were observed after treatment with oxLDL (Supplementary Fig. 8B–E). Collectively, our data indicates that oxLDL increases miR-125a-5p expression, which decreases tafazzin expression; furthermore, both oxLDL and miR-125a-5p reduce mitochondrial respiration and cell proliferation, and increase apoptosis, suggesting that the decrease in tafazzin might contribute to these processes. However, as oxLDL and miR-125a-5p may have multiple targets, we next performed specific tafazzin-targeted experiments to identify functional consequences of reduced tafazzin expression or activity.

### Tafazzin is an important determinant of cardiolipin content and mitochondrial function in VSMCs

Tafazzin acylates monolysocardiolipin to form tetra-acyl, mature cardiolipin, an important component of the mitochondrial inner membrane[10]. To examine the functional consequences of loss of tafazzin in VSMCs, we performed siRNA-mediated cell silencing, which reduced tafazzin abundance (Fig. 4A). We also used lentivirus-mediated gene transfer to stably express either wild-type tafazzin (Taz) or tafazzin with a histidine to glutamine mutation at the transacylase catalytic site (TazH69Q, TazH) that decreases tafazzin transacylase activity[26,27] (Fig. 4A) in mouse VSMCs (mVSMCs).

Complex I subunit abundance was reduced after tafazzin depletion, but there were no changes in citrate synthase (Fig. 4A) or CII-CV subunit abundance (Supplementary Fig. 9). We also determined tetra-acyl cardiolipin (CL) content in VSMCs using lipid chromatography mass spectrometry (LCMS). *TAFAZZIN* silencing or expression of TazH reduced CL content, whilst expression of Taz increased CL content (Fig. 4B). As CL is important for regulating mitochondrial dynamics[28], we examined mitochondrial morphology using confocal imaging. *TAFAZZIN* silenced or Taz-overexpressing VSMCs showed no changes in mitochondrial morphology (Supplementary Fig. 10, 11). Whilst both FIS1 and OPA1 were increased in *TAFAZZIN* silenced VSMCs, no changes in mitochondrial fusion and fission protein abundance were observed in Taz overexpressing VSMCs (Supplementary Fig. 10, 11).

We next assessed mitochondrial respiration using a Seahorse extracellular flux analyser. Maximal OCR was reduced in *TAFAZZIN* silenced cells and in VSMCs overexpressing TazH (Fig. 4C–E), and this was associated with reduced ATP/ADP ratio as determined by a luciferin-luciferase assay (Fig. 4F). In contrast, VSMCs overexpressing wild-type tafazzin showed increased OCR (Fig. 4D, E) with no change in ATP/ADP ratio (Fig. 4F). Impaired mitochondrial function may also lead to changes in mitochondrial ROS, which were assessed with flow cytometric analysis of MitoSOX staining. *TAFAZZIN* silenced and VSMCs over-expressing TazH showed an increase in MitoSOX fluorescence (Fig. 4G). No changes in mitochondrial membrane potential (ΔΨm), as determined by live-cell imaging with tetramethyl rhodamine ethyl ester (TMRE), were found in *TAFAZZIN* silenced VSMCs or in VSMCs over-expressing wild-type tafazzin or TazH (Supplementary Fig. 12).

As mitochondria are important regulators of both cell proliferation and cell death[29,30] we assayed proliferation using EdU, which is incorporated into newly synthesised DNA, and performed annexin V flow cytometric analysis of cell death. *TAFAZZIN* silenced VSMCs showed reduced EDU incorporation (Fig. 4H), whilst Taz over-expressing VSMCs showed reduced cell death (Fig. 4I). In addition, both *TAFAZZIN* silenced and TazH-overexpressing VSMCs showed increased cell death upon exposure to oxidative stress by treatment with tert-butyl hydroperoxide, whilst Taz-overexpressing VSMCs continued to show protection against cell death (Fig. 4J). Collectively, our data indicate that tafazzin is an important determinant of cardiolipin content, mitochondrial function, cell proliferation and cell death in VSMCs.

### Characterisation of VSMC-specific Taz and TazH mice

Whole body tafazzin knockdown induces left ventricular dilatation, decreased ejection fraction, neutropenia and altered plasma cholesterol levels, that could confound a study in atherosclerosis[13,31]. We therefore generated C57BL/6 mice expressing *TAFAZZIN* (Sm22a-Taz) or TazH69Q functional mutant (*Sm22a-TazH69Q*) from the minimal *Sm22a* promoter. The minimal *Sm22a* promoter is only expressed in VSMCs in large arteries in adult mice[32], and three base pair mutations have been introduced into its CArG elements to avoid downregulation upon phenotypic switching[33]. Importantly, unlike the full-length *Sm22a* promoter, this version of the minimal *Sm22a* promoter is not expressed in bone marrow, peripheral blood cells, or the spleen[34].

Transgenic mice expressed either human *TAFAZZIN* (Taz) or TazH69Q that was restricted to the aorta with no change in endogenous mouse *Tafazzin* expression (Fig. 5A, B). We isolated VSMCs from these mice and confirmed positive SMA expression and negative CD68 expression (Supplementary Fig. 13). The VSMCs showed similar levels of either Taz or TazH69Q expression, again with no change in mouse *Tafazzin* expression (Fig. 5C and Supplementary Fig. 14A). Western blotting confirmed increased tafazzin protein in Taz or TazH69Q VSMCs together with increased complex I subunit abundance in Taz VSMCs (Fig. 5D). In contrast, VSMCs from *Sm22a-TazH69Q* mice showed reduced complex I subunit abundance with no changes observed in citrate synthase or other respiratory complex subunits (Fig. 5D and Supplementary Fig. 14B). As there were changes in complex I NDUFB8 subunit abundance, we next examined mRNA expression of multiple complex I subunit genes (*Ndufb8, Ndufs1, mt-Nd1, mt-Nd2, Ndufs8*), covering the p, q and n modules[35], and both mitochondrial and nuclear-encoded genes. Expression of these genes was unchanged in Taz and TazH69Q VSMCs, consistent with post-transcriptional regulation of protein abundance (Supplementary Fig. 14C). Cardiolipin stabilises respiratory supercomplexes[11] therefore, we next examined respiratory complex

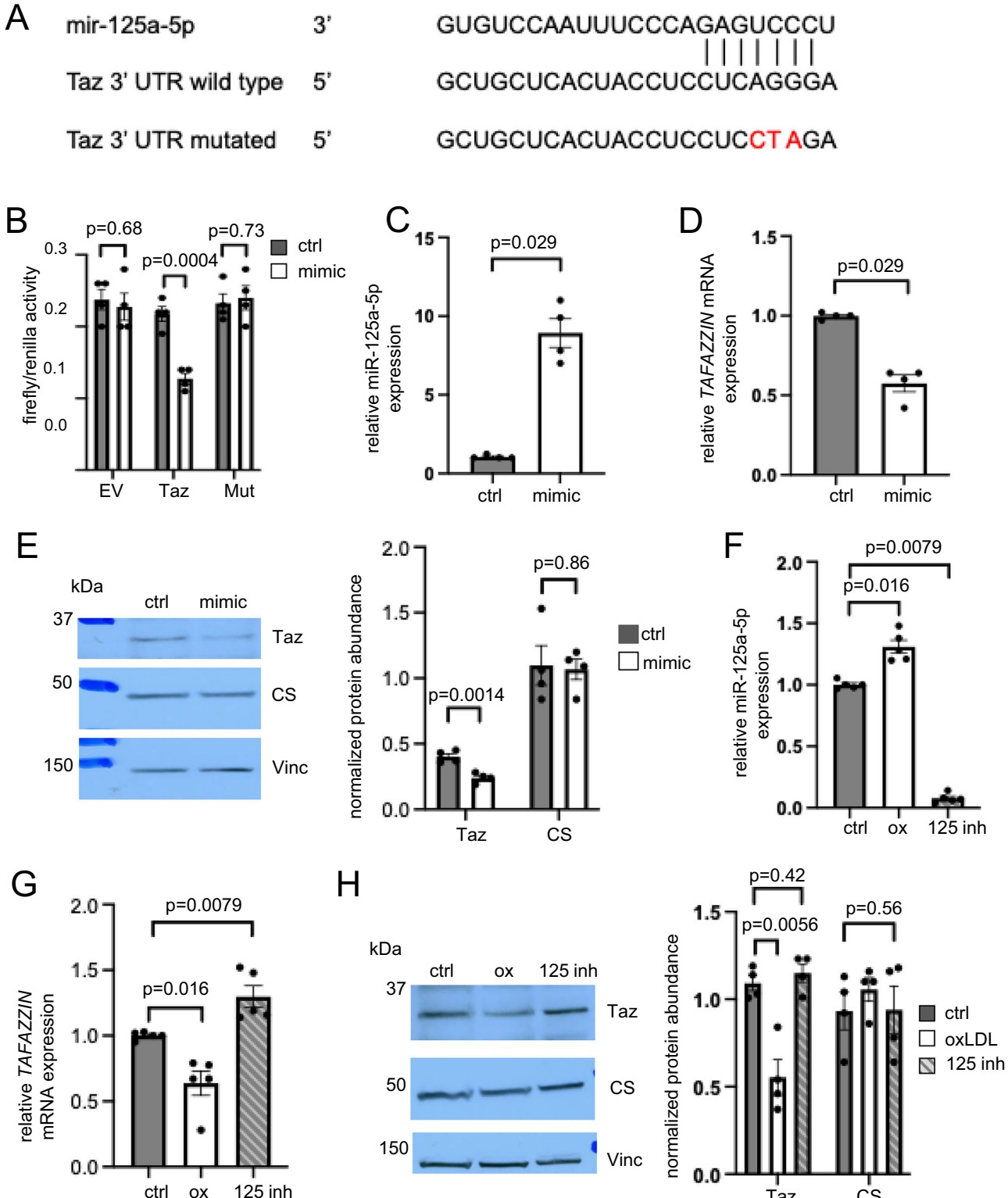

assembly using blue native polyacrylamide gel electrophoresis and immunoblot analysis. Whilst levels of complex I/III$_2$/IV supercomplex and individual complex I were increased in Taz VSMCs, lower levels of these complexes were seen in Taz$^{H69Q}$ VSMCs, which would explain the changes observed in subunit abundance (Supplementary Fig. 14D). These data indicate that tafazzin regulates complex I subunit abundance through effects on holoenzyme and supercomplex stability, not at the transcriptional level.

We next assessed mitochondrial function in Taz and Taz$^{H69Q}$ mVSMCs. Taz VSMCs show increased basal OCR, maximal OCR and ATP/ADP ratio, and decreased mitochondrial ROS, compared with control VSMCs (Fig. 5E–I). In contrast, VSMCs from *Sm22a-Taz$^{H69Q}$* mice showed reduced basal and maximal OCR, reduced ATP/ADP ratio and increased mitochondrial ROS (Fig. 5E–I). These data are consistent with increased mitochondrial function in Taz VSMCs and reduced function in Taz$^{H69Q}$ VSMCs.

**Fig. 2 | Tafazzin expression is regulated by microRNA 125a-5p (miR-125a-5p).**
**A** Wild type and mutated *TAFAZZIN* (Taz) 3′ UTR sequences with putative binding of miR-125a-5p as shown. **B** Firely/renilla activity in VSMCs co-transfected with dual luciferase empty vector (EV) or dual luciferase vectors expressing wild type *TAFAZZIN* (Taz) or mutated (mut) Taz 3′ UTR sequences and control (ctrl) or miR-125a-5p mimic (mimic) (*n* = 4, two-sided unpaired *t* test). **C, D** Quantitative PCR for miR-125a-5p (**C**) and *TAFAZZIN* mRNA (**D**) in hVSMCs transfected with non-targeting control (ctrl) or miR-125a-5p mimic (mimic) (*n* = 4, two-sided Mann-Whitney test). **E** Representative western blot for tafazzin (Taz) and citrate synthase (CS) in hVSMCs transfected with non-targeting control (ctrl) or miR-125a-5p mimic (mimic) with quantification (*n* = 4, two-sided unpaired *t* test). Vinc = vinculin. **F, G** Quantitative PCR for miR-125a-5p (**F**) and *TAFAZZIN* mRNA (**G**) in control hVSMCs (ctrl) or

hVSMCs treated with oxidised LDL (ox) or oxidised LDL + miR-125a-5p inhibitor (125 inh) (*n* = 5, Kruskal-Wallis test, post-hoc Bonferroni-Holm). **H** Representative western blot for tafazzin (Taz) and citrate synthase (CS) in control hVSMCs (ctrl) or hVSMCs treated with oxidised LDL (ox) or oxidised LDL + miR-125a-5p inhibitor (125 inh) with quantification (*n* = 4, 1-way ANOVA, post hoc Bonferroni-Holm). Vinc = vinculin. For QPCR data, expression is shown relative to control, normalised to U6 small nuclear RNA (*RNU6-1*) (**C, F**) or *B2M* (**D, G**). For western blot data (**E, H**), protein abundance is normalised to citrate synthase for tafazzin or vinculin for citrate synthase. For (**B–H**) *n* = number of independent experiments. For all experiments hVSMCs from 2 male (M) and 2 female (F) donors were used except (**F, G**): 2M, 3 F. Data are shown as mean ± SEM with nominal or multiplicity adjusted *p*-values. Source data are provided as a Source Data file.

## VSMC tafazzin has important effects on atherogenesis and apoptosis

To study the functional consequences of reduced endogenous VSMC tafazzin on atherosclerosis, mice expressing VSMC-restricted *TAFAZZIN* or Taz$^{H69Q}$ were crossed with *Apoe*$^{-/-}$ mice. Once weaned, littermate *Apoe*$^{-/-}$, *Sm22a-Taz/Apoe*$^{-/-}$ and *Sm22a-Taz*$^{H69Q}$/*Apoe*$^{-/-}$ mice were fat fed from 6 to 20 weeks of age. There were no significant differences in blood pressure, body weight, full blood count or serum lipid profile between genotypes (Supplementary Fig. 15A–D). Both male and female mice were used in the study, and all parameters were similar apart from weight (Supplementary Fig. 15B). The weight data has therefore been segregated by sex whilst remaining data is presented with males and females combined.

Atherosclerosis burden was examined using *en-face* Oil red O staining of the descending aorta together with hematoxylin and eosin staining of the aortic root. Plaque composition was analysed using Masson's trichrome staining; fibrous cap areas were defined as the regions rich in VSMCs, overlying the cholesterol rich, acellular core (Supplementary Fig. 16). *Sm22a-Taz/Apoe*$^{-/-}$ mice showed no change in plaque area compared with control *Apoe*$^{-/-}$ mice (Fig. 6A–C) but had increased fibrous cap and reduced necrotic core areas, together with increased SMA-positive area (Fig. 6D–F and Supplementary Fig. 16). In contrast, *SM22a-Taz*$^{H69Q}$/*Apoe*$^{-/-}$ mice showed increased plaque area in both the descending aorta and the aortic root (Fig. 6A–C) compared with *Apoe*$^{-/-}$ controls. *SM22a-Taz*$^{H69Q}$/*Apoe*$^{-/-}$ mice also showed increased features of plaque vulnerability with increased necrotic core and reduced fibrous cap areas (Fig. 6D, E and Supplementary Fig. 16), reduced SMA-positive area and increased Mac3-positive (macrophage marker) area (Fig. 6F, G).

To further examine the changes in plaque burden and composition, we assessed apoptosis using TUNEL immunofluorescence staining, and co-stained for SMA with nuclei labelled with DAPI (Fig. 7A and Supplementary Fig. 17A). We found *Sm22a-Taz/Apoe*$^{-/-}$ mice showed reduced percentage of VSMCs undergoing apoptosis, whilst *Sm22a-Taz*$^{H69Q}$/*Apoe*$^{-/-}$ mice showed increased VSMC apoptosis compared with controls (Fig. 7A, B). Proliferation was assessed using Ki67 staining (Supplementary Fig. 17). Cellular proliferation was increased in *Sm22a-Taz/Apoe*$^{-/-}$ mice compared with wild type Apoe$^{-/-}$ mice but unchanged in *Sm22a-Taz*$^{H69Q}$/*Apoe*$^{-/-}$ mice (Supplementary Fig. 17B and Fig. 7C). There were no changes in mRNA expression of specific cytokines in the aortic arches from *Sm22a-Taz/Apoe*$^{-/-}$ mice whilst *Sm22a-Taz*$^{H69Q}$/*Apoe*$^{-/-}$ mice showed increased mRNA expression of *Tnfa* with no changes in mRNA expression of *Il1b*, *Il6* or *Mcp-1* (Fig. 7D).

As changes in apoptosis, proliferation and *Tnfa* mRNA expression were observed between mouse genotypes in vivo, we examined these parameters in VSMCs isolated from wild-type, *Sm22a-Taz*, and *Sm22a-Taz*$^{H69Q}$ mice. *Sm22a-Taz* VSMCs showed reduced apoptosis and increased proliferation (Fig. 7E, F), whilst *Sm22a-Taz*$^{H69Q}$ VSMCs showed increased apoptosis (Fig. 7E); *Tnfa* mRNA expression was similar in all VSMCs (Fig. 7G).

## Treatment with SS-31 improves mitochondrial function and reduces apoptosis

Our data indicate that human atherosclerotic plaque VSMCs show reduced tafazzin expression, which may be due to ox-LDL-induced upregulation of miR-125a-5p. Impaired VSMC tafazzin decreases CL content, promotes mitochondrial dysfunction, apoptosis and atherosclerosis. Augmentation of tafazzin might therefore have beneficial effects on plaque development and stability. Whilst there is no current drug that directly targets tafazzin, SS-31 is a small peptide that is reported to bind and stabilise cardiolipin[36]. SS-31 modestly, but significantly, increased maximum OCR in *Sm22a-Taz*$^{H69Q}$ VSMCs (Fig. 8A, B). SS-31 treatment also reduced mitochondrial ROS and apoptosis of *Sm22a-Taz*$^{H69Q}$ VSMCs after tert-butyl hydroperoxide (Fig. 8C, D). Although we observed changes in mitochondrial respiration and ROS production in SS31-treated *Sm22a-Taz*$^{H69Q}$ VSMCs, no changes in tafazzin, complex I subunit or citrate synthase abundance were found in these cells (Fig. 8E).

## Discussion

Our study reports several important findings. We show that tafazzin mRNA and protein are decreased in human plaque tissue and plaque VSMCs; microRNA 125a-5p expression is increased in atherosclerotic plaque tissue and downregulates tafazzin in human VSMCs; microRNA 125-5p expression is induced by oxidised LDL; inhibiting NFκB activation blocks oxidised LDL-induced increase of miR-125a-5p and a decrease of tafazzin; loss of tafazzin or defective tafazzin decreases mature cardiolipin content, decreases mitochondrial respiration and promotes apoptosis in VSMCs; VSMC-specific expression of tafazzin protects against apoptosis and increases fibrous cap area; VSMC-specific expression of defective tafazzin increases atherosclerosis and features of plaque vulnerability; SS-31 improves respiration, reduces ROS and apoptosis of VSMCs with defective tafazzin.

In atherosclerosis, the plaque cap is important for plaque stability and decreased mitochondrial respiration has been identified in both the cap and in plaque-derived VSMCs[4]. However, the cause of VSMC mitochondrial dysfunction is unknown. We show that both tafazzin mRNA and protein are decreased in human atherosclerotic plaques compared with normal aortic tissue. This may reflect differential expression between vascular beds that have structural and developmental differences, as has been observed with other mitochondrial proteins[37]. However, immunofluorescence analysis confirmed that medial VSMCs in diseased tissue retain tafazzin expression. This suggests that the reduction in tafazzin is specifically associated with plaque development.

The decrease of both tafazzin mRNA and protein are consistent with pre-translational regulation, and the TargetScan miRNA database identifies *TAFAZZIN* as a potential target of miR-125a-5p[20]. miR-125a-5p is of particular interest in atherosclerosis, as increased expression is associated with symptomatic carotid plaques and myocardial infarction[38,39]. We show that miR-125a-5p is increased in atherosclerotic tissues, downregulates tafazzin expression in hVSMCs and is induced by oxidised LDL in an NFκB-dependent manner. Our data is consistent

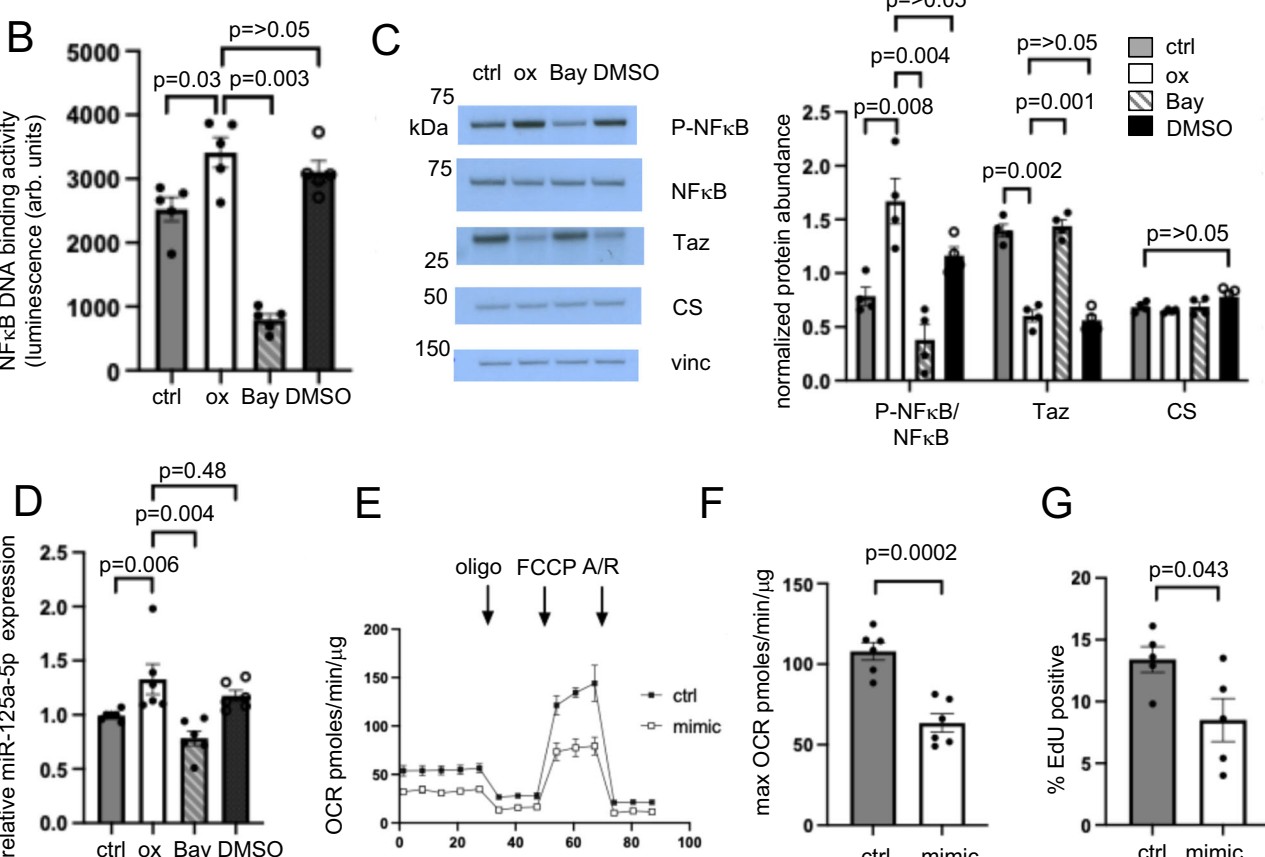

| Transcription Factor | Gene | Distance (bp) from TSS | Region |
|---|---|---|---|
| ETS2 | microRNA 125a | -994 | Upstream |
| NF-kappaB | microRNA 125a | -537 | Upstream |
| NFKB1 | microRNA 125a | -541 | Upstream |
| NFKB-p50 | microRNA 125a | -541 | Upstream |

Fig. 3 | Oxidised LDL upregulates microRNA 125a-5p (miR-125a-5p) in a NFκB dependent manner. A Putative binding sites of NFκB in the upstream region of the miR-125a-5p transcription start site indicated by MotifMap[24]. TSS = Transcription start site. B NFκB DNA binding activity determined by ELISA using TransAM NFκB p65 Transcription Factor Assay in control hVSMCs (ctrl) or hVSMCs treated with oxidised LDL (ox), oxLDL + Bay 11-7082 (Bay) or oxLDL + DMSO (DMSO) (n = 5, 2 male (M), 3 female (F) hVSMC donors, 1-way ANOVA, post hoc Bonferroni-Holm). arb. units = arbitrary units. C Representative western blot image for phosphorylated NFκB (P-NFκB) p65, total NFκB p65 (NFκB), tafazzin (Taz), citrate synthase (CS) in control hVSMCs (ctrl) or hVSMCs treated with oxidised LDL (ox), oxLDL + Bay 11-7082 (Bay) or oxLDL + DMSO (DMSO) with quantification. Vinc = vinculin. Data normalised to citrate synthase for tafazzin or vinculin for citrate synthase (n = 4, 2 M, 2 F, 1-way

ANOVA, post hoc Bonferroni-Holm). D Quantitative PCR for miR-125a-5p in control hVSMCs (ctrl) or hVSMCs treated with oxidised LDL (ox), oxLDL + Bay 11-7082 (Bay) or oxLDL + DMSO (DMSO). Expression is shown relative to control, normalised to RNU6-1 (n = 6, 3 M, 3 F, Kruskal-Wallis test, post-hoc Bonferroni-Holm).
E Representative Seahorse profiles of oxygen consumption rate (OCR) with sequential addition of oligomycin (oligo), FCCP and antimycin/rotenone (A/R) in hVSMCs transfected with non-targeting control (ctrl) or miR-125a-5p mimic (mimic) (n = 3, 3 M). F, G Maximal OCR after FCCP (n = 6, 3 M, 3 F, two-sided unpaired t test) (F) and percentage of cells positive for EdU (G) in hVSMCs transfected with control or miR-125a-5p mimic (n = 5, 3 M, 2 F, two-sided unpaired t test). For (B–G) n = number of independent experiments. Data are shown as mean ± SEM with nominal or multiplicity adjusted p- values. Source data are provided as a Source Data file.

with previous studies, which showed that oxidised LDL increases miR-125a-5p expression in macrophages and endothelial cells[40,41]. However, the link between oxidised LDL, NFκB activation, miR-125a-5p expression and tafazzin regulation in VSMCs has not been described previously. NFκB is a central regulator of inflammation, and it is well recognised that a pro-inflammatory phenotype in macrophages leads to altered metabolism, with increased expression of pro-glycolytic genes including glucose transporter protein type 1 and 6-phospho-fructo-2-kinase/fructose-2,6-bisphosphatase[42]. Our data now suggest a further potential link between inflammatory signalling and VSMC mitochondrial function.

Our findings show that disruptions to tafazzin not only affect VSMC cardiolipin content and mitochondrial function but whole cell function. TAFAZZIN mutations result in abnormal cardiomyocyte contraction, whilst loss of tafazzin in myeloid cells increases apoptosis[43,44]. We find that both Taz-depleted and Taz[H69Q] over-expressing VSMCs show increased apoptosis after tert-butyl hydro-peroxide treatment. VSMC apoptosis is important in atherosclerosis, promoting both plaque development and features of plaque vulnerability[45,46]. Apoptosis is associated with the release of cyto-chrome C, which is directly bound by cardiolipin[12]. Decreased cardiolipin content and/or increased cardiolipin peroxidation secondary to

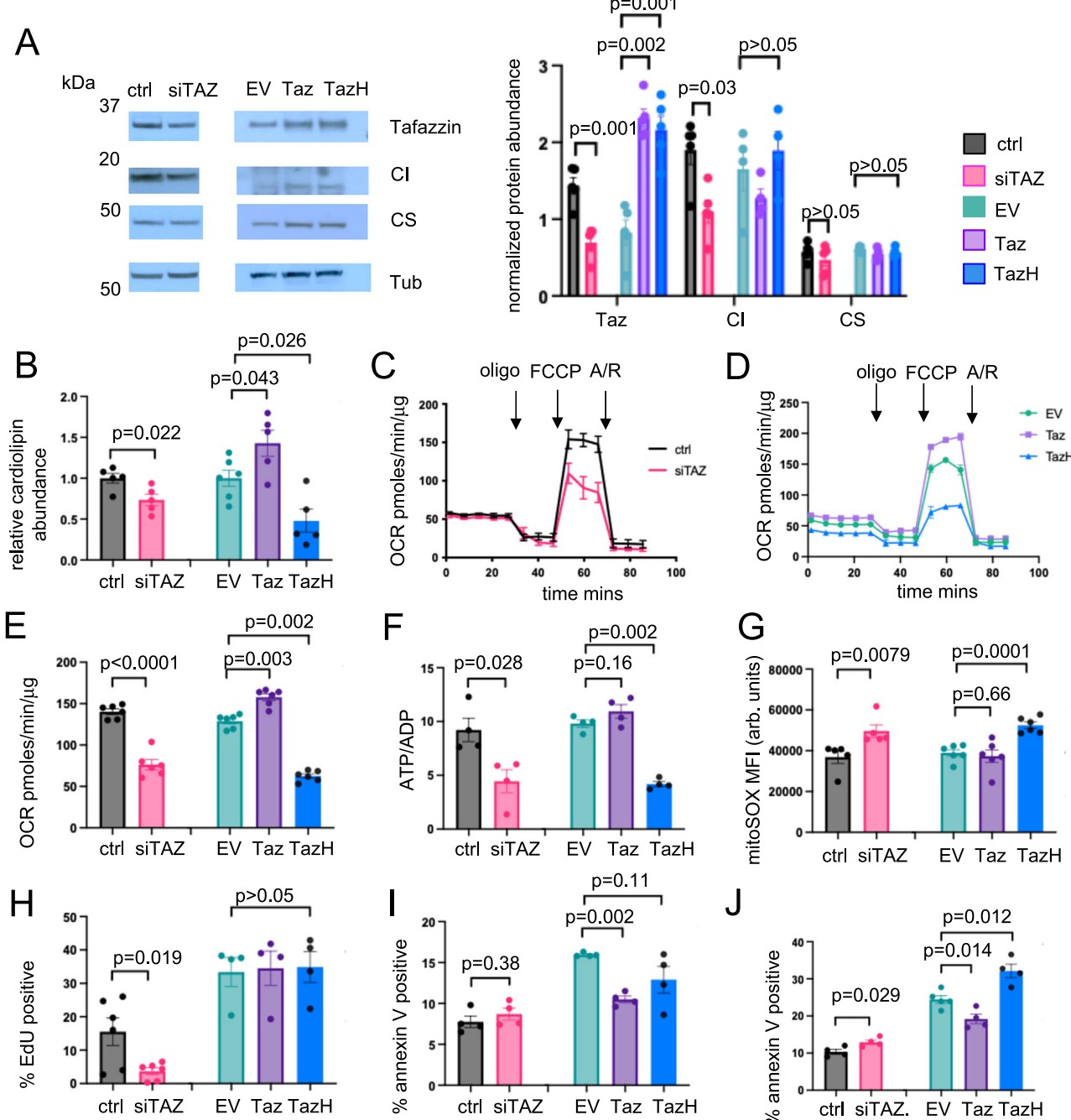

**Fig. 4 | Tafazzin is an important determinant of VSMC cardiolipin content and mitochondrial function. A** Representative western blot for tafazzin, complex I (CI) and citrate synthase (CS) in hVSMCs transfected with non-targeting control (ctrl) or siRNA against *TAFAZZIN* (siTAZ) or in mVSMCs overexpressing empty vector (EV), tafazzin (Taz) or transacylase mutant tafazzin (Taz^H69Q or TazH) with quantification. Tub = tubulin. Data normalised to citrate synthase for tafazzin and CI or tubulin for citrate synthase (*n* = 5, 2 male (M) and 3 female (F) VSMC donors). **B** Tetra-acyl cardiolipin abundance determined by mass spectrometry in siTAZ hVSMCs and in Taz/TazH mVSMCs (*n* = 5, 3 M, 2 F for all groups except EV, *n* = 6, 3 M, 3 F). **C, D** Representative Seahorse profiles of oxygen consumption rate (OCR) with sequential addition of oligomycin (oligo), FCCP and antimycin/rotenone (A/R) in control or siTAZ hVSMCs (**C**) and in EV, Taz, TazH mVSMCs (**D**)(*n* = 3, 2 M, 1 F). **E** Maximal OCR after FCCP in siTAZ hVSMCs and in Taz/TazH mVSMCs (*n* = 6, 3 M, 3 F). **F** ATP/ADP ratio (*n* = 4, 2 M, 2 F) and (**G**) MitoSOX mean fluorescence intensity (MFI) in siTAZ hVSMCs and in Taz/TazH mVSMCs (*n* = 6, 3 M, 3 F). arb. units = arbitrary units. **H–J** Percentage of cells positive for EdU (**H**), annexin V (**I**) and annexin V after tert-butyl hydrogen peroxide treatment (**J**) in siTAZ hVSMCs and in Taz/TazH mVSMCs (for **H** *n* = 6, 2 M, 4 F, for **I, J** *n* = 4, 2 M, 2 F.) For (**A–J**) *n* = number of independent experiments. For (**A, B, E–J**) two-sided unpaired t test for siTAZ vs. control comparisons; 1-way ANOVA, Bonferroni-Holm post hoc for EV, Taz and TazH VSMC comparisons. Data are shown as mean ± SEM with nominal or multiplicity-adjusted *p*-values. Source data are provided as a Source Data file.

increased ROS may therefore facilitate mitochondrial cytochrome C release and promote apoptosis[47].

Cardiolipin also has an important role in regulating mitochondrial dynamics through its interactions with OPA1 to promote mitochondrial inner membrane fusion, and DRP1 to promote fission[28].

However, changes in tafazzin do not consistently alter mitochondrial morphology. Indeed cardiac mitochondria from tafazzin deficient mice appear morphologically normal at 8 weeks old, and even at 8 months the majority of mitochondria showed normal size and structure[13]. Furthermore, when morphological changes occur, they

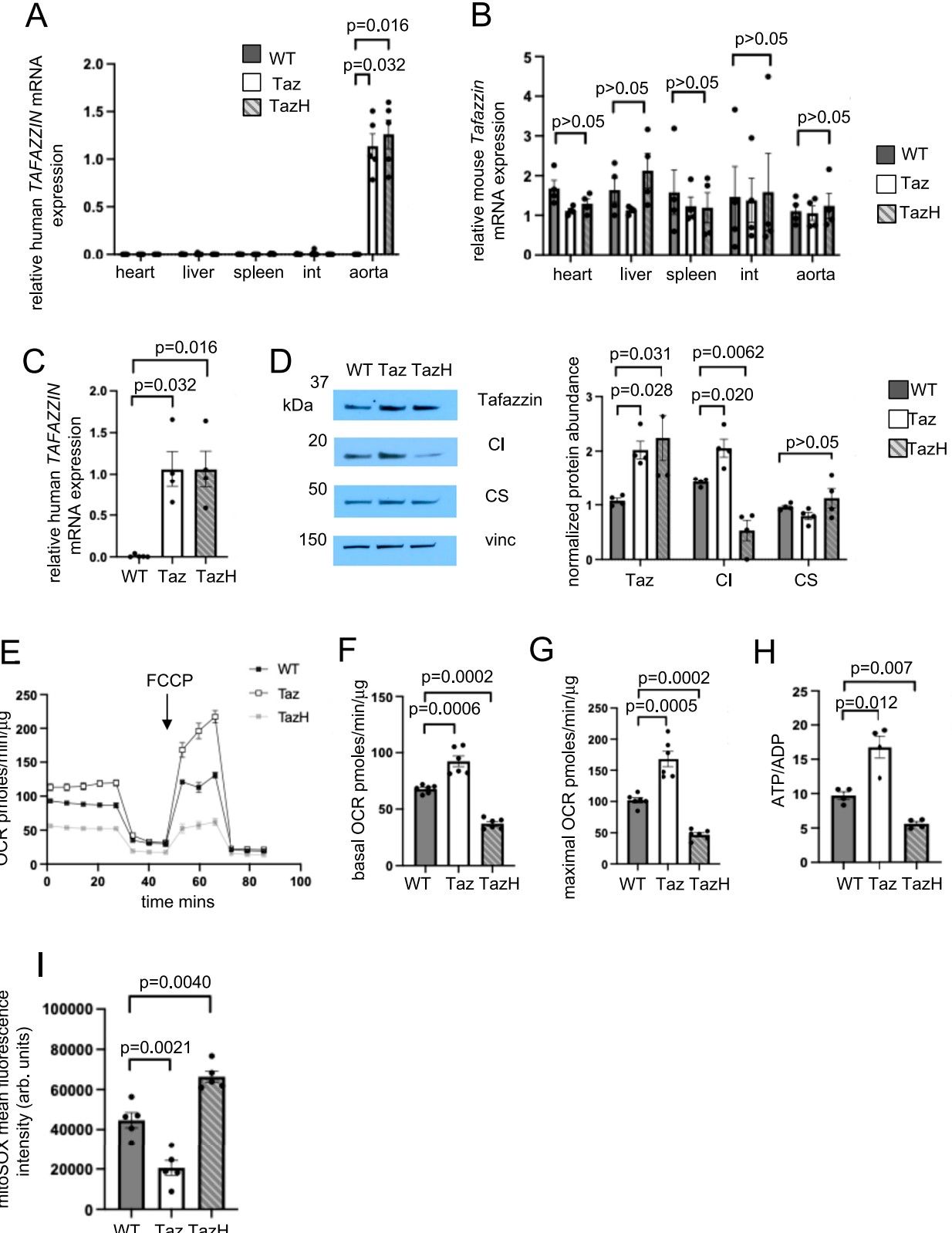

can vary, with reports of both enlarged and fragmented mitochondria in tafazzin deficiency[31,48]. Whether changes in mitochondrial morphology are observed may depend on the balance between the fusion and fission proteins. No consistent pattern in mitochondrial fusion and fission proteins has been reported in tafazzin deficiency; one group observed increased FIS1, whilst others found decreased MFN2[48,49]. We suggest that increased FIS1 and OPA1 in *TAFAZZIN*

silenced VSMCs may promote both fission and fusion with therefore no overall morphological change. In contrast, tafazzin overexpression increases CL content without altering the abundance of mitochondrial dynamics proteins or mitochondrial morphology. The relationship between tafazzin, cardiolipin, mitochondrial dynamics proteins and mitochondrial morphology is therefore complex and remains to be fully elucidated.

**Fig. 5 | Characterisation of *SM22α-Taz* (Taz) and *SM22α-Taz^{H69Q}* (TazH) mice and VSMCs. A, B** QPCR for human *TAFAZZIN* mRNA (*n* = 5, 3 male (M), 2 female (F) donors) (**A**) and mouse *Tafazzin* mRNA (*n* = 4, 2 M, 2 F) (**B**) in heart, liver, intestine (int), spleen and aortas from wild type (WT), *Sm22a-Taz* (Taz) and *Sm22a-Taz^{H69Q}* (TazH) mice (Kruskal-Wallis, Bonferroni Holm post hoc.) **C** QPCR for human *TAFAZZIN* mRNA in VSMCs from WT, Taz and TazH mice (*n* = 4, 2 M, 2 F, Kruskal-Wallis, Bonferroni Holm post hoc.) For data (**A–C**), expression is shown relative to *Sm22a-Taz* mice, normalised to *B2m* housekeeping gene. *n* = number of mice per group. **D** Representative western blot for tafazzin, complex I (CI) and citrate synthase (CS) in VSMCs from experimental mice with quantification. Vinc = vinculin.

Data normalised to citrate synthase for tafazzin and CI or vinculin for citrate synthase (*n* = 4, 2 M, 2 F). **E** Representative Seahorse profiles of oxygen consumption rate (OCR) in WT, Taz and TazH VSMCs (*n* = 3, 2 M, 1 F). **F, G: F** Basal and **G** maximal OCR after FCCP in WT/Taz/TazH VSMCs (*n* = 6, 3 M, 3 F). **H** ATP/ADP ratio (*n* = 4, 2 M, 2 F) and (**I**) mitoSOX mean fluorescence intensity in WT/Taz/TazH VSMCs (*n* = 5, 3 M, 2 F). arb. units = arbitrary units. For (**D–I**): *n* = number of VSMC isolates per group;1-way ANOVA with Bonferroni-Holm post hoc. Data are shown as mean ± SEM with multiplicity-adjusted *p*-values. Source data are provided as a Source Data file.

To study the functional consequences of reduced endogenous VSMC tafazzin, we generated transgenic mouse models expressing either VSMC-specific Taz or the Taz^{H69Q} functional mutant. Taz and Taz^{H69Q} were expressed at similar levels from the same promoter, an important control to be able to compare their effects. VSMCs from *Sm22a-Taz* mice showed increased complex I subunit (NDUFB8) abundance, which could increase electron transport and therefore mitochondrial respiration and ATP/ADP ratio. Cardiolipin is important for respiratory supercomplex stability[11] and indeed our data indicate that tafazzin regulates complex I subunit abundance through effects on holoenzyme and supercomplex stability, not at the transcriptional level.

Whilst VSMC Taz increases features of plaque stability, we find that VSMC Taz^{H69Q} promotes apoptosis, atherosclerotic plaque development, and features of plaque vulnerability. We therefore suggest that loss of VSMC tafazzin is not merely a consequence of atherosclerosis but can have significant effects on plaque composition. We recognise that autosomal transgenic up-regulation of *TAFAZZIN* may have different effects to endogenous *Tafazzin*, which is an X-linked gene. The differences are difficult to predict; whilst *Tafazzin* expression is subject to X inactivation, there can also be dosage compensation via upregulation of X chromosome gene expression[50]. Similarly, autosomal transgenic up-regulation of *TAFAZZIN* could have different effects in male and females. However, our observations were similar in male and female mice, apart from for weight, where male mice showed increased weight compared with females across all genotypes.

We note that atherosclerosis is not a described feature of inherited *TAFAZZIN* mutations/Barth's syndrome. However, until recently, Barth's syndrome had a poor prognosis with a 5-year survival rate of just 22% pre 2000[51], whilst atherosclerosis usually does not manifest until the 5th decade. With supportive therapies, life expectancy for Barth's Syndrome patients has improved, and vasculopathy may prove to be a feature. Indeed, a variant in *PTPMT1*, which is involved in cardiolipin synthesis, is associated with hypertensio,n suggesting that defects in cardiolipin may have a role in vasculopathy[52,53].

Mitochondrial physiology is now well-recognised to have important effects on phagocytic cell function, including immune signalling and cytokine regulation[54]. Interestingly, we found that VSMC Taz^{H69Q} leads to increased plaque Mac3-positive area and increased aortic *Tnfa* expression. VSMCs from *Sm22a-Taz^{H69Q}/Apoe^{-/-}* mice show increased apoptosis but no change in *Tnfa* expression. Our findings are therefore consistent with increased *Sm22a-Taz^{H69Q}/Apoe^{-/-}* VSMC apoptosis leading to increased monocyte/macrophage recruitment and thus increased *Tnfa* expression. However, given that VSMCs manifest marked phenotypic plasticity, we cannot exclude the possibility that increased switching of *Sm22a-Taz^{H69Q}/Apoe^{-/-}* VSMCs to a macrophage-like phenotype leads to increased Mac3-positive area[55]. Mitochondrial dysfunction can also promote pro-inflammatory responses via other mechanisms. Monocytes can release mitochondria, either free or in vesicles, that induce inflammatory responses in endothelial cells[56]. MITOtempo, a mitochondrial

anti-oxidant, reduces this response. Since extracellular vesicles are found in plaques[57], mitochondria may act as signals in plaques that could be targeted for therapeutic benefit.

Overexpressing VSMC tafazzin reduces features of plaque vulnerability, therefore suggesting that tafazzin-targeted treatments may have beneficial effects on atherosclerosis. Our finding that miR-125a-5p downregulates tafazzin in hVSMCs and is induced by oxidised LDL in a NFκB -dependent manner raises the possibility that NFκB -targeted therapies could address both inflammation and mitochondrial dysfunction in atherosclerosis. Furthermore, there could be synergistic effects on endothelial cells, where NFkB signalling promotes atherosclerosis[58]. However, NFκB is also well-recognised to be important for VSMC survival and proliferation[59,60]. A targeted approach to selectively regulate NFκB binding to miR-125a-5p and pro-inflammatory genes may therefore be required to have beneficial effects on plaque burden and composition.

Whilst there are no current pharmacological therapies that directly target tafazzin, SS-31 (also known as elamipretide) is a small peptide that is reported to bind and stabilise cardiolipin[36]. No significant difference in MLCL/CL ratio was noted in cardiac mitochondria after SS31 treatment[61] but SS31 has been shown to bind with mitochondrial inner membrane proteins, including respiratory complex subunits[62]. SS-31 is of particular interest as human trials have shown that it is safe, well-tolerated and may improve cardiac parameters in patients with Barth's Syndrome[63]. Furthermore, SS-31 treatment in *Apoe^{-/-}* mice reduces aortic ROS, macrophage lipid uptake and atherosclerotic plaque burden[64]. However, the effects on VSMCs have not previously been studied. We highlight that SS-31 treatment improves mitochondrial respiration, reduces ROS and apoptosis in VSMCs from *Sm22a-Taz^{H69Q}/Apoe^{-/-}* mice. Taken together, SS-31 shows promising effects on VSMCs and atherosclerosis.

Our study has some limitations. First, we show reduced tafazzin expression in human plaques and VSMCs, and that microRNA 125a-5p downregulates tafazzin expression and is induced by oxidised LDL in a NFκB-dependent manner. Although there are likely multiple mechanisms of tafazzin regulation, both pre- and post-translational that we have not yet examined, our potential mechanism is biologically relevant as both oxidised LDL and activated NFκB are present in atherosclerotic lesions[65]. We recognise that we were unable to quantify miR-125a-5p in VSMC specifically using in situ hybridisation. However, by examining immediately adjacent sections, we have assessed tafazzin expression in SMA-positive cells and miR-125a-5p expression in the same region. We also recognise that our study cannot exclude the potential effects of miR-125a-5p on other mitochondrial proteins, and further experiments will be needed to specifically address this area. Second, we have generated mice expressing Taz or Taz^{H69Q} under the *Sm22a* promoter to examine the effects of VSMC tafazzin. Myeloid cells that differentiate and migrate into the plaque can also express VSMC promoters and markers, such as *Sm22a*[66]. However, the minimal *Sm22a* promoter we used is not expressed in peripheral blood or bone marrow cells[34]. Our observations were similar in male and female mice. However, a limitation of our study is that we combined these data for statistical analyses to

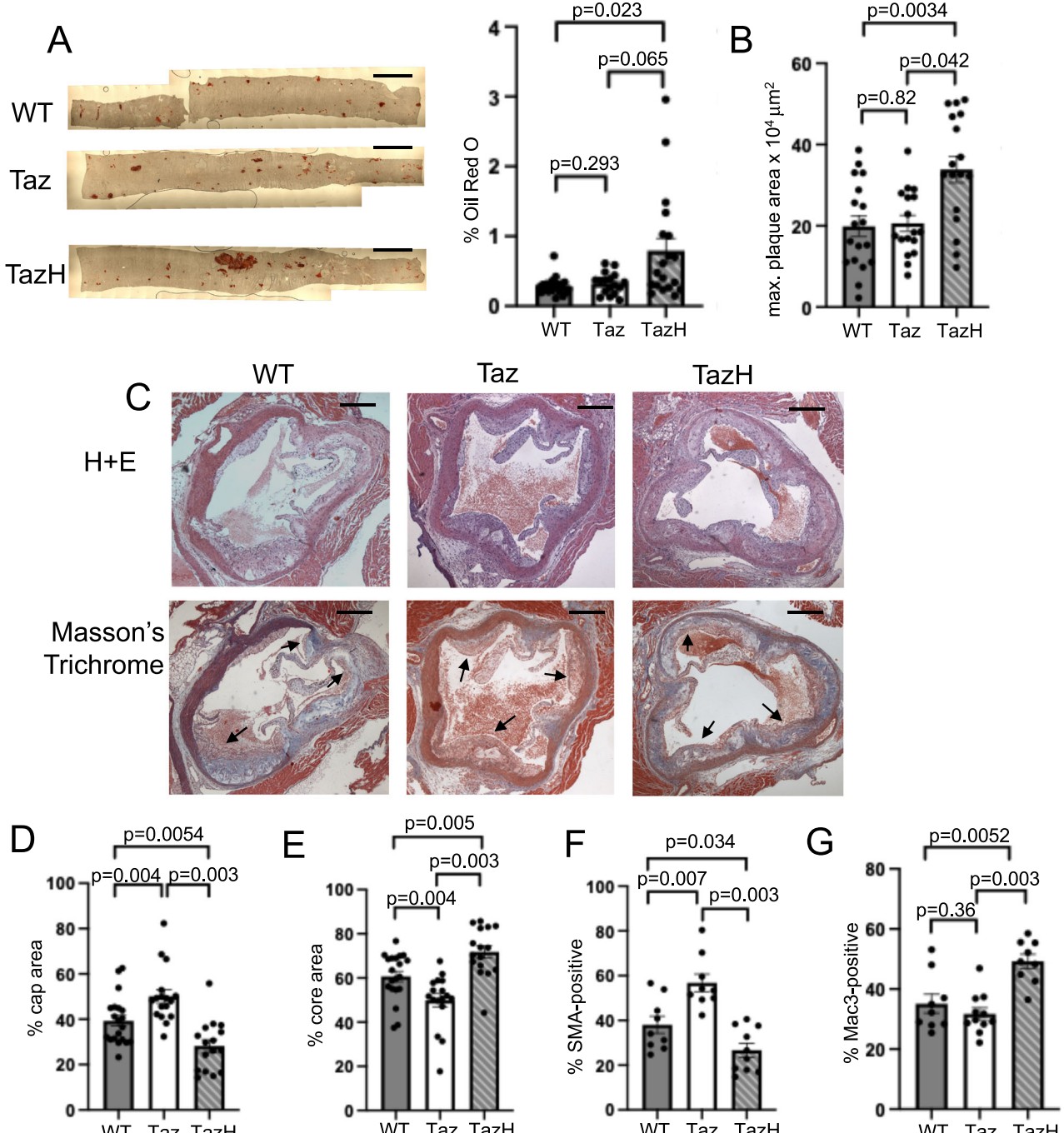

**Fig. 6 | VSMC tafazzin has important effects on atherogenesis. A** Oil Red O (ORO) staining of the descending aorta and quantification of ORO-positive area in wild-type *Apoe^{-/-}* (WT), *Sm22a-Taz/Apoe^{-/-}* (Taz) and *Sm22a-Taz^{H69Q}/Apoe^{-/-}* (TazH) mice after 14 weeks high fat diet (WT $n$ = 20, 11 male (M), 9 female (F); Taz $n$ = 17, 9 M, 8 F; TazH $n$ = 19, 10 M, 9 F; Kruskal-Wallis, Bonferroni Holm post hoc). Scale bars represent 4 mm. **B** Maximal (max) plaque area in aortic root sections of experimental mice (WT $n$ = 18, 9 M, 9 F; Taz $n$ = 17, 9 M, 8 F; TazH $n$ = 17, 8 M, 9 F; 1-way ANOVA, Bonferroni-Holm post hoc). **C** Aortic root plaques stained with haematoxylin and eosin (H + E) and Masson's Trichrome. Arrows indicate plaque caps. Scale bars represent 500 μm. **D, E** Quantification of percentage cap (**D**) and core (**E**) in WT/Taz/TazH mice (WT $n$ = 20, 11 M, 9 F; Taz $n$ = 17, 9 M, 8 F; TazH $n$ = 16, 7 M, 9 F; Kruskal Wallis, Bonferroni Holm post hoc). **F** Quantification of % smooth muscle actin (SMA) (WT $n$ = 9, 6 M, 3 F; Taz $n$ = 9, 5 M, 4 F; TazH $n$ = 10, 5 M, 5 F; 1-way ANOVA, Bonferroni-Holm post hoc) and **G** quantification of % Mac3 positive areas in experimental mice (WT $n$ = 9, 5 M, 4 F; Taz $n$ = 11, 6 M, 5 F; TazH $n$ = 9, 4 M, 5 F; 1-way ANOVA, Bonferroni-Holm post hoc.) For (**A**, **B**, **D**–**G**), $n$ = number of mice per group. Data are shown as mean ± SEM with multiplicity adjusted *P*-values. Source data are provided as a Source Data file.

ensure sufficient sample size. Third, we have used SS-31 to try to improve mitochondrial function in VSMCs expressing defective tafazzin. Whilst SS-31 is reported to bind to and stabilise cardiolipin, its mechanism of action is not fully elucidated[36]. However, we have shown that impaired mitochondrial respiration in VSMCs and increased vulnerability to apoptosis are potentially amenable to

pharmacological treatment. Given the importance of tafazzin/cardiolipin for cellular function, addressing the paucity of cardiolipin-directed therapies could be a target for future studies.

Our data is consistent with the following model: oxidised LDL accumulates within the arterial wall, which is taken up by VSMCs. MicroRNA 125a-5p is upregulated by oxidised-LDL via NFκB activation

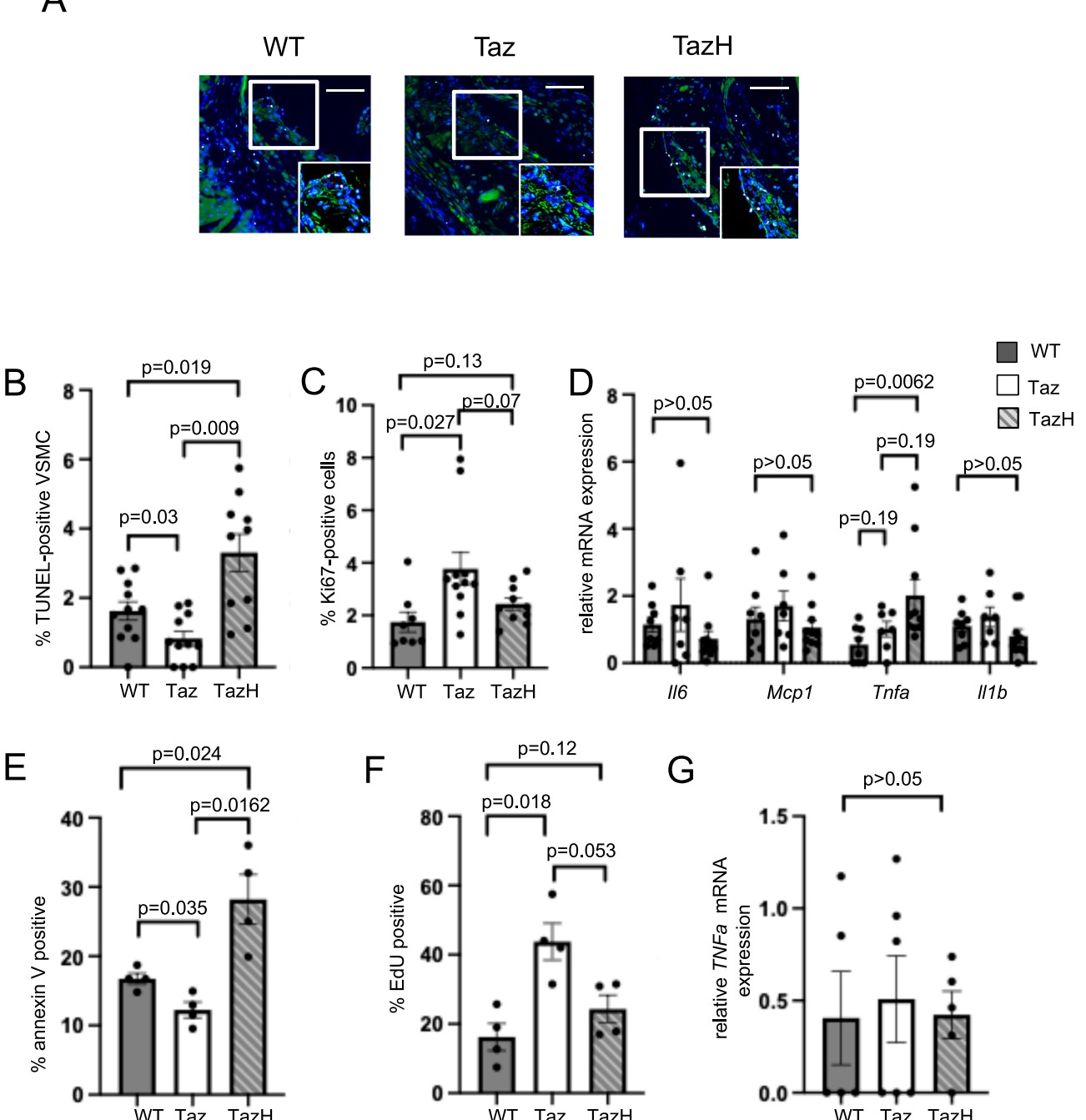

**Fig. 7 | Tafazzin protects against apoptosis. A** Aortic root sections of wild type *Apoe*[-/-] (WT), *Sm22a-Taz/Apoe*[-/-] (Taz) and *Sm22a-Taz*[H69Q]*/Apoe*[-/-] (TazH) mice after 14 weeks high fat diet stained with TUNEL (white), DAPI (blue) and SMA (green). Scale bars represent 100 μm. **B** Percentage of TUNEL-positive cells in aortic plaques of experimental WT, Taz and TazH mice (WT, Taz *n* = 11, 6 male (M), 5 female (F) mice; TazH *n* = 10, 6 M, 4 F; 1-way ANOVA, Bonferroni-Holm post hoc). **C** % of Ki67-positive cells in aortic plaques of experimental WT, Taz and TazH mice (WT *n* = 8, 5 M, 3 F; Taz *n* = 11, 6 M, 5 F; TazH *n* = 10, 5 M, 5 F; Kruskal Wallis, Bonferroni Holm post hoc). **D** QPCR for interleukin 6 (*Il6*), monocyte chemoattractant protein-1 (*Mcp1*), tumour necrosis factor-a (*Tnfa*) and interleukin 1b (*Il1b*) mRNA expression in aortas of experimental mice. Expression is shown relative to wild type,

normalised to *B2m* housekeeping gene (WT *n* = 8, 5 M, 3 F; Taz *n* = 7, 3 M, 4 F; TazH *n* = 10, 5 M, 5 F; Kruskal-Wallis, Bonferroni-Holm post hoc). **E, F: E** Percentage of annexin V-positive cells after tert-butyl hydrogen peroxide treatment and (**F**) percentage EdU-positive cells in VSMCs from wild type (WT), *Sm22a-Taz* (Taz) and *Sm22a-Taz*[H69Q] (TazH) mice (*n* = 4, 2 M, 2 F, 1-way ANOVA, Bonferroni-Holm post hoc). **G** QPCR for *Tnfa* mRNA expression in WT, Taz and TazH VSMCs. Expression is shown relative to wild type, normalised to *B2m* housekeeping gene (WT, TazH *n* = 5, 2 M, 3 F; Taz *n* = 6, 3 M, 3 F; Kruskal-Wallis, Bonferroni-Holm post hoc). For (**B**–**D**) *n* = number of mice per group; for (**E**–**G**) *n* = number of VSMC isolates per group. Data are shown as mean ± SEM with multiplicity adjusted *p*- values. Source data are provided as a Source Data file.

and downregulates tafazzin expression and abundance. Cardiolipin content, complex I abundance, mitochondrial respiration and ATP are decreased, reducing VSMC proliferation and promoting apoptosis. Whilst defective VSMC tafazzin promotes atherosclerosis, restoring

tafazzin protects against apoptosis and reduces features of plaque vulnerability.

In summary, we show that tafazzin is an important determinant of VSMC cardiolipin content, mitochondrial function, apoptosis and

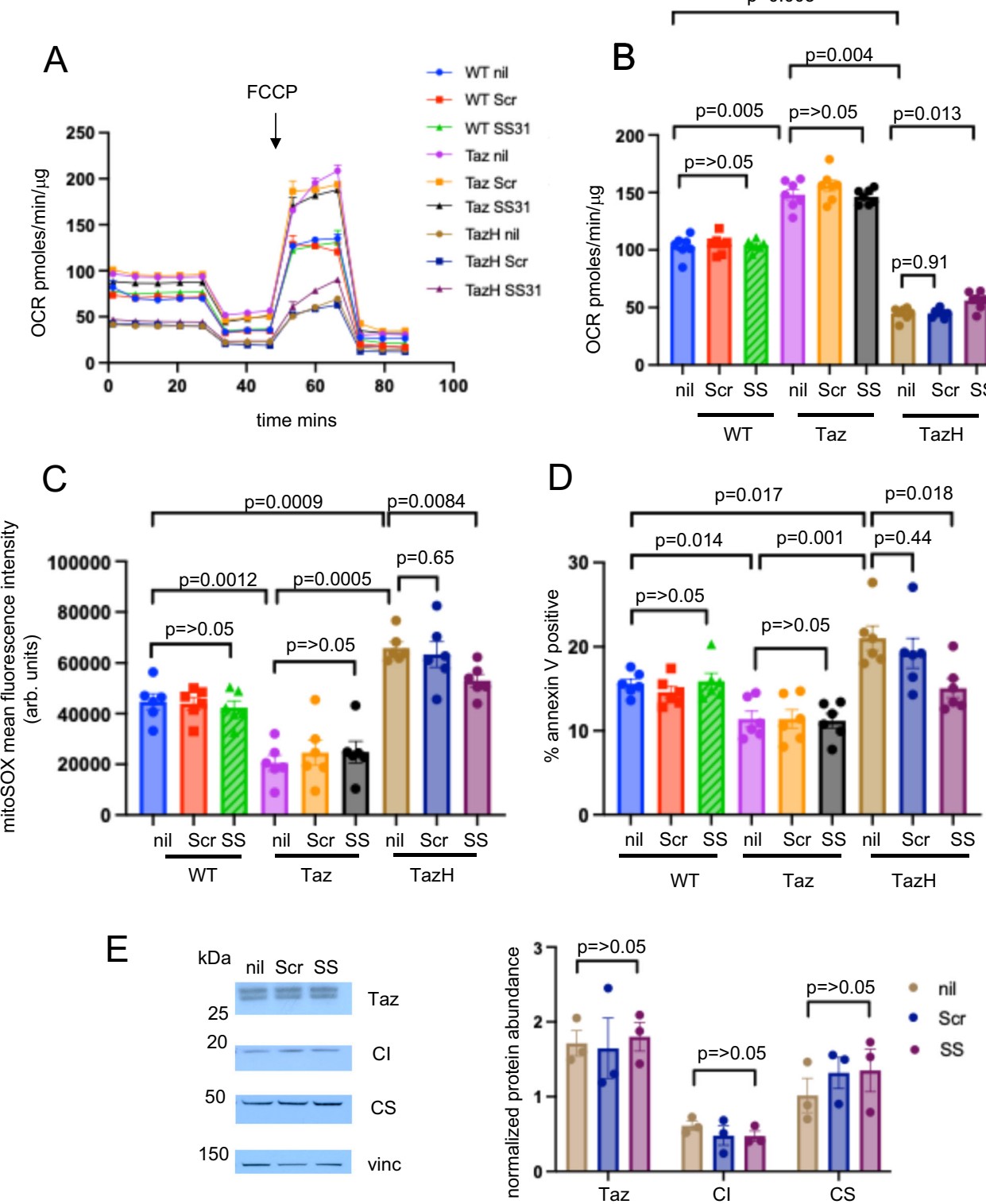

**Fig. 8 | SS-31 partially rescues mitochondrial respiration, reduces reactive oxygen species and apoptosis in tafazzin defective VSMCs. A** Representative Seahorse profiles of oxygen consumption rate (OCR) in VSMCs from wild type (WT), *Sm22a-Taz* (Taz), and *Sm22a-Taz^{H69Q}* (TazH) mice after nil treatment (nil) or after treatment with scrambled peptide control (Scr) or SS31 ($n = 3$, 1M, 2F). **B** Maximal OCR after FCCP in WT/Taz/TazH VSMCs after nil treatment (nil) or after treatment with scrambled peptide control (Scr) or SS31 (SS) ($n = 7$, 3 male (M), 4 female (F) VSMC donors). **C, D** MitoSOX mean fluorescence intensity fluorescence (**C**) and percentage of annexin V-positive cells after tert-butyl hydrogen peroxide

treatment (**D**) in nil, scrambled peptide control (Scr) or SS31 (SS) treated WT, Taz, TazH VSMCs ($n = 6$, 3 M, 3 F). arb. units = arbitrary units. **E** Western blot with quantification for tafazzin (Taz), complex I subunit (CI) and citrate synthase (CS) in nil, scrambled peptide control (Scr) or SS31 (SS) treated TazH VSMCs. Vinc = vinculin. Data normalised to citrate synthase for tafazzin and CI or vinculin for citrate synthase ($n = 3$, 2 M, 1 F). For (**A**–**E**) $n$ = number of independent experiments, 1-way ANOVA with Bonferroni-Holm post hoc. Data are shown as mean ± SEM with multiplicity adjusted $p$-values. Source data are provided as a Source Data file.

atherogenesis. Targeting tafazzin and/or its substrate cardiolipin are potential areas for therapeutics in atherosclerosis.

## Methods

The research was performed using protocols compliant with all relevant ethical regulations. Human samples were obtained using protocols approved by the Cambridgeshire 1 or Huntingdon Research Ethics Committees. All animal experiments were performed under the Animals (Scientific Procedures) Act 1986 Amendment Regulations 2012 (project licence P452C9545) and approved by Cambridge University Animal Welfare and Ethical Review Body (AWERB). Samples from both male and female human donors or mice were used, with numbers per group as indicated in the figure legends. Data disaggregated for sex is included in the source data.

### Human atherosclerosis

Human tissue was obtained using protocols approved by the Cambridgeshire 1 Research Ethics Committee (carotid and aortic tissue) or Huntingdon Research Ethics Committee (aortic tissue). Patients gave their written, informed consent for tissue collection. Plaque tissue was obtained from patients undergoing carotid endarterectomy, and normal aortic tissue from patients undergoing aortic valve surgery or root replacement.

### RNA extraction and QPCR

RNA extraction from human VSMCs, human tissues and mouse aorta was performed using the miRNAeasy kit (Qiagen) and the RNAeasy kit (Qiagen) used for mouse VSMC RNA extraction. cDNAs were synthesised using QuantiTect Reverse Transcriptase (Qiagen) or TaqMan™ MicroRNA Reverse Transcription Kit (Applied Biosystems) for mir-125a-5p experiments. QPCR was performed in triplicate in 20 µl reactions on a BioRad CFX96 thermocycler using BioRad SsoAdvanced™ Universal SYBR® Green Supermix (BioRad). Primers were used at 0.5 µM with sequences listed in Supplementary Table S1. TaqMan gene expression assays (Applied Biosystems) were used for mir-125a-5p experiments with TaqMan Fast Advanced Mastermix (Applied Biosystems). Data were collected and analysed using BioRad CFX Maestro 2.2.

### Western blotting

Protein lysates were prepared in NP40 lysis buffer, supplemented with proteinase inhibitor and phosphatase inhibitors. Protein concentration was determined using the BCA method (Pierce BCA protein assay kit, Thermo Fisher Scientific) and 20 µg of protein loaded onto 4–12% polyacrylamide gels. After methanol-based wet transfer, membranes were blocked in 5% milk and probed with primary antibodies as follows: Tafazzin (Ab105104, Abcam, 1:1000), tafazzin (sc365810, Santa Cruz Biotechnology, 1:1000), citrate synthase (ab96600, Abcam, 1:1000), total OXPHOS rodent antibody (Ab110413, Abcam, 1:500), tubulin (cs2148, Cell Signalling Technologies, 1:500), vinculin (Ab219649, Abcam, 1:1000), phospho NFκB p65 (cs3031, Cell Signalling, 1:500), NFκB p65 (cs8242, Cell Signalling, 1:1000), MFN1 (cs 14739, Cell Signalling, 1:1000), MFN2 (cs11925, Cell Signalling, 1:1000), DRP1 (611113, BD Transduction Laboratories, 1:1000), FIS1 (10956-1-AP, Proteintech, 1:1000), OPA1 (612606, BD Transduction Laboratories, 1:1000). Chemiluminescence detection was performed after incubation with HRP-labelled secondary antibodies.

### Histological analysis

After fixation in 10% formalin, human arteries and mouse ascending aortas were embedded in paraffin and sectioned at 5 µm intervals. For immunofluorescence analysis of human arterial sections and mouse aortic plaques specimens were de-waxed and microwaved in citrate buffer for antigen retrieval. Sections were permeabilised in 0.1% Triton X-100 and blocked in 3% bovine serum albumin (BSA). Human sections were then incubated with primary antibodies specific for anti-smooth

muscle actin (#14395-1-AP, Proteintech, 1 in 100) and tafazzin (sc365810, Santa Cruz Biotechnology,1:100) followed by Alexa Fluor™ 488 anti-rabbit antibody (#A-21206, Invitrogen, 1:200) and Alexa Fluor™ 594 anti-mouse antibody (#A-11032, Invitrogen, 1:200). Mouse sections were incubated with primary antibody specific for anti-smooth muscle actin (SMA, 1A4, Dako,1:100) followed by Alexa Fluor™ 488 anti-mouse antibody (#A-11001, Invitrogen, 1:200). Rabbit IgG (Abcam 172730) and Mouse IgG (Vector I2000) were used as isotype controls. Nuclei were stained with DAPI (Thermofisher), and sections mounted in Dako fluorescence mounting medium (Dako). Slides were imaged using a Leica SP8 scanning laser microscope (Leica) and images compiled from the z-stacks using "max projection" (LAS X software, Leica). For human arteries, the average percentage of SMA-positive cells expressing tafazzin of 3 random fields of view was used to quantify the % tafazzin positive VSMCs per sample.

Aortas from Myh11-CreERt2/Rosa26-Confetti/Apoe−/− mice were fixed in 4% formaldehyde, equilibriated in 30% sucrose/PBS, embedded in Tissue-Tek O.C.T Compound (Sakura), prior to cutting into 14 µm sections as described[18]. For immunofluorescence analysis, sections were permeabilised in 0.5% Triton X-100 and blocked in M.O.M. Blocking Reagent (Vector Laboratories). Sections were then incubated with tafazzin-Alexa Fluor 647™ antibody (sc365810, Santa Cruz Biotechnology,1:500) with mouse IgG2b- Alexa Fluor 647™ antibody (sc24638, Santa Cruz Biotechnology) used as an isotype control. Nuclei were stained with DAPI (Thermofisher), and sections mounted in RapiClear 1.52 (Sunjin Lab). Slides were imaged using a Leica STELLARIS 8 confocal microscope and images compiled from the z-stacks using "max projection" (LAS X software, Leica). The number of tafazzin-positive, Confetti-positive VSMCs was expressed as a percentage of the total number of Confetti-positive VSMCs for each region.

Mouse ascending aortas were stained with hematoxylin and eosin or Masson's Trichrome. Slides were viewed using an Olympus BX61 fluorescence microscope (Olympus Life Science) with a Lumenera Infinity 3 Microscopy Camera (Teledyne Lumenera). Images were captured and analysed using Image-Pro Insight (Media Cybernetics) software by an observer blind to the experimental group. Total plaque area was measured, and fibrous cap and necrotic core areas determined. Fibrous cap areas were defined as the regions rich in VSMCs (red), overlying the cholesterol rich, acellular core (Supplementary Fig. 16). Cap and core areas were expressed as a percentage of plaque area.

For immunohistochemical analysis of aortic root plaques, specimens were de-waxed and microwaved in citrate buffer for antigen retrieval. Endogenous peroxidase activity was quenched with hydrogen peroxidase before sections were blocked in bovine serum albumin (BSA). Sections were then incubated with primary antibodies specific for anti-smooth muscle actin (SMA, 1A4, Dako,1:500), Mac3 (553322, BD Biosciences,1:400) and Ki67 (Ab16667, Vector Laboratories,1:100). Mouse IgG (Vector I2000) and rat IgG (Vector I4000) were used as isotype controls. Sections were stained with biotinylated secondary antibodies detected with Avidin/Biotinylated enzyme Complex (ABC) reagents (Vector Laboratories), visualised with diaminobenzidene (DAB, Vector Laboratories) or Vector Blue. Digital images were acquired and, for mouse ascending aortas, stained areas were expressed as a percentage of total plaque area (for SMA and Mac3) or as a percentage of the total number of lesional cells (Ki67 staining).

For apoptosis detection, Click-iT™ Plus Terminal deoxynucleotidyl transferase deoxyuridine triphosphate (dUTP) Nick End Labelling (TUNEL) assay was used (Invitrogen) following the manufacturer's instructions. Briefly, mouse aortic plaques specimens were de-waxed and microwaved in citrate buffer for antigen retrieval. Sections were then fixed in 4% paraformaldehyde and permeabilized using Proteinase K. Sections were equilibrated with TdT Reaction Buffer and then incubated with TdT reaction mixure for 60 min at 37 °C followed

by Click-IT™ Plus TUNEL reaction cocktail for 30 minutes at 37 °C, protected from light. Slides were washed in 3% BSA in PBS, followed by PBS, and immunofluorescence staining for smooth muscle cell actin performed as described above. For negative controls, TUNEL staining was performed with the TdT reaction omitted (Supplementary Fig. 17A). The number of TUNEL-positive VSMCs was expressed as a percentage of the total number of plaque VSMCs.

For Oil Red O analysis, descending aortas were dissected and incised along the longitudinal axis. The samples were rinsed with de-ionised water and 60% isopropranolol for 30 s prior to incubation with 60% Oil Red O stain for 15 mins. Slides were viewed using an Olympus BX61 fluorescence microscope (Olympus Life Science) with a Lumenera Infinity 3 Microscopy Camera (Teledyne Lumenera). Images were captured and analysed using Image-Pro Insight (Media Cybernetics) software by an observer blind to the experimental group. The percentage-stained area was determined.

### miRNAscope

Human arteries were fixed, embedded, sectioned and underwent dual staining for tafazzin and smooth muscle cell actin as described above. RNA in situ hybridisation was performed on the adjacent section using miRNAscope™ assay (Advanced Cell Diagnostics) as per the manu-facturer's instructions. Sections were dewaxed, and epitope retrieval performed with ER2 buffer (ACD). Endogenous peroxidase activity was quenched with hydrogen peroxidase, and sections permeabilized with ACD enzyme (Protease III) before probe hybridisation. Hsa-miR-125a5p coupled to Opal 570 (ACD) was used, and nuclei were stained with DAPI prior to imaging with Akoya PhenoImager HT imager (Akoya Biosciences). Images were analysed in QuPath with the subcellular detection feature[67]. The average percentage of miR-125a5p positive cells of 3 random fields of view was used to quantify the % miR-125a5p positive cells per sample.

### Cell culture

Human VSMCs (hVSMCs) were cultured from aortic samples, and plaque VSMCs from carotid endarterectomy samples. For aortic tissue the endothelial and adventitial layers were removed first. Aortic and plaque samples were then divided into small pieces (2-3mm²) and cultured in 6 well plates for 1-2 weeks to allow cells to emerge. Aortic and plaque hVSMCs were cultured in VSMC medium (PromoCell GmbH, SMC-GM2 C22062) with 5% Foetal Bovine Serum, supplemented with 100U/ml penicillin, 100 µg/ml streptomycin. hVSMCs were passaged when 90% confluent, reseeding at 13,333 cells/cm², and studied at passages 2–8.

Mouse VSMCs (mVSMCs) were isolated from aortas using enzymatic digestion with type II collagenase (Gibco) and elastase (Worthington Biochemical Corporation). The adventitia was removed and the aorta digested for 1 hour at 37 °C. VSMCs were then gently pipetted and plated in a 48-well plate in DMEM/F12 medium (Gibco) supplemented with 20% Foetal bovine serum, 100U/ml penicillin, 100 µg/ml streptomycin. mVSMCs were split 1:2 when 90% confluent and at 3 weeks switched to DMEM (Sigma) supplemented with 20% Foetal bovine serum, 100 µ/ml penicillin, 100 µg/ml streptomycin, 2 mM glutamine. mVSMCs were studied at passages 2-12.

VSMCs were identified by immunocytochemistry for alpha-smooth muscle actin, smooth muscle myosin heavy chain, and also examined for the expression of the macrophage marker CD68. VSMCs were plated onto 8 well chamber slides, fixed with 4% paraformaldehyde and permeabilised with 0.5% Triton™ X-100. The slides were blocked in a solution of 3% bovine serum albumin before incubation with antibodies as follows: anti-Actin, α-smooth muscle-Cy3™ (C6198, Merck,1:200), smooth muscle myosin heavy chain (Ab124679, Abcam, 1:100), anti-human CD68 (M0814, Dako, 1:200) followed by Alexa Fluor™ 488 anti-rabbit antibody (#A-21206, Invitrogen, 1:200), Alexa Fluor™ 594 anti-mouse antibody (#A-11032, Invitrogen, 1:200) or anti-

mouse CD68 (ab125212 Abcam, 1;200) followed by Alexa Fluor™ 594 anti-rabbit antibody (#A-11012, Invitrogen, 1:200). Slides processed without primary antibodies served as negative controls. Nuclei were stained with 1 µg/mL Hoechst 33342 (Sigma) and slides viewed using an Olympus BX61 fluorescence microscope (Olympus Life Science) with a Lumenera Infinity 3 Microscopy Camera (Teledyne Lumenera). Images were captured using Image-Pro Insight (Media Cybernetics). THP-1 cells (TIB-202, American Type Culture Collection), treated with 50 ng/ml phorbol 12-myristate 13-acetate for 72 h, or mouse bone marrow-derived macrophages were used as positive controls for CD68 staining.

VSMCs were treated with 30 µmol/L tert-butyl hydrogen peroxide (Sigma) for 16 hours, 100 µg/ml LDL (Sigma) for 24 h or 100 µg/ml oxidised LDL for 24 h (Kalen Biomedical). Cells were pre-treated with 5 µM Bay 11-7082 for 2 h or transfected with siRNA or microRNA for 48 hours before addition of oxLDL.

### scRNAseq analysis

Integrated scRNA-seq data, including profiles from different studies was obtained from https://zenodo.org/records/14007461[14]. The log normalised data was subset to atherosclerotic samples from coronary[15] and carotid arteries[16,17] using Seurat v.5.1.0[68,69] in R v.4.4.1, resulting in a dataset with 70,025 cells. The author-provided metadata ('Main_Cell_Types')[14] was used for cell type annotation. Differential expression analysis for *TAFAZZIN* was performed between VSMCs ("VSMC" cluster) and either endothelial cells (clusters annotated as "Endothelial", "Endothelial (DKK2 + )", "Lymphatic Endothelial") or macrophages (clusters annotated as "Macrophage", "pDC", "cDC1") with the Wilcoxon Rank Sum test.

### MicroRNA, siRNA and lentiviral transfections

For human microRNA 125a-5p experiments, human VSMCs were transfected with 25 nM mir-125a-5p mimic (C-300624-05-0005, Horizon Discovery), 50 nM mir-125a-5p inhibitor (IH-300624-06-0005, Horizon) or control (CN-001000-01, Horizon).

For transient cell silencing, hVSMCs were transfected with 25 nM tafazzin siRNA (ON-TARGET*plus* SMART Pool, L-009608-00-0005, Dharmacon), 25 nM NFκB p65 siRNA (ON-TARGET*plus* SMART Pool, L-003533-00-0005, Dharmacon) or siRNA control (ON-TARGET*plus* SMART Pool, D-001810-10-05, Dharmacon). Both microRNA and siRNA transfections were performed using 200,000 VSMCs per well of a 6 well plate in antibiotic-free hVSMC media with Lipofectamine RNAiMAX transfection reagent (Invitrogen). After 24 h the media was changed to full hVSMC media, and the cells studied at 48 h.

Lentiviruses were generated for stable overexpression of *TAFAZZIN* or Taz^H69Q in mVSMCs using the following plasmids: pLenti-Tafazzin-Myc-DDK (OriGene), pMDLg/pRRE, pMD2.G, pRSV-Rev (all gifts from Didier Trono, Addgene plasmid #12251, 12259, 12253). PCR-based mutagenesis was used to generate the pLenti-Taz^H69Q-Myc-DDK vector (PFU Hot Start II, Agilent Technologies) using 30 ng of input DNA with the following primers at 10 µM: 5' GTG TCC AAT CAG CAG TCC TGC AT 3' (forward), 3' ATG CAG GAC TGC TGA TTG GAC AC 5' (reverse). The lentiviral vectors were then used to transfect HEK293FT cells (R7007, Invitrogen) using *Trans*IT-LT1 transfection Reagent (Mirus Bio LLC). Lentivirus-containing medium was harvested and pooled at 48 and 72 h, and concentrated down using Lenti-X Concentrator (Takara). For VSMC transduction, VSMCs were infected with lentivirus (MOI 1-10) with 8µg/ml polyberene. After 48 h, cells were selected for puromycin resistance with 1µg/ml puromycin.

### Dual luciferase assay

For the dual luciferase assay vectors were used as follows: empty vector control (CmiT000001-MT06, GeneCopoeia™) and dual luciferase vector expressing wild type *TAFAZZIN* 3'UTR sequences inserted downstream of the firefly luciferase reporter gene (HmiT110411-MT06, GeneCopoeia™). PCR-based mutagenesis was used to generate the

mutated *TAFAZZIN* 3'UTR vector (PFU Hot Start II, Agilent Technologies) using 30 ng of input DNA with the following primers at 10 μM: 5' ACTACCTCCTCCTAGATGGCCGTTG 3' (forward), 3' CAACGGCCATC-TAGGAGGAGGTAGT 5' (reverse). Human VSMCs were co-transfected with control, wild type or mutated *TAFAZZIN* 3'UTR dual luciferase vector and either control or 25 nM mir-125a-5p mimic (Horizon) with *Trans*IT-LT1 transfection Reagent (Mirus Bio, 1.5 μl/0.5 μg vector) and lipofectamine RNAiMAX transfection reagent in antibiotic-free hVSMC media. After 48 h cells were lysed and firefly and renilla luminescence determined using Dual-Glo Luciferase Assay System (Promega) according to the manufacturer's instructions. Briefly, to measure firefly luciferase, cells were incubated in Dual-Glo® Luciferase Reagent for 10 min and luminescence measured. Renilla luciferase activity was then determined after the cells were incubated with Dual-Glo® Stop & Glo® Reagent for 10 min. The ratio of firefly/renilla luminescence was calculated. Data were collected using Omega software 5.50R4 and analysed in MARS 3.32 R5W (BMG Labtech).

### NFκB p65 Activity ELISA

Whole cell lysates of hVSMCs were prepared in RIPA buffer supplemented with proteinase and phosphatase inhibitors. Protein concentration was determined using the BCA method and 4 μg of protein assayed for NFκB p65 DNA binding activity, using the Trans-AM NFκB p65 Transcription Factor Assay kit (Active Motif) according to the manufacturer's instructions. Briefly, samples were mixed with complete lysis buffer (Active Motif) and then incubated with complete binding buffer (Active Motif) for 60 min at room temperature. After 3 washes, samples were incubated with the supplied NFκB p65 antibody, followed by incubation with horse radish peroxidase conjugated secondary antibody; both antibody incubations were performed for 60 min at room temperature. Developing solution was added, and chemiluminescence measured. Data were collected using Omega software 5.50R4 and analysed in MARS 3.32 R5W (BMG Labtech).

### Mitochondrial morphology assessment

Immunofluorescence was performed to assess mitochondrial morphology[70]. Cells were fixed using 5% paraformaldehyde (PFA) in phosphate-buffered saline (PBS) at 37 °C for 15 min, followed by three PBS washes and quenching with 50 mM ammonium chloride in PBS. The cells were permeabilized in 0.1% Triton X-100 in PBS for 10 minutes, followed by three PBS washes. The cells were blocked using 10% Foetal bovine serum (FBS) in PBS, followed by incubation with anti-TOMM20 (#311802-1-AP, Proteintech, 1 in 1000) or anti-TOMM70 antibodies (#14528-1-AP, Proteintech, 1 in 1000) in 5% FBS in PBS for 2 h at room temperature or overnight at 4 °C. Post-incubation, cells underwent three washes in 5% FBS in PBS, followed by incubation with Alexa Fluor™ 488 anti-rabbit antibody (#A-11008, Invitrogen, 1 in 1000) prepared in 5% FBS in PBS. After three PBS washes, coverslips were briefly rinsed in water and mounted onto slides using Dako fluorescence mounting medium (Dako) for spinning disk confocal microscopy.

Cells were then imaged using a 100 × objective lens (NA 1.4) on a Nikon Eclipse TiE inverted microscope using an Andor Dragonfly 500 spinning disk system and a Zyla 4.2 PLUS sCMOS camera (Andor) integrated with Fusion software (Oxford instruments). For the analysis of mitochondrial morphology, a total of nine stacks, each with a slice thickness of 0.2 μm, were acquired using the 100× objective. Uniform conditions, including laser intensity and exposure time, were maintained throughout the acquisition of images from the same experiment. Subsequently, these images were compiled via 'max projection,' and mitochondrial morphology was quantified in a semi-automated and unbiased manner using the imageJ macro MitoMAPR as previously described[71]. In brief, three 225 μm² mitochondrial ROIs were selected from each cell per max projected image and saved as input for the MitoMAPR macro. For mouse VSMCs, ~60 cells were analysed per

replicate of each condition. For human VSMC,s ~40 cells were analysed per replicate of each condition. Three ROIs were selected per cell as indicated above. The saved ROIs were batch processed by the algorithm and the output i.e., mitochondrial number, length and area were plotted using Graphpad Prism. The experiment was repeated three times for all conditions and cell types. All images shown have been smoothed one time using the "smooth" function in ImageJ.

### Cardiolipin analysis

Cardiolipin quantification was performed based on liquid chromatography-mass spectrometry[72]. Vascular smooth muscle cells were harvested by trypsinisation, snap frozen in liquid nitrogen and stored at −80 °C before use. Stable isotope-labelled internal standards purchased from Sigma Aldrich (Haverhill, Suffolk, UK) include: N-palmitoyl-d31-D-erythro-sphingosine (abbreviated to IS_Cer_16:0-d31); order number: 868516 P, 1-palmitoyl-d31-2-oleoyl-sn-glycero-3-phosphate (abbreviated to IS_PA_34:1-d31); order number: 860453 P, 1-palmitoyl-d31-2-oleoyl-sn-glycero-3-phosphocholine (abbreviated to IS_PC_34:1-d31); order number: 860399 P, 1-palmitoyl-d31-2-oleoyl-sn-glycero-3-phosphoethanolamine (abbreviated to IS_PE_34:1-d31); order number: 860374 P, 1-palmitoyl-d31-2-oleoyl-sn-glycero-3-[phospho-rac-(1-glycerol)] (abbreviated to IS_PG_34:1-d31); order number: 860384 P, 1-palmitoyl-d31-2-oleoyl-sn-glycero-3-phosphoinositol (abbreviated to IS_PI_34:1-d31); order number: 860042 P, 1,2-dimyristoyl-d54-sn-glycero-3-[phospho-L-serine] (abbreviated to IS_PS_28:0-d54); order number: 860401 P, N-palmitoyl-d31-D-erythro-sphingosyl-phosphorylcholine (abbreviated to IS_SM_34:1-d31); order number: 868584 P. Stable isotope-labelled internal standards purchased from QMX Laboratories Ltd. (QMX Laboratories Ltd., Thaxted, Essex, UK) include: Heptadecanoic-d33 acid (abbreviated to IS_FA_17:0-d33); order number: D-5261, N-tetradecylphosphocholine-d42 (abbreviated to IS_LPC_14:0-d42); order number: D-5885, Glyceryl tri(pentadecanoate-d29) (abbreviated to IS_TG_45:0-d87); order number: D-5265, Butyryl-d7-L-carnitine (abbreviated to IS_Car_4:0-d7); order number: D-7761, Hexadecanoyl-L-carnitine-d3 (abbreviated to IS_Car_16:0-d3); order number: D-6646. To a sample amount equalling 500,000 vascular smooth muscle cells, we added 650 μL of chloroform to each sample, followed by thorough mixing. Then 100 μL of the lipid internal standard (1-10 μM in methanol) and 150 μL of methanol was added to each sample, followed by 400 μL of acetone. The samples were vortexed and sonicated for 20 minutes (Advantage-Lab, Switzerland). The samples were then centrifuged for 10 min at ~20,000 × g to pellet any insoluble material. The supernatant was pipetted into separate 2 mL screw cap amber-glass auto-sampler vials (Agilent Technologies). The extracts were dried down to dryness using a Concentrator Plus system (Eppendorf) run for 60 min at 60 degrees Celsius.

For lipidomics LC-MS analysis, the samples were reconstituted in 100 μL of 2: 1: 1 (propan-2-ol, acetonitrile and water, respectively) then thoroughly vortexed. The reconstituted sample was transferred into a 250 μL low-volume vial insert inside a 2 mL amber glass auto-sample vial ready for liquid chromatography with mass spectrometry detection (LC-MS) lipidomics analysis. Full chromatographic separation of intact lipids was achieved using Waters Acquity H-Class HPLC System (Waters) with the injection of 10 μL onto a Waters Acquity Premier UPLC® CSH C18 column; 1.7 μm, I.D. 2.1 mm × 50 mm, maintained at 55 degrees Celsius. Mobile phase A was 6:4, acetonitrile and water with 10 mM ammonium formate. Mobile phase B was 9:1, propan-2-ol and acetonitrile with 10 mM ammonium formate. The flow was maintained at 500 μL per minute through the following gradient: 0.00 minutes_40% mobile phase B; 1.5 minutes_40% mobile phase B; 8.00 minutes_99% mobile phase B; 10.00 minutes_99% mobile phase B; 10.10 minutes_40% mobile phase B; 12.00 minutes_40% mobile phase B. The sample injection needle was washed using 9:1, propan-2-ol and acetonitrile [strong wash] and 2: 1: 1 (propan-2-ol, acetonitrile and water) [weak wash]. The mass spectrometer used was the Thermo Scientific Q-Exactive Orbitrap with a heated

electrospray ionisation source (Thermo Fisher Scientific). The mass spectrometer was calibrated immediately before sample analysis using a positive and negative ionisation calibration solution (recommended by Thermo Scientific). Additionally, the mass spectrometer scan rate was set at 4 $Hz$, giving a resolution of 35,000 (at 200 $m/z$) with a full-scan range of $m/z$ 120 to 1,800 with continuous switching between positive and negative mode. Samples were analysed in groups of ten samples flanked by solvent blank samples, internal standard blank samples and quality control samples (Commercially available blank human serum was purchased from BioIVT (Royston, Hertfordshire, UK; order number: HUMANSRMPNN)). A total of 26 VSMC samples were analysed, representing 5 biological replicates per group, except for EV VSMCs, which had 6 biological replicates per group. At the start of the sequence, a mixture of cardiolipins (Cardiolipin solution from bovine heart; order number: C1649 from Cayman Chemical Company (Cambridge Bioscience, Cambridge, UK)) was run for the correct retention times.

The data processing was done using Thermo Scientific Xcalibur (Version 4.1.31.9) in the Quan browser. The data processing involved the integration of the extracted ion chromatogram peaks for each target lipid species (within a mass accuracy of 5 ppm range) and the stable isotope labelled internal standards at their expected retention time (full list of cardiolipins and internal standards with their retention time and m/z are in the Source Data). The area ratio response of the target lipid to the corresponding internal standard were converted into nmole results normalised to cell count, and subjected to blank correction and comprehensive quality checking before further statistical analysis. Statistical analysis was performed using GraphPad Prism (Version 10.2.3) with a two-sided unpaired $t$ test used for siTAZ vs. control comparisons, 1-way ANOVA, Bonferroni-Holm post hoc for EV, Taz and TazH comparisons.

### Seahorse assay

Mitochondrial respiration was determined using a Seahorse XF96e extracellular flux analyser (Agilent). To achieve a high degree of confluence, 12,000 VSMCs per well were plated into XF96 microplates and basal oxygen consumption rate (OCR) determined after allowing 24 h for adherence. Sequential additions of oligomycin (1 μg/ml), carbonyl cyanide 4-(trifluoromethoxy)phenylhydrazone (FCCP, 12 μmol/L) and antimycin/rotenone (10/1 μmol/L) were performed. Oxygen consumption rate (OCR) was normalised to protein content as determined by the BCA method. Data were collected and analysed using Seahorse Wave software (Agilent).

### ATP/ADP ratio

10,000 VSMCs per well were plated into 96-well plates. Positive control cells were treated with oligomycin 1 μg/ml (Sigma) for 20 minutes at 37ºC. The cells were then lysed and ATP/ADP ratio determined using the ADP/ATP Ratio Assay Kit (Sigma) as per the manufacturer's instructions. Briefly, to determine intracellular ATP cells were incubated with ATP Reagent and luminescence measured (Reading A). After 10 min, luminescence was measured again (Reading B) before the cells were incubated with ADP reagent. After 1 minute, luminescence was determined as a measure of ADP (Reading C). The ATP/ADP ratio was calculated as Reading A/(Reading C − Reading B). Data were collected using Omega software 5.50R4 and analysed in MARS 3.32 R5W (BMG Labtech).

### MitoSOX Assay

For mitochondrial ROS assessment, VSMCs were incubated in 1 μM MitoSOX (Invitrogen #M36008) for 20 min 37 °C. For positive controls, VSMCs were incubated with 10 μM antimycin A (Sigma) for 20 minutes at 37 °C. After washing twice with PBS, cells were analysed on a BD Accuri C6 flow cytometer (BD Biosciences). Cells were selected based on FSC/SSC plots, and single cells identified on FSC-H/FSC-A plots. The gate for MitoSOX positive cells was identifed based on cells

that had not been incubated with MitoSOX (Supplementary Fig. 18A). Mean FL2H fluorescence was determined using BD CSampler software (BD Biosciences).

### Mitochondrial membrane potential

For live-cell videomicrography to examine mitochondrial membrane potential cells were seeded overnight in μ-Slide 8 Well high Glass Bottom (Ibidi, Grafelfing, Germany). Prior to imaging, the cells were incubated in tetramethyl rhodamine ethyl ester (TMRE) and Mito-Tracker Deep Red (MTr) (Thermo Fisher; 2.5 nM TMRE and 3 nM MTr for human VSMC, and 4 nM TMRE and 3 nM MTr for mouse VSMC) for 20 minutes. Cells were imaged using a 60× Apochromat TIRF objective lens (NA 1.49) on a Nikon Eclipse TiE inverted microscope using an Andor Dragonfly 500 spinning disk system and a DU897 Ixon camera (Andor) integrated with Fusion software (Oxford Instruments). During imaging, the cells were housed within a stage-top live-cell imaging chamber (OkoLab, Naples, Italy) under physiological conditions (37 °C ± 0.5 °C, 5% $CO_2$, and 95% humidity). At the end of the incubation period, the cells were illuminated using the 561 nm and 647 nm ILE laser lines, and images were acquired every 10 seconds. Channel-specific dichroic settings were employed during acquisition to prevent cross-channel bleedthrough. To induce depolarisation, the mitochondrial uncoupler carbonyl cyanide 4-(trifluoromethoxy)phenylhydrazone (FCCP; 15 μM for human VSMC and 10 μM for mouse VSMC) was added to the cells in situ at $t = 200$ s, and images were acquired until t = 600 seconds. The acquired timelapse images were then processed for intensity quantification using ImageJ. In brief, the images were corrected for background, and three 900 μm² ROIs were selected per image. For mouse VSMCs, ~ 30 cells were analysed per replicate of each condition. For human VSMCs, ~ 20 cells were analysed per replicate of each condition. Three ROIs were selected per cell as indicated above. The fluorescence intensity of these ROIs in both TMRE and MTr channels over the duration of the video was determined using the "Plot z-axis profile" function of ImageJ. The TMRE intensity values were normalised to the intensity of MTr, followed by normalisation against intensity at $t = 0$ s. The data were then plotted using GraphPad Prism. Four replicates (four wells) per condition was imaged for each experiment. The experiment was repeated thrice for all conditions and cell types.

### EdU, Annexin V and propidium iodide flow cytometric analysis

VSMCs were analysed using the Click-iT EdU kit (Thermo Fisher) to examine cellular proliferation. To avoid a high degree of confluence, 130,000 VSMCs per well were plated in 6-well plates. Cells were incubated with 10 μM EdU for 24 h and then washed, fixed and permeabilised before incubation with the Click-iT reaction cocktail (prepared as per manufacturer recommendations). Cells were analysed on a BD Accuri C6 flow cytometer (BD Biosciences). Cells were selected based on FSC/SSC plots, and single cells identified on FSC-A/FSC-H plots. The gate for EdU positive cells was identifed based on cells that had not been incubated with EdU (Supplementary Fig. 18B). The percentage EdU positive cells was determined using BD CSampler software (BD Biosciences).

To assess VSMC apoptosis, 100,000 cells per well were plated into 24-well plates. Prior to staining, cells were incubated with 30 μM tert-butyl hydroperoxide (t-BHP) for 16 h. Cells were then harvested, washed in cold PBS and stained with propidium iodide and Annexin V Alexa Fluor 488 (Thermo Fisher Scientific) for 15 min at room temperature. Samples were then immediately analysed on a BD Accuri C6 flow cytometer (BD Biosciences). FSC/SSC plots were used to identify all cells but exclude debris. The gate for positive cells was determined based on unstained cells (Supplementary Fig. 18C). The percentage annexin V-positive cells was determined using BD CSampler software (BD Biosciences).

## Blue native page gel electrophoresis and immunoblot analysis

$5 \times 10^6$ VSMCs were harvested by trypsinisation, washed twice in PBS, and resuspended in 200 µl cold PBS supplemented with protease inhibitors (PBS/PI). The VSMCs were treated with 8 mg/ml digitonin (Millipore 300410) on ice for 10 minutes, and the reaction stopped by the addition of 1 ml of PBS/PI. Permeabilised cells were centrifuged at 10000 g for 5 minutes at 4 °C, and the resulting pellet washed with 1 ml cold PBS/PI before resuspension in 100 µl of solubilisation buffer (1.5 M aminocaproic acid, 50 mM Bis-Tris, pH 7). Complexes were extracted with 10 µl of 10% digitonin on ice for 5 min, then centrifuged at $18000 \times g$ for 30 min at 4 °C. The supernatants were transferred to a clean tube and stored at −80 °C until use. Protein concentration was determined using the BCA method. Sample buffer (750 mM aminocapronic acid, 50 mM Bis-Tris. 0.5 mM EDTA, 5% Coomassie Brilliant Blue G250) was added to 15 µg of protein and the samples loaded onto an Invitrogen NativePage 3–12% Bis-Tris gel (10064012). Electrophoresis was performed using the anode running buffer (Invitrogen, BN2001) and cathode anode buffer containing 5 % buffer additive (Invitrogen, BN2002). Samples were initially ran for 130 V for 1 hour and then resolved at 30 V for 20 hours using 0.5% cathode buffer. After transfer in bicarbonate buffer (3 mM $Na_2CO_3$ and 10 mM $NaHCO_3$), membranes were blocked in 5% milk and probed with primary antibodies as follows: NDUFB8 (14794-1-AP, Proteintech, 1:1000), SDHA (ab14715, abcam, 1:1000), UQCRC2 (14742-1-AP, Proteintech, 1:1000), MTCO2 (55070-1-AP, Proteintech, 1:1000), ATP5A1 (660371-1-Ig, Proteintech, 1:1000). Chemiluminescence detection was performed after incubation with HRP-labelled secondary antibodies.

## Experimental Animals and atherosclerosis protocols

All animal experiments were performed under the Animals (Scientific Procedures) Act 1986 Amendment Regulations 2012 (project licence P452C9545) and approved by Cambridge University Animal Welfare and Ethical Review Body (AWERB).

All animals used in the study were mice (*Mus musculus*). Transgenic mice overexpressing either VSMC specific *TAFAZZIN* or Taz[H69Q] cDNA were generated as follows: Human *TAFAZZIN* cDNA from pLenti-Tafazzin-Myc-DDK (Origene) was subcloned between the minimal *Sm22a* promoter and polyA sequence in pBluescript vector. Site directed mutagenesis (PFU Hot Start II, Agilent) was used to create the *Taz*[H69Q] mutant using 30 ng of input DNA with the following primers at 10 µM: 5' GTG TCC AAT CAG CAG TCC TGC AT 3' (forward), 3' ATG CAG GAC TGC TGA TTG GAC AC 5' (reverse). BSSHII and ClaI were then used to digest *SM22a-Taz* and *SM22a-Taz*[H69Q] transgenes and the transgene-containing fragments purified from an agarose gel using gel extraction kit (Qiagen). Transgenic mice were generated by pro-nuclear injection of *SM22α-Taz* and *SM22α-Taz*[H69Q] fragments into C57BL/6 embryos, and positive progeny used to breed individual lines with 2 founders per line. For atherosclerosis studies, mice were crossed with Apoe[−/−] mice on a C57BL/6 background (strain # 002052, Jackson Laboratory) to generate *Sm22a-Taz/Apoe*[−/−] and *Sm22a-Taz*[H69Q]/*Apoe*[−/−] mice.

## Genotyping

Ear notches from transgenic mice were incubated with chelex/proteinase K for 2 hours at 55[0]C and then heat inactivated at 95[0]C for 15 min. *Apoe*[−/−] genotyping was performed by PCR using company protocols (Jackson Laboratory). *TAFAZZIN* transgene expression was determined using primers that amplify a 131 bp region of *TAFAZZIN* fused to the C-terminal DDK Tag. The primers sequences used were: FWD: TCAAGAGGAATTCCAGCATCTGA, REV: TCGTCGTCATCCTTGTAATCCA. The cycling parameters were 1 cycle of 94[0]C, 1 min and 31 cycles of 94[0]C, 30 s; 62[0]C, 30 s; 72[0]C, 1 min with final extension 72[0]C, 3 min.

## Atherosclerosis protocols

All mice used in this study were housed in a sterile facility, with 12 h light/dark cycle, 55 +/−10% humidity, at 19–21 °C. Animals were

provided with diet and water *ad libitum*. Mice were weighed once per week and their health status regularly checked during the study. Animal welfare was prioritised and no animal reached the predefined endpoints (significant weight loss, signs of severe distress or suffering) for humane euthanasia.

The Myh11-CreERt2[73] (strain # 019079, Jackson Laboratory), Rosa26-Confetti[74] (strain # 013731, Jackson Laboratory) have been described. Myh11-CreERt2/Rosa26-Confetti/Apoe[−/−] mice[18] used in this study were on a C57BL/6 background and generated through in house breeding. Only male mice were used as the *Myh11-CreERt2* transgene is Y-linked. Intraperitoneal injections of tamoxifen (Sigma) were used to induce recombination in 6-8 week old animals, which were rested for 1 week to flush out tamoxifen prior to fat feeding for 14–23 weeks (21% fat, Special Diet Services # 829100).

In the high fat feeding study, male and female littermate *Apoe*[−/−] mice and C57BL/6 ApoE wild type mice (*Apoe*[+/+], strain # 027, Charles River Laboratories) were randomly assigned to experimental groups. The mice were fed a chow diet (5.1% fat, SAFE 105, SAFE-Lab) until 6 weeks of age and then either sacrificed or fed high fat diet (21% fat, Special Diet Services # 829100) for 14 weeks before sacrifice.

In the tafazzin atherosclerosis study, both male and female littermate *Apoe*[−/−], *Sm22a-Taz/Apoe*[−/−] and *Sm22a-Taz*[H69Q]/*Apoe*[−/−] mice were fat-fed from 6–20 weeks of age (21% fat, Special Diet Services # 829100). To achieve randomisation, *SM22a-Taz/Apoe*[−/−] or *SM22a-Taz*[H69Q]/*Apoe*[−/−] were housed in the same cages as their littermate *Apoe*[−/−] controls during both breeding and during atherosclerosis study. Experimental groups always consisted of both *SM22a-Taz/Apoe*[−/−] or *SM22a-Taz*[H69Q]/*Apoe*[−/−] mice with their littermate controls, and these were sacrificed on the same day. The atherosclerosis protocol was followed by all experimental groups in the same time frame. The animals were included in the study if they successfully underwent the high fat feeding protocol. No animals were excluded due to death or illness. 7 samples were excluded from histological analysis (3 control, 2 *SM22a-Taz/Apoe*[−/−], 2 *SM22a-TazH/Apoe*[−/−] mice) due to aortic section damage that occurred during tissue processing.

For all studies, mice were culled ($CO_2$ asphyxiation for Myh11-CreERt2/Rosa26-Confetti/Apoe[−/−] mice or cervical dislocation for Apoe[−/−], Apoe[+/+], Sm22a-Taz/Apoe[−/−] and Sm22a-Taz[H69Q]/Apoe[−/−] mice), arteries perfused with PBS and tissues removed once the feeding protocol was complete.

For *Apoe*[−/−], *Apoe*[+/+], *Sm22a-Taz/Apoe*[−/−] and *Sm22a-Taz*[H69Q]/*Apoe*[−/−] mice male and female data were used for the analyses. Post hoc sex based statistical analysis has not been performed due to the low sample number. The data disaggregated for sex is included in the source data.

## Blood pressure, full blood count and lipid analysis

After prior familiarisation, blood pressure was measured at 19 weeks of age, using Visitech BP-2000 Blood Pressure Analysis System (Visitech Systems Inc). Peripheral blood sampling was performed at 20 weeks of age. Full blood count was analysed using the ABC vet blood counter (ABX vet 16p, ABX Diagnostics Inc). Lipid profiles were assayed on serum, using commercial enzymatic assays as per the manufacturer instructions (Department of Clinical Biochemistry, Addenbrooke's Hospital).

## SS31 treatment

VSMCs were treated with 1 nM SS-31(Stratech) or scrambled peptide control [2,6-dimethyl-Tyr]-[D-Arg]-Lys-Phe-amide (Calibre Scientific) for 24 h.

## Statistical analysis

Statistical analysis was performed using GraphPad Prism (Version 10.2.3, GraphPad Software). Data were tested for a normal distribution by examination of the histogram and the Shapiro-Wilk test. Data with a normal distribution were analysed by two-tailed Student's *t* test for

pair-wise comparison or ANOVA with Bonferroni-Holm post-hoc analysis for multiple comparisons. Data not following a normal distribution were analysed by a two-tailed Mann Whitney U test or Kruskal-Wallis test. Following the Kruskal-Wallis test, Mann-Whitney tests adjusted for multiple comparisons with the Bonferroni-Holm procedure were performed. Values are presented as means ± SEM. $p < 0.05$ was considered significant.

## Reporting summary

Further information on research design is available in the Nature Portfolio Reporting Summary linked to this article.

## Data availability

All data supporting the findings of this study are available within the main text, Supplementary Information, and Source data. The mass spectrometry lipidomics data generated in this study have been deposited in the MassIVE database under accession code MSV000099336 [https://doi.org/10.25345/C5RB6WF71]. The ScRNA-seq datasets from coronary[15] and carotid lesions[16,17] used in this study have been previously published and are available in the NCBI Gene Expression Omnibus database under accession numbers GSE131778, GSE155512, GSE159677. The datasets GSE 131778, GSE155512 and GSE159677 were re-analysed from integrated scRNA-seq data available at https://zenodo.org/records/14007461[14]. Source data are provided in this paper.

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

## Acknowledgements

This study was supported by British Heart Foundation Grants FS/19/2/34017 (E.Y.), RG71070 (M.B.), RG84554 (M.B.), FS/SBSRF/24/31040 (H.F.J.), the National Institute of Health Research Cambridge Biomedical Research Centre, and the BHF Centre for Research Excellence (RE/18/1/34212, RE/24/130011). B.J. & A.K. are supported by the NIHR Cambridge Biomedical Research Centre (NIHR203312). Medical Research Council UK (MC_UU_00028/4) and a Wellcome Trust Investigator award (220257/Z/20/Z) support work in the MPM lab. This work was funded by the Medical research Council UK (MC_UU_00028/5) for J.P. We would like to acknowledge staff at the Institute of Metabolic Science Metabolic Research Laboratories Histopathology Core Facility for assistance with confocal microscopy. This work was supported by the Wellcome Trust [grant number 226800/Z/22/Z]; and the Medical Research Council [grant number MC_UU_00039]. We would also like to acknowledge staff at Histopathology, Cancer Reserch UK Cambridge Institute for assistance with in situ hybridisation staining. The authors gratefully acknowledge the Advanced Imaging Facility, University of Cambridge

for their support and assistance in this work. This work was supported by the MRC MDU Mouse Biochemistry Laboratory [MC_UU_00014/5].

## Author contributions

E.Y. conceptualised the study. C.D., A.F., N.F., A.T., B.J., S.R.C. and E.Y. conducted experiments during the study and interpreted the results. J.L., S.O. and H.F.J. analysed data. B.J., S.O., H.F.J., S.R.C. and E.Y. wrote the manuscript. A.K., J.P., M.P.M. and M.B. reviewed and revised the manuscript.

## Competing interests

The authors declare no competing interests.
