## [Peer Review File · Nature Communications]

Defective vascular smooth muscle cell tafazzin impairs mitochondrial function and promotes atherosclerosis in preclinical models

Corresponding Author: Dr Emma Yu

Version 0:

Reviewer comments:

Reviewer #1

(Remarks to the Author)

Comments to the Authors:

The manuscript by Dr. Emma Yu and co-authors aimed to investigate the role of the trans-acylase Tafazzin in atherosclerosis and specifically in vascular smooth muscle cells (SMC). The work is novel because the authors, for the first time, propose the role of Tafazzin in regulating SMC mitochondrial functions and protecting against atherosclerosis. While the studies provide solid data supporting the link between miR125a-5p, Tafazzin, and mitochondrial respiration in cultured SMC, validating these in vitro findings in vivo is not well established, and human data are unconvincing. In general, most of the data presented in the manuscripts are seen as preliminary and need more validation.

Main concerns:

Figure 1: Although it is clear that media has more staining for Tafazzin than plaque, it is unclear how the authors quantified the % SMA-positive cells expressing Tafazzin. It does not look like the blue and brown signals are overlapped.

Immunofluorescence staining, followed by z-stack confocal microscopy, should be performed to demonstrate the localization of Tafazzin, specifically in SMA-positive cells. Moreover, presented data clearly demonstrate that media from healthy and diseased arteries has the same levels of Tafazzin. The same is true for in situ hybridization for miR-125a-5p; it is unclear how this staining was quantified specifically in SMC. In addition, the description in the text said that the authors used 9 human vessels per group (para 66), but Figures 1 B, D, and E show only 5 individual samples. This discrepancy needs to be addressed.

Supplementary Figure 1: How plaque SMCs were isolated and characterized is unclear. The main problem during SMC isolation is the contamination with fibroblasts (not with macrophages). SMA does not allow the distinction between SMC and fibroblasts to be made.

Supplementary Figures 2 and 3: Although it is a great idea to look for the expression of Tafazzin in the publicly available scRNAseq datasets, this reviewer does not see any difference in TAZ expression between the plaque (ACTA2-Myh11-TNFRSF11B+) vs. non-plaque SMC (Myh11+ACTA2+TNFRSF11B-). What is the purpose of comparing the level of Tafazzin in different cell types? Also, it is unclear why neither scRNAseq dataset has any fibroblast clusters.

Supplementary Figure 5: The authors should use specimens from the SMC-lineage tracing atherosclerotic mice to demonstrate TAZ and miR-125a-5p colocalization with medial vs. phenotypically modulated (ACTA2-MYH11-) plaque SMC. If authors do not have access to such samples, they may try to contact one of the many labs that have them.

Figure 2-4 + Supplementary Figs 6-8: It is impossible to evaluate these results due to the very minimalistic details of the experimental design. How were mouse and human SMC cultured (medium, passages, confluency, serum starvation, etc)? What was the concentration of LDL/oxLDL? If the concentration was too high, all observed effects may be the results of massive cell death. Additional rescue experiments are needed to prove the proposed mechanisms. The legend's color schematic did not match the graphs (Fig. 4).

Figure 6 and 7/SM22-Taz and TazH mice. The main concern about these animal experiments is that comparisons should be made between the Sm22a-Taz and SM-22a-TazH groups, not with Apoe^{-/-} mice. Moreover, Apoe^{-/-} cannot be used as a WT group; it should be Apoe^{-/-} mice carrying the SM22 minimal promoter. Moreover, TAZ is an x-chromosome gene. Thus, it is highly unlikely for people assigned female at birth to get Barth's Syndrome. Therefore, the autosome transgenic up-regulation of this gene can lead to very different effects compared to the endogenous levels of TAZ, as well as very different effects in males vs. females. It should at least be discussed.

Again, due to the minimalistic experimental details and the absence of representative images, the data quality cannot be

evaluated. How were the cap area and core area designated? TUNEL staining should be done as IF, simultaneously with the nuclear staining (e.g., DAPI) and SMA and MAC3.

Figure 8: 22a-TazH SMC should be compared with 22a-Taz SMC, not only WT cells.

Discussion: The discussion section repeats the results. It should be rewritten to discuss results in the context of the established literature.

Materials and Methods: This section should be significantly rewritten to provide sufficient information about the methods.

Experimental Design schematics within the figures would help readers understand experimental design.

Additional concerns:

1. The author mentioned that TAZ silencing reduced CL content, which is an essential regulator of mitochondrial dynamics, while the expression of TAZ increased CL. However, TAZ silencing or overexpression showed no change in mitochondrial morphology in VSMCs. The reasons for this and the mechanisms behind it should be addressed in the discussion section. Additionally, the levels of mitochondrial dynamics-related proteins (MFN1, MFN2, Drp1, Fis1, and OPA1) should be measured to elucidate the phenotype.

2. They demonstrated oxidative stress using DCFDA staining, which measures ROS levels at the cellular level but not mitochondrial-specific oxidative stress. Therefore, mitochondrial ROS levels should be assessed for mitochondrial functions, and mitoSOX staining is recommended.

3. To further demonstrate mitochondrial functions, mitochondrial membrane potentials and mitochondrial biogenesis markers (PGC-1 α , SIRT1, and TFAM) should be measured.

4. The antibody information was missing in the method section for assessing mitochondrial morphology.

Reviewer #2

(Remarks to the Author)

In this manuscript, Dong et al investigated the role of tafazzin in atherosclerosis. Tafazzin is a mitochondrial protein that remodels the fatty acid composition of the mitochondria-specific lipid cardiolipin. Mutations in the tafazzin gene cause mitochondrial dysfunction in animals and cell cultures and Barth syndrome in humans. The fact that tafazzin has not been studied in the context of atherosclerosis is given as the sole justification for the study.

First, the authors showed that the expression of tafazzin was reduced in human atherosclerotic plaques and smooth muscle cells, which, as expected, led to a decreased concentration of cardiolipin and to diminished mitochondrial function.

Second, they created transgenic mice that expressed either normal or mutated human tafazzin in smooth muscle cells. The expression of normal, but not mutated human tafazzin, increased respiratory activity but the mechanism by which the expression of human tafazzin, in addition to the endogenous mouse tafazzin, caused such change was not studied.

Third, the transgenic mice were crossed with ApoE deficient mice in order to study the effect of tafazzin on atherosclerosis. The expression of mutated, but not wild-type human tafazzin, increased the atherosclerotic plaque area. SS-31, a drug that is believed to act by binding to cardiolipin, increased oxygen consumption in smooth muscle cells derived from mice expressing mutated human tafazzin.

Critique:

1. This work does not provide any novel insight into the function of tafazzin nor does it prove a "regulatory" role of tafazzin in the pathogenesis of atherosclerosis. The transgenic mouse model is of questionable value because it is difficult to know what the expression of human tafazzin on top of the endogenous mouse tafazzin does. No attempts were made to understand why the expression of human tafazzin increased mitochondrial oxygen consumption, which is surprising given that endogenous tafazzin is sufficient to remodel endogenous cardiolipin. The effect of the exogenous tafazzin on cardiolipin was left to imagination. The difference between normal and mutated exogenous tafazzin remains even more difficult to interpret. The idea that tafazzin "regulates" smooth muscle mitochondrial function and atherosclerosis, as stated in the abstract, is not supported by the data. Changes in the tafazzin level may be associated with atherosclerosis, but this does not constitute a specific mechanistic involvement.

2. Work related to the microRNA miR-125a-5p is also superficial as the authors failed to demonstrate whether this microRNA inhibits specifically tafazzin or also other mitochondrial or non-mitochondrial proteins.

3. Patients with Barth syndrome are not known to exhibit altered sensitivity to atherosclerosis, which further undermines the basic tenet of the manuscript.

4. The effects of reduced tafazzin on cardiolipin are not new. They have been demonstrated in numerous publications for different cell types and organisms. More granular data (i.e. molecular species compositions) are required to justify an investigation of the specific effect of tafazzin on smooth muscle cells.

Reviewer #3

(Remarks to the Author)

Summary of the manuscript:

The manuscript by Dong et al. uses a clinical syndrome (Barth's Syndrome) where tafazzin mutations lead to clear consequences of vascular smooth muscle cell (VSMC) dysfunction such as dilated cardiomyopathy and skeletal myopathy. Given the role for tafazzin in regulating cardiolipin – a phospholipid found on the inner membrane of mitochondria – and the known mitochondrial dysfunction (and VSMC alterations) seen in atherosclerosis, they sought to determine whether tafazzin was dysregulated in atherosclerosis. This work builds upon a 2017 publication in ATVB by Dr Yu. Atherosclerotic plaques (human and murine) had decreased tafazzin (mRNA and protein) expression overall and specifically in plaque VSMCs, which they determined was due to increased miRNA 125a-5p binding to tafazzin mRNA (presumably targeting it for degradation with consequent decreased protein expression). The induction of miRNA 125a-5p was increased by NFκB (an upstream regulator of miRNA 125a-5p) driven by oxidized LDL. Mechanistically, the authors were able to recapitulate their human plaque observations whereby tafazzin silencing or mutant overexpression decreased VSMC cardiolipin, mitochondrial respiration, inducing apoptosis and promoting atherosclerosis in murine models. The small peptide SS-31 partially rescued VSMC mitochondrial function and prevented apoptosis. They propose that tafazzin is 1) a newly identified regulator of VSMC mitochondrial function, 2) regulator of atherosclerosis and 3), suggest it may be a novel therapeutic target for atherosclerotic disease.

Significance: This study advances an important body of knowledge on the role of VSMCs in human atherosclerotic plaque. It not only identifies a novel regulator of VSMC dysfunction with meticulous mechanistic insight and acknowledgement of the limitations, but also the pathological implications with models that demonstrate loss of tafazzin in VSMCs leads to more vulnerable plaques as evidenced by increased necrotic core and (thinner) fibrous cap*. *see comment below.

COMMENTS:

1. Comparison was normal aortic tissue (AVR or root Sx) – different region than typical athero formation in humans. There are substantial developmental/structural differences between the ascending aorta and the distal sections including the carotid (e.g. elastin is greater the more proximal the segment). Given that VSMCs are exquisitely sensitive to the tensile forces and matrix upon which they reside, this makes finding a control tissue difficult...Options include deceased donor tissue from comparable regions or adjacent plaque regions. The authors notably used media and intimal staining to localize their finding within a biological context.
 - a. Medial VSMCs had 94% tafazzin expression (along with aortic controls) vs intimal VSMCs only 23% for tafazzin expression – very good. It is an important aspect of this paper and supports the biological process of athero, which is taking place in the intimal layer. This should be emphasized in the discussion regarding control tissues.
 - b. Establishing tafazzin expression in the plaque (and specifically the region), and corroborating this data with plaque-derived VSMCs is a strength of this study. Further identifying tafazzin transcripts from human carotid and coronary scRNAseq datasets is a further asset and validating feature, showing this is repeatable and occurs in plaques from different vascular beds. Alsaigh et al have a scRNAseq dataset that includes the adjacent plaque regions – it may provide insight over what a nearby 'non-diseased' VSMC tafazzin transcripts would be. It is publicly available and would strengthen their argument about the focal implications of tafazzin loss.
2. oxLDL, but not native unoxidized LDL, increased miRNA 125a-5p expression and decreased tafazzin mRNA and protein in hVSMCs, which could be blocked by a miRNA 125a-5p inhibitor.
 - a. Is there a dose-dependent effect of oxLDL in initiating this process?
 - b. Does early athero have reduced mitochondrial function? Or is this an early feature of plaque development? 12wk HFD-fed mice and human plaques provide established lesions, but the advantage of the murine models is to better understand what events are happening along the timelines of plaque development – this knowledge will shape therapeutic strategies and timing for any considered clinical trial (where we do need translational development).
 - c. Do the authors have any data from early athero lesions in their mice?
3. OxLDL and NFκB are well established. NFκB has robust effects in atherosclerosis and naturally addresses the inflammatory component of atherosclerosis – why not consider interfering with this well-known pathway in VSMCs? This should be further contextualized in the discussion, particularly since there could be synergistic benefits on endothelial and macrophage dysfunction.
4. silencing experiments to examine the functional consequences of loss of tafazzin (done with siRNA and lentivirus gene transfer to express WT tafazzin OR a mutant with decreased efficiency) is strong.
 - a. How much does the latter decrease tafazzin?
 - b. Is it meaningful functionally – the authors suggest it is based upon the improved mitochondrial readouts done in isolated settings. What this means in a plaque milieu might be difficult to ascertain specifically. Can they comment?
 - c. Could increasing tafazzin be a therapeutic strategy? If so, how much is needed to preserve mitochondrial function? Any dose-response understanding of this process?
5. Exciting that the Sm22a-Taz/ApoE^{-/-} mice had no change in plaque area after 14wks HFD, but increased fibrous cap and reduced necrotic core – stabilizing plaques (not only preventing or regressing them) is a valuable clinical target. Seeing that mutant tafazzin increased plaque vulnerability with the inverse observations is quite relevant. However, the data are hard to appreciate -particularly the fibrous cap thickness.
 - a. IN FIGURE 6 – can the authors show regions of fibrous caps that are being evaluated?
 - b. Can a supplemental be provided that shows more histological examples please?
6. Does SS-31 stabilize the acetylated form of cardiolipin? Assume this is the goal as it bypasses the tafazzin role required. Please specify in the manuscript.
7. Do the authors expect that tafazzin would be low in macrophages and endothelial cells? These are also highly metabolically active cells that require tight control of their mitochondrial function. As a potential therapeutic target, it will be important to understand how any antagonist or small molecule therapeutic will affect these crucial cells that contribute to

plaque progression.

a. Do the authors have any data of their own on the small molecule inhibitor effects on ECs or macrophages?

8. Looking for upstream regulators – particularly a microRNA – is another strength of this manuscript. miRNAs are microregulators of many disease processes. For a function that requires such tight control, this is an excellent upstream regulator and amenable to therapeutic designs.

a. Did the authors try an antagomir strategy in their transgenic mouse model?

b. Similar to # 9: How would targets get to inner leaflet? Any drugs already likely to work on this pathway? The authors mention that the mechanism of action is not known for the small molecule SS-31. This will be an issue advancing forward in human studies. Are there antagomir strategies?

c. miRNA often work collectively – were there other miRNA candidates that should be considered? There are groups actively working on multiple miRNA (multiplex) strategies for clinical approaches.

9. Cardiolipin is clearly a highly regulated process – in nature this usually signifies importance of tight regulation.

a. Are there other transacylases to consider?

b. Are there GWAS data with SNPs in cardiolipin or tafazzin that might help demonstrate a role in vasculopathy – beyond the Barth's syndrome with premature death as they noted?

10. Mitochondria have been released in EVs and picked up in blood.(Nat Comm 2023). It is known that EVs can be found in plaques (Blaser et al Circulation 2023). Might it be possible that they would serve as long-range dysfunctional signals to VSMCs? OR long-range therapeutics? While not part of the scope of this project, it should be acknowledged.

11. Is there a way to harness delivery to the endothelium for release to local plaque environments (examples would include VCAM-1 targeted nanopeptides for regional delivery, PNAS 2021)?

12. hVSMCs isolated from tissue were divided into small pieces and cultured in 6 well plates for 1-2wks to allow cells to emerge.

a. Is there any data on how the cells change over time once ex vivo? Given the meticulous controls, it is unlikely an issue.

b. With the VSMC tafazzin and miRNA-stimulation experiments, was there any significant cell death? The MOI was high and presumably these cells are somewhat less hardy after several weeks in a 6 well plate.

MINOR

1. How were the THP-1 macrophages generated? Assume PMA for 24-48%, but the dose and duration should be in the methods section.

2. There are several places in the methods where the authors have indicated "according to the manufacturers instructions". A brief statement should be included if possible.

Version 1:

Reviewer comments:

Reviewer #1

(Remarks to the Author)

Comments to the Authors:

The authors performed many of the requested experiments, including in vitro rescue experiments (Fig. 2F-H and Fig. 3B-D), additional in vitro experiments characterizing mitochondrial morphology and gene/protein expression profiles, and mitochondrial ROS measurement, showed additional comparison between experimental groups, included discussion about the endogenous X-chromosome location of TAZ gene vs. transgenic expression of human TAZ gene in the SM22a-Taz/TazH mice, and added more details about methods.

However, several critical concerns were left unanswered.

1 – Although the authors replaced IH staining for SMA/TAZ with IF and confocal microscopy for human aortic and carotid plaques, these data only show that medial SMA+ cells (presumably SMC) express TAZ in both aorta and carotid artery with plaque and SMA+ cells in plaque do not express TAZ. Given that SMA is not a specific SMC marker in the lesion, the author can only claim that they see a significant decrease in SMA+TAZ+ cells within the lesion compared to the medial cells. The same is true about miR-125a-5p staining (here they can only discuss media vs. plaque cells). Therefore, all conclusions about TAZ signaling in plaque SMCs are overstated.

As suggested, these data could be supplemented by IF staining on mouse atherosclerotic tissues from SMC-lineage tracing mice. The authors wrote in their rebuttal letter that they had technical issues combining SMC-lineage tracing staining with the miR125a-5p hybridization protocol. Still, it is unclear why they did not perform SMC lineage tracing with TAZ staining to show the SMC-specific TAZ expression within the plaque.

Although the technical difficulties are understandable, the revised version of the manuscript still has the same issues about the origin of the SMA+TAZ+ cells in the plaque as the original one.

2 – The rationale of having publicly available scRNAseq datasets in the manuscript and which important conclusions have been made from these analyses (except that "quantification of lowly expressed genes is problematic") is unclear and confusing.

The authors' relabeling SMCs and fibroblasts to "structural" cells and the rationale for doing this (from the rebuttal letter) is also confusing. Many SMC labs, including Thomas Quertermous' (the scRNAseq dataset that the authors use in this manuscript), Gary Owens', and Muredach Reilly's labs (and many others), performed scRNAseq on SMC lineage tracing atherosclerotic mice and human atherosclerotic arteries and had very well characterized marker panels for phenotypically modulated SMC and fibroblasts.

The difference between TAZ expression levels in SMC vs macrophages and endothelial cells is not supported by any statistical analysis.

As presented, all these scRNAseq data do not support the authors' hypothesis about the role of TAZ in SMC in atherosclerosis and should be removed from the manuscript.

3 – The authors did not answer how plaque SMCs were isolated and characterized and how they were distinguished from fibroblasts. Instead, the authors reference Supplementary Fig. 1, which shows staining for SMA, MYH11, and CD68 without explaining which hVSMC they show (plaque vs. aortic) and how these cells were cultured (e.g., in serum vs. in serum-free media). Should the plaque phenotypically modulated SMC have lower (or minimal if completely de-differentiated) levels of the SMC contractile markers?

4 – IF staining in Figure 7. There are concerns about this staining. SMA looks overexposed, MAC3 is invisible, and TUNEL staining in the TAZH group looks unspecific (sticky necrotic core unspecificity?). A better representation that includes negative controls would be helpful.

5 – The authors partially revised the Discussion section, but it is still very long and repeating results.

6 – Given the X-chromosome location of the TAZ gene, the authors should indicate the sex of mice/cells/human specimens they used for analyses.

Reviewer #2

(Remarks to the Author)

I appreciate the detailed response by the authors. Still, in my opinion, the work merely describes observations that have unproven biological significance and that provide little mechanistic insight. If indeed, TAZ affects the expression of complex I in any specific manner, one would have to reveal the underlying mechanism of transcriptional regulation.

Reviewer #3

(Remarks to the Author)

The Authors have satisfied my questions and comments as requested.

Version 2:

Reviewer comments:

Reviewer #1

(Remarks to the Author)

The authors addressed all my comments and concerns.

Reviewer #2

(Remarks to the Author)

I remain unconvinced of the specific connection between tafazzin and atherosclerosis. It is not clear whether microRNA 125a-5p suppresses tafazzin specifically or mitochondrial proteins in general. If the former, I would agree with the authors. If the latter, the work would show that mitochondria are involved in the pathogenesis of atherosclerosis.

Response to referees

We thank the referees for their comments and our point-by-point responses are as below.

REVIEWER COMMENTS

Reviewer #1 (Remarks to the Author):

Comments to the Authors:

The manuscript by Dr. Emma Yu and co-authors aimed to investigate the role of the trans-acylase Tafazzin in atherosclerosis and specifically in vascular smooth muscle cells (SMC). The work is novel because the authors, for the first time, propose the role of Tafazzin in regulating SMC mitochondrial functions and protecting against atherosclerosis. While the studies provide solid data supporting the link between miR125a-5p, Tafazzin, and mitochondrial respiration in cultured SMC, validating these in vitro findings in vivo is not well established, and human data are unconvincing. In general, most of the data presented in the manuscripts are seen as preliminary and need more validation.

We thank the reviewer for their comments and are pleased that the “work is novel” and that the “studies provide solid data supporting the link between miR125a-5p, tafazzin and mitochondrial respiration in cultured SMC.” The reviewer’s concerns are addressed as below.

Main concerns:

Figure 1: Although it is clear that media has more staining for Tafazzin than plaque, it is unclear how the authors quantified the % SMA-positive cells expressing Tafazzin. It does not look like the blue and brown signals are overlapped. Immunofluorescence staining, followed by z-stack confocal microscopy, should be performed to demonstrate the localization of Tafazzin, specifically in SMA-positive cells. Moreover, presented data clearly demonstrate that media from healthy and diseased arteries has the same levels of Tafazzin. The same is true for in situ hybridization for miR-125a-5p; it is unclear how this staining was quantified specifically in SMC.

We thank the reviewer for their comment. As suggested, we have now performed immunofluorescence staining with confocal microscopy to demonstrate the localisation of tafazzin in SMA-positive cells (**Figure 1C**). We found that whilst $90.5\pm 1.6\%$ of VSMCs expressed tafazzin in normal aortic tissue, with similar expression in medial VSMCs of plaques, only $7.0\pm 4.3\%$ intimal VSMCs expressed tafazzin in plaques (**Figure 1D**). This data is now described in the revised manuscript (para 1, page 4).

‘Immunofluorescence using α -smooth muscle cell actin as a VSMC marker demonstrated that $90.5\pm 1.6\%$ of VSMCs expressed tafazzin in normal aortic tissue,

with similar expression in medial VSMCs of plaques. However, only $7.0 \pm 4.3\%$ intimal VSMCs expressed tafazzin in plaques (**Figures 1C-D**)’.

For miR-125a-5p we have shown that % of cells expressing miR-125a-5p are the same in the media from healthy and diseased arteries but increased in the plaque intima (**Figure 1D and E**). We attempted to combine miRNAscope for miR-125a-5p with RNAscope for SMA/ACTA2 but unfortunately this proved extremely technically challenging. We were therefore unable to accurately quantify miR-125a-5p in VSMCs specifically in the same section and have now acknowledged this limitation in the discussion (para 2, page 21).

However, the immunofluorescence staining for tafazzin/SMA with confocal microscopy was performed in the section immediately adjacent to the miR-125a-5p probed section. We have therefore assessed tafazzin expression in SMA-positive cells and miR-125a-5p expression in the same region.

In addition, the description in the text said that the authors used 9 human vessels per group (para 66), but Figures 1 B, D, and E show only 5 individual samples. This discrepancy needs to be addressed.

We first performed qPCR analysis of tafazzin mRNA expression in healthy and diseased arteries (**n=9, Figure 1A**). We then performed western blotting to examine whether there was a change of tafazzin abundance at protein level. Power calculations estimated a group size of 5 to detect a 2-fold difference in protein levels. We saw clear differences in tafazzin protein expression with **n=5 (Figure 1B)** and therefore decided to conserve our limited human samples rather than increase sample numbers. The results section has been re-written to describe the RNA and protein findings separately (para 1, page 4).

Supplementary Figure 1: How plaque SMCs were isolated and characterized is unclear. The main problem during SMC isolation is the contamination with fibroblasts (not with macrophages). SMA does not allow the distinction between SMC and fibroblasts to be made.

We agree, but *Myh11* is considered to be the most specific VSMC marker¹, and immunofluorescence staining of cultured plaque VSMCs shows high levels of MYH11 expression consistent with a VSMC cell type (**Supplemental Figure 1A**),

Supplementary Figures 2 and 3: Although it is a great idea to look for the expression of Tafazzin in the publicly available scRNAseq datasets, this reviewer does not see any difference in TAZ expression between the plaque (ACTA2-Myh11-TNFRSF11B+) vs. non-plaque SMC (Myh11+ACTA2+TNFRSF11B-).

We agree that supplementary Figures 2 and 3 show that tafazzin expression is low across different VSMC subtypes and there is no marked difference between clusters. However, the inherent noise and "dropout" events in scRNA-seq data means that

detection and hence quantification of lowly expressed genes is problematic, including in modulated (ACTA2-MYH11-TNFRSF11B+) vs. contractile SMC (ACTA2+ MYH11+TNFRSF11B-). We therefore also examined Tafazzin expression in a preliminary analysis of our unpublished single nuclei (sn) RNA-Seq data from diseased human carotid tissue (Figure below). Similar to the scRNA-seq analysis, Tafazzin expression is low in this dataset, and while this analysis is not conclusive, TAZ expression appears lower in VSMCs expressing fibromyocyte markers such as osteoprotegerin (TNFRSF11B, "Fibro-VSMC") than in contractile VSMCs with high expression of smooth muscle cell myosin heavy chain (MYH11, "VSMC1"). While this data supports our hypothesis that 'fibromyocyte' SMCs express less TAZ, we would prefer not to include it in the revised paper unless specifically requested.

Figure for reviewer

Analysis of single nuclear RNA-seq profiles of carotid plaque. Dot plot showing scaled expression levels for tafazzin (TAZ) and markers of structural cell subsets (MYH11, ACTA2, TNFRSF11B), macrophages (CD68) B cells (CD79B, PTPRC), CCND1, T cells (CD3E), endothelial cells (CDH5). Expression level is shown using a grey-blue color scale. Dot size show the percentage of expressing cells for each cluster.

What is the purpose of comparing the level of Tafazzin in different cell types? Also, it is unclear why neither scRNAseq dataset has any fibroblast clusters.

Since tafazzin expression is higher in macrophages and endothelial cells than in VSMCs this indicates that the expression is low in VSMCs, rather than reflecting a technical limitation in the detection of tafazzin mRNA in the samples.

Cell annotation for plaque scRNA-seq datasets is complex and highly debated due to the plasticity of implicated cell types in disease. The scRNAseq cell populations have therefore been relabelled as “structural” rather than “mural” or “VSMC”

(**Supplemental Figures 2,3**), a term that encompass both VSMCs and fibroblasts.

Supplementary Figure 5: The authors should use specimens from the SMC-lineage tracing atherosclerotic mice to demonstrate TAZ and miR-125a-5p colocalization with medial vs. phenotypically modulated (ACTA2-MYH11-) plaque SMC. If authors do not have access to such samples, they may try to contact one of the many labs that have them.

We thank the reviewer for the elegant suggestion and have attempted to examine tafazzin and miR-125a-5p expression in medial vs phenotypically modulated plaque VSMCs in SMC-lineage tracing atherosclerotic mice (Myh11-CreERT2/Rosa26-Confetti/ApoE^{-/-2}). Unfortunately, the fluorescence signal from the VSMC-lineage tracing confetti reporter is lost after processing for miRNAscope. Whilst it was therefore not possible to demonstrate tafazzin/miR-125a-5p colocalization specifically in VSMCs, we have found that tafazzin expression is decreased in miR-125a-5p⁺ vs. miR-125a-5p⁻ arterial cells as shown below:

Figure for the reviewer

RNA in situ hybridization for microRNA 125a-5p (miR-125a-5p)(pink), tafazzin (white) with nuclei stained with DAPI (blue) in an aortic section from Myh11-CreERT2/Rosa26-Confetti/ApoE^{-/-} mouse fat fed for 22 weeks. Scale bar = 10 μm. Quantification of tafazzin expression in miR-125a-5p⁻ or miR-125a-5p⁺ cells (n=3 samples per group, unpaired t test). Data are shown as mean ± SEM.

Figure 2-4 + Supplementary Figs 6-8: It is impossible to evaluate these results due to the very minimalistic details of the experimental design. How were

mouse and human SMC cultured (medium, passages, confluency, serum starvation, etc)? What was the concentration of LDL/oxLDL? If the concentration was too high, all observed effects may be the results of massive cell death.

We thank the reviewer for their comment and the methods have now been revised to include extensive further details regarding mouse and human VSMC culture media, passage, and confluency. Mouse VSMCs were initially cultured in DMEM/F12 for 3 weeks before switching to DMEM, split 1:2 at 90% confluency and studied at passages 2-12. Human VSMCs were cultured in VSMC media (Promocell), passaged at 90% confluent (reseeding at 13,333 cells/cm²) and studied at passages 2-8. The concentration of LDL and oxLDL was 100 µg/ml and the methods have been revised to include these details (paras 2,3 page 27; para 2, page 28). We acknowledge that high doses (eg 300 µg/ml of oxLDL) can cause significant cell death³, but we found only 15±2.2 % annexin V-positive cells after oxLDL treatment (**Supplemental Figure 8E**) suggesting that massive cell death was not present. However, given potential concerns regarding cell viability, VSMCs were not exposed to the additional stress of serum starvation prior to oxLDL treatment.

Additional rescue experiments are needed to prove the proposed mechanisms.

In response to the reviewer's comments we have now added new rescue experiments and describe the rescue experiments that underpin our hypothesis. We proposed that oxidised LDL increases miR-125a-5p which decreases tafazzin, an effect that was rescued with miR-125a-5p inhibitor (**Figures 2F-H**). We next suggest that oxLDL results in NFκB activation, increased miR-125a-5p and decreased tafazzin. Treatment with the NFκB inhibitor Bay 11-7082 rescued the oxLDL effect on miR-125a-5p and tafazzin (**Figures 3B-D**). We believe that these rescue experiments support the proposed mechanism and have now also performed siRNA-mediated knockdown of NFκB to further strengthen the data. We show that with siRNA-mediated knockdown of NFκB, the oxLDL-induced increases in NFκB binding, NFκB phosphorylation and miR-125a-5p expression were blocked, and the oxLDL effect on tafazzin abundance was rescued (**Supplemental Figure 7**).

The legend's color schematic did not match the graphs (Fig. 4).

The legend's colour scheme has been updated to more closely match the graphs.

Figure 6 and 7/SM22-Taz and TazH mice. The main concern about these animal experiments is that comparisons should be made between the Sm22a-Taz and SM-22a-TazH groups, not with Apoe—/—mice. Moreover, Apoe—/— cannot be used as a WT group; it should be Apoe-/- mice carrying the SM22 minimal promoter.

In response to the reviewer's comments, we have also now shown comparisons between Sm22a-Taz/ApoE^{-/-} and SM22a-TazH/ApoE^{-/-} groups in Figures 6 and 7; we confirm differences between these groups for multiple parameters where a difference between ApoE^{-/-} and Sm22a-TazH/ApoE^{-/-} was observed, namely max plaque area, and percentage cap, core, SMA, Mac3, tunel and annexin V positive cells. This would be consistent with the effects being driven by Taz or TazH expression, and not due to the Sm22a promoter that is in both groups. We thank the reviewer for the

suggestion of ApoE^{-/-} mice carrying the Sm22a minimal promoter as an alternative control group. However, we have previously found that expressing the minimal SM22a promoter coupled to an activatable gene has no effect without activating the transgene⁴, indicating that expressing the promoter without a gene of interest would not be anticipated to affect atherosclerosis. In addition, we have used 2 founders for each transgenic line to mitigate against non-specific effects in transgenic mice.

Moreover, TAZ is an x-chromosome gene. Thus, it is highly unlikely for people assigned female at birth to get Barth's Syndrome. Therefore, the autosome transgenic up-regulation of this gene can lead to very different effects compared to the endogenous levels of TAZ, as well as very different effects in males vs. females. It should at least be discussed.

We agree and have now included in the discussion that autosomal transgenic up-regulation of tafazzin may have different effects to endogenous tafazzin, which is an X chromosome gene (para 2, page 18). The differences are difficult to predict; whilst tafazzin expression would be subject to X inactivation, dosage compensation via upregulation of X chromosome gene expression can also occur⁵. Similarly, autosomal transgenic up-regulation of tafazzin could potentially have different effects in males and females. Our observations were similar in male and female mice apart from for weight; male control, Sm22a-Taz and SM22a-TazH all had increased weight compared with female mice of the same genotype (**Supplemental Figure 15B**, data disaggregated for sex are shown in source data).

Again, due to the minimalistic experimental details and the absence of representative images, the data quality cannot be evaluated. How were the cap area and core area designated?

The cap was defined as the area rich in VSMCs (red) overlying the cholesterol rich, acellular core, as now stated in the main text (para 3, page 12) and methods (para 1, page 25), and shown in Supplemental Figure 16. Supplemental Figure 16 also shows further representative images of the Masson's staining.

TUNEL staining should be done as IF, simultaneously with the nuclear staining (e.g., DAPI) and SMA and MAC3.

We thank the reviewer for the comment. We have now performed TUNEL staining as immunofluorescence and co-stained for DAPI, SMA and Mac3 (**Figure 7A**). We found 0.82±0.21% TUNEL-positive VSMCs in *Sm22a-Taz/ApoE^{-/-}* mice and 3.3±0.53% TUNEL-positive VSMCs in *Sm22a-Taz^{H69Q}/ApoE^{-/-}* mice (**Figure 7A-B**).

Figure 8: 22a-TazH SMC should be compared with 22a-Taz SMC, not only WT cells.

In response to the reviewer's comments, we have treated VSMCs from wild type, SM22a-Taz and SM22a-TazH mice with scrambled peptide control and SS31, and assessed mitochondrial respiration, ROS and apoptosis (**Figure 8A-D**). We have now compared SM22a-TazH VSMCs with SM22a-Taz VSMCs and wild type cells as shown in Figures 8B-D. SS-31 modestly, but significantly, increased maximum OCR in Taz^{H69Q} VSMCs (**Figures 8A-B**). SS-31 treatment also reduced mitochondrial ROS

and apoptosis of *Sm22a-Taz^{H69Q}* VSMCs after tert-butyl hydroperoxide (**Figures 8C-D**).

Discussion: The discussion section repeats the results. It should be rewritten to discuss results in the context of the established literature.

The discussion has been revised to discuss the results in the context of the established literature more fully.

Materials and Methods: This section should be significantly rewritten to provide sufficient information about the methods. Experimental Design schematics within the figures would help readers understand experimental design.

We thank the reviewer for the comment and have significantly rewritten the materials/methods to give sufficient information. We have also included experimental design schematics in the Source Data for Figures 2-6.

Additional concerns:

1. The author mentioned that TAZ silencing reduced CL content, which is an essential regulator of mitochondrial dynamics, while the expression of TAZ increased CL. However, TAZ silencing or overexpression showed no change in mitochondrial morphology in VSMCs. The reasons for this and the mechanisms behind it should be addressed in the discussion section. Additionally, the levels of mitochondrial dynamics-related proteins (MFN1, MFN2, Drp1, Fis1, and OPA1) should be measured to elucidate the phenotype.

We recognise that cardiolipin has an important role in regulating mitochondrial dynamics through its interactions with Opa1 to promote mitochondrial inner membrane fusion, and Drp1 to promote fission⁶. However, changes in tafazzin do not always result in a change in mitochondrial morphology. Indeed, cardiac mitochondria from tafazzin-deficient mice show no change in morphology at 8 weeks old, and even at 8 months the majority of mitochondria showed normal size and structure⁷. Furthermore, if morphological changes occur they can vary, with some groups observing large mitochondria in tafazzin deficiency whilst others reported smaller, fragmented mitochondria^{8, 9}.

Whether changes in mitochondrial morphology are observed may depend on the balance between the fusion and fission proteins, and no consistent pattern in the fusion and fission proteins has been reported in tafazzin deficiency; one group observed increased Fis1 and MFN2, whilst others have noted decreased MFN2^{9, 10}. We have now examined the levels of MFN1, MFN2, Drp1, Fis1 and Opa 1 and found increased Fis1 and Opa1 in tafazzin-silenced VSMCs (**Supplemental Figure 10E**). We suggest that this may promote both fission and fusion with therefore no overall change in morphology observed. In contrast, although tafazzin over-expression increased CL content, no changes in either the abundance of the mitochondrial dynamics proteins or mitochondrial morphology were found (**Supplemental Figure 11**). The relationship between tafazzin, cardiolipin, mitochondrial dynamics proteins and mitochondrial morphology therefore remains to be fully elucidated and we have discussed our mitochondrial morphology findings in the revised discussion (paras 2,3 page 17, para 1 page 18).

2. They demonstrated oxidative stress using DCFDA staining, which measures ROS levels at the cellular level but not mitochondrial-specific oxidative stress. Therefore, mitochondrial ROS levels should be assessed for mitochondrial functions, and mitoSOX staining is recommended.

We thank the reviewer for their comment and have now examined mitochondrial ROS levels using mitoSOX. Similar to DCFDA, we observed increased mitochondrial ROS in mVSMCs overexpressing Taz^{H69Q} after lentiviral transduction and in VSMCs derived from Sm22a-Taz^{H69Q} mice (**Figures 4G, 5I**). In addition, we have now found increased mitochondrial ROS in tafazzin-silenced VSMCs and decreased levels in Sm22a-Taz VSMCs (**Figures 4G, 5I**), consistent with tafazzin having important effects on VSMC mitochondrial function.

3. To further demonstrate mitochondrial functions, mitochondrial membrane potentials and mitochondrial biogenesis markers (PGC-1alpha, SIRT1, and TFAM) should be measured.

We thank the reviewer for their comments and have examined mitochondrial membrane potential ($\Delta\Psi_m$) using live cell videomicrography of tetramethyl rhodamine ethyl ester (TMRE) -stained VSMCs. We found no changes in $\Delta\Psi_m$ in tafazzin silenced VSMCs nor in VSMCs overexpressing wild-type tafazzin or TazH (**Supplemental Figure 12**). This is consistent with both the decreased respiration and the decreased ATP/ADP ratio observed in Taz-silenced and TazH VSMCs. With both decreased proton pumping across the mitochondrial inner membrane as well as decreased proton uptake by the ATP synthase, there would be no overall effect on $\Delta\Psi_m$, which is determined by the balance of respiration and ATP synthesis

We have also examined mitochondrial biogenesis markers PGC1a, sirt1 and TFAM by western blotting. As below, there was no change in these markers in Taz-silenced VSMCs or in VSMCs overexpressing wild-type tafazzin or TazH.

Figure for reviewer

Representative western blots for PGC1a, Sirt1, TFAM in hVSMCs transfected with control siRNA (ctrl) or siRNA against tafazzin (siTaz) (A) or in mVSMCs overexpressing empty vector (EV), tafazzin (Taz) or transacylase mutant tafazzin (Taz^{H69Q} or TazH)(B) with quantification. CS= citrate synthase, Tub = tubulin. Data normalized to citrate synthase for TFAM or tubulin for PGC1a and Sirt1 (n=4 independent experiments; unpaired t test for siTaz vs control comparisons; 1-way ANOVA for EV, Taz and TazH VSMC comparisons). Data are shown as mean ± SEM.

4. The antibody information was missing in the method section for assessing mitochondrial morphology.

The antibodies used for assessing mitochondrial morphology were TOMM20 (311802-1-AP, Proteintech), TOMM70 (#14528-1-AP, Proteintech) and Alexa Fluor™ 488 anti-rabbit secondary antibody (#A-11008, Invitrogen). This information is now in the methods section, para 1 page 32.

Reviewer #2 (Remarks to the Author):

In this manuscript, Dong et al investigated the role of tafazzin in atherosclerosis. Tafazzin is a mitochondrial protein that remodels the fatty acid composition of the mitochondria-specific lipid cardiolipin. Mutations in the tafazzin gene cause mitochondrial dysfunction in animals and cell cultures and Barth syndrome in humans. The fact that tafazzin has not been studied in the context of atherosclerosis is given as the sole justification for the study.

First, the authors showed that the expression of tafazzin was reduced in human atherosclerotic plaques and smooth muscle cells, which, as expected, led to a decreased concentration of cardiolipin and to diminished mitochondrial function.

Second, they created transgenic mice that expressed either normal or mutated human tafazzin in smooth muscle cells. The expression of normal, but not mutated human tafazzin, increased respiratory activity but the mechanism by which the expression of human tafazzin, in addition to the endogenous mouse tafazzin, caused such change was not studied.

Third, the transgenic mice were crossed with ApoE deficient mice in order to study the effect of tafazzin on atherosclerosis. The expression of mutated, but not wild-type human tafazzin, increased the atherosclerotic plaque area. SS-31, a drug that is believed to act by binding to cardiolipin, increased oxygen consumption in smooth muscle cells derived from mice expressing mutated human tafazzin.

Critique:

1. This work does not provide any novel insight into the function of tafazzin nor does it prove a “regulatory” role of tafazzin in the pathogenesis of atherosclerosis. The transgenic mouse model is of questionable value because it is difficult to know what the expression of human tafazzin on top of the endogenous mouse tafazzin does.

We thank the reviewer for their comments. We entirely agree that tafazzin is well-recognized as a mitochondrial transacylase that acylates immature monolysocardiolipin to mature cardiolipin. However, we believe our work is both novel and of clinical significance, identifying VSMC tafazzin as a potential new pathological mechanism and therapeutic target in atherosclerotic disease. Specifically, we show for the first time that VSMC tafazzin is decreased in atherosclerosis, is important for VSMC function and atherosclerosis development, and identify a potential mechanism for its downregulation in atherosclerosis.

We respectfully disagree that the Sm22a-Taz model is of questionable value in atherosclerosis research. We show that endogenous Tafazzin is reduced in human and mouse atherosclerosis, and have used the Sm22a-Taz model to act as a rescue of the decreased endogenous tafazzin expression. Indeed, there were marked effects on atherosclerosis in Sm22a-Taz/ApoE^{-/-} mice. Importantly, tafazzin expression in Sm22a-Taz mice was increased only two-fold compared to control mice, while any effect of human tafazzin on top of the endogenous mouse tafazzin is controlled by use of TazH mice driven by the same promoter at similar levels (**Figure 5D**).

No attempts were made to understand why the expression of human tafazzin increased mitochondrial oxygen consumption, which is surprising given that endogenous tafazzin is sufficient to remodel endogenous cardiolipin. The effect of the exogenous tafazzin on cardiolipin was left to imagination. The difference between normal and mutated exogenous tafazzin remains even more difficult to interpret

We believe that expression of human tafazzin increases mitochondrial oxygen consumption in part through expression of complex I. We find that VSMCs from Sm22a-Taz mice not only show increased mitochondrial oxygen consumption but also increased complex I subunit abundance (**Figure 5D**), which could increase electron transport and therefore OCR. In contrast VSMCs from Sm22a-TazH mice show decreased complex I subunit abundance and decreased OCR (**Figures 5D-G**). This possible mechanism is now explained in the discussion (para 2, page 18).

Regarding the effects of exogenous tafazzin on cardiolipin we found VSMCs overexpressing tafazzin after lentiviral mediated transduction showed increased cardiolipin whilst VSMCs overexpressing mutated tafazzin showed decreased cardiolipin (**Figure 4B**).

The idea that tafazzin “regulates” smooth muscle mitochondrial function and atherosclerosis, as stated in the abstract, is not supported by the data. Changes in the tafazzin level may be associated with atherosclerosis, but this does not constitute a specific mechanistic involvement.

We agree that the changes in tafazzin expression in our human data are only associated with atherosclerosis. However, the findings of altered plaque composition in Sm22a-Taz/ApoE^{-/-} mice and increased plaque burden in Sm22a-TazH/ApoE^{-/-} mice indicate that tafazzin levels have important effects on atherogenesis. We also show that silencing tafazzin or expression of mutated tafazzin in VSMCs decreases mitochondrial respiration and ATP/ADP (**Figures 4C-F**), which is consistent with tafazzin having important mechanistic effects on VSMC mitochondrial function. However, in response to the reviewer’s comments, the abstract has been revised to include this wording and “regulator” removed.

‘Tafazzin therefore has important effects on VSMC mitochondrial function and atherosclerosis, and is a novel potential therapeutic target in atherosclerotic disease’ (Abstract)

2. Work related to the microRNA miR-125a-5p is also superficial as the authors failed to demonstrate whether this microRNA inhibits specifically tafazzin or also other mitochondrial or non-mitochondrial proteins.

The TargetScan miRNA database identified tafazzin as a potential target of miR-125a-5p. We show that miR-125a-5p expression is increased in atherosclerosis, miR-125a-5p binds to the 3’ UTR of tafazzin of VSMCs and transfection with miR-125a-5p mimic reduces VSMC tafazzin expression. We therefore show that miR-125a-5p is present in atherosclerosis and inhibits VSMC tafazzin expression.

However, we recognise that TargetScan identifies other possible targets of miR-125a-5p and that microRNAs often have a range of targets. In response to the query, we have now assessed the impact of this microRNA globally by bulk RNA-seq analysis of hVSMCs transfected with a miR-125a-5p mimic. As might be predicted, we found an overall downregulation of multiple predicted targets of miR-125a-5p as

shown below. This included both mitochondrial components eg TAZ, BAK1 and non-mitochondrial proteins eg KLF13, ETS1.

Figure for the reviewer:

Volcano plot depicting the differential expression of the top 100 predicted targets of miR-125a-5p after miR-125a-5p mimic transfection in human VSMCs. Y axis denotes $-\log_{10}$ p values, X axis shows \log_2 fold change values.

After transfection with miR-125a-5p mimic we found decreased VSMC mitochondrial respiration and proliferation (**Figures 3E-G**). As already noted in the manuscript, given that miR-125a-5p has multiple targets these effects could not simply be attributed to changes in tafazzin expression thus the specific tafazzin-targeted experiments were then performed (**Figure 4**). These data clearly show that the effects of silencing tafazzin are similar to the effects of the miR-125a-5p mimic, suggesting that the major effects of miR-125a-5p may be on tafazzin. Indeed, the TargetsCan database predicts that the predicted miR-125a-5p binding site in tafazzin 3' UTR has a probability of conserved targeting >0.9.

3. Patients with Barth syndrome are not known to exhibit altered sensitivity to atherosclerosis, which further undermines the basic tenet of the manuscript.

We entirely agree that Barth syndrome patients are not known to exhibit altered sensitivity to atherosclerosis and have commented on this in the discussion (para 2, p19). However, as previously discussed, until recently Barth's syndrome had a poor prognosis with a 5-year survival rate of just 22% pre 2000¹¹, whilst atherosclerosis usually does not manifest until the 5th decade. With supportive therapies, life expectancy for Barth's Syndrome patients has improved and vasculopathy may yet prove to be a feature. Indeed, a variant in *PTPMT1*, which is involved in cardiolipin synthesis, is associated with hypertension suggesting that defects in cardiolipin may have a role in vasculopathy^{12,13}.

4. The effects of reduced tafazzin on cardiolipin are not new. They have been demonstrated in numerous publications for different cell types and organisms. More granular data (i.e. molecular species compositions) are required to justify an investigation of the specific effect of tafazzin on smooth muscle cells.

We agree that tafazzin is well-recognised as a trans-acylase that acylates immature monolysocardiolipin to mature cardiolipin in mitochondria. However, ours is a functional study on Tafazzin in atherosclerosis, and we are not trying to replicate studies on molecular species composition in VSMCs compared to other cells. Our findings of decreased VSMC tafazzin expression in atherosclerosis and a potential explanation for this finding, and the effects of reduced tafazzin in VSMCs and on atherosclerosis have not been reported before. These are all important findings to address whether tafazzin could be both a pathogenetic mechanism and/or a therapeutic target in atherosclerosis.

Reviewer #3 (Remarks to the Author):

Summary of the manuscript:

The manuscript by Dong et al. uses a clinical syndrome (Barth's Syndrome) where tafazzin mutations lead to clear consequences of vascular smooth muscle cell (VSMC) dysfunction such as dilated cardiomyopathy and skeletal myopathy. Given the role for tafazzin in regulating cardiolipin – a phospholipid found on the inner membrane of mitochondria – and the known mitochondrial dysfunction (and VSMC alterations) seen in atherosclerosis, they sought to determine whether tafazzin was dysregulated in atherosclerosis. This work builds upon a 2017 publication in *ATVB* by Dr Yu. Atherosclerotic plaques (human and murine) had decreased tafazzin (mRNA and protein) expression overall and specifically in plaque VSMCs, which they determined was due to increased miRNA 125a-5p binding to tafazzin mRNA (presumably targeting it for degradation with consequent decreased protein expression). The induction of miRNA 125a-5p was increased by NFκB (an upstream regulator of miRNA 125a-5p) driven by oxidized LDL. Mechanistically, the authors were able to recapitulate their human plaque observations whereby tafazzin silencing or mutant overexpression decreased VSMC cardiolipin, mitochondrial

respiration, inducing apoptosis and promoting atherosclerosis in murine models. The small peptide SS-31 partially rescued VSMC mitochondrial function and prevented apoptosis. They propose that tafazzin is 1) a newly identified regulator of VSMC mitochondrial function, 2) regulator of atherosclerosis and 3), suggest it may be a novel therapeutic target for atherosclerotic disease.

Significance: This study advances an important body of knowledge on the role of VSMCs in human atherosclerotic plaque. It not only identifies a novel regulator of VSMC dysfunction with meticulous mechanistic insight and acknowledgement of the limitations, but also the pathological implications with models that demonstrate loss of tafazzin in VSMCs leads to more vulnerable plaques as evidenced by increased necrotic core and (thinner) fibrous cap*. *see comment below.

We thank the reviewer for their comments and are pleased that the study “advances an important body of knowledge” and “identifies a novel regulator of VSMC dysfunction with meticulous mechanistic insight”.

COMMENTS:

1. Comparison was normal aortic tissue (AVR or root Sx) – different region than typical athero formation in humans. There are substantial developmental/structural differences between the ascending aorta and the distal sections including the carotid (e.g. elastin is greater the more proximal the segment). Given that VSMCs are exquisitely sensitive to the tensile forces and matrix upon which they reside, this makes finding a control tissue difficult....Options include deceased donor tissue from comparable regions or adjacent plaque regions. The authors notably used media and intimal staining to localize their finding within a biological context.

a. Medial VSMCs had 94% tafazzin expression (along with aortic controls) vs intimal VSMCs only 23% for tafazzin expression – very good. It is an important aspect of this paper and supports the biological process of athero, which is taking place in the intimal layer. This should be emphasized in the discussion regarding control tissues.

We thank the reviewer for the comment and entirely agree that the carotid and aorta are distinct vascular beds with substantial developmental differences. However, the finding that tafazzin expression is retained in medial VSMCs yet decreased in plaque intimal VSMCs (now shown as immunofluorescence rather than immunohistochemistry) suggests that the reduction in tafazzin is specifically associated with atherosclerosis (as discussed in para 2, page 15). We have emphasized this in the revised discussion.

b. Establishing tafazzin expression in the plaque (and specifically the region), and corroborating this data with plaque-derived VSMCs is a strength of this study. Further identifying tafazzin transcripts from human carotid and coronary scRNAseq datasets is a further asset and validating feature, showing this is repeatable and occurs in plaques from different vascular beds. Alsaigh et al have a scRNAseq dataset that includes the adjacent plaque regions – it may provide insight over what a nearby ‘non-diseased’ VSMC tafazzin

transcripts would be. It is publicly available and would strengthen their argument about the focal implications of tafazzin loss.

We thank the reviewer for their suggestion and have now examined the Alsaigh et al data (as shown below.) Similar to the datasets shown in Supplemental Figures 2 and 3, tafazzin expression is low across a range of VSMC subtypes. Due to the technical difficulties in robust detection of lowly expressed genes, it is difficult to compare levels between the VSMC subtypes.

Figure for reviewer

Analysis of single cell RNA-seq profiles of carotid plaque lesions (GSE159677). Left panel: UMAP (Uniform manifold approximation and projection) of plaque cells showing cell type annotation. Right panel: Dot plot showing scaled expression levels for tafazzin (TAZ) and markers of structural cell subsets (MYH11, ACTA2, CD200, TNFRSF11B), endothelial cells (CDH5), leukocytes (PTPRC) and macrophages (CD68). Expression level is shown using a grey-blue color scale. Dot size show the percentage of expressing cells for each cluster

2. oxLDL, but not native unoxidized LDL, increased miRNA 125a-5p expression and decreased tafazzin mRNA and protein in hVSMCs, which could be blocked by a miRNA 125a-5p inhibitor.

a. Is there a dose-dependent effect of oxLDL in initiating this process?

As shown below, we have treated VSMCs with different concentrations of oxLDL and found a dose-dependent effect in initiating this process. We therefore chose to use 100 µg/ml, a concentration associated with significant increase in miR-125a-5p and decrease in tafazzin with minimal cell death.

Figure for reviewer

Quantitative PCR for microRNA 125a-5p (miR-125a-5p) (left) and tafazzin mRNA (right) in hVSMCs treated with oxidized LDL (oxLDL) at the concentrations indicated. Expression is shown relative to untreated control, normalized to U6 small nuclear RNA (left) or $\beta 2$ microglobulin housekeeping gene (right). Data are shown as mean \pm SEM from 4 independent experiments.

- b. Does early athero have reduced mitochondrial function? Or is this an early feature of plaque development? 12wk HFD-fed mice and human plaques provide established lesions, but the advantage of the murine models is to better understand what events are happening along the timelines of plaque development – this knowledge will shape therapeutic strategies and timing for any considered clinical trial (where we do need translational development).**
- c. Do the authors have any data from early athero lesions in their mice?**

We thank the reviewer for their comment. We have previously studied ApoE^{-/-} mice fed high fat diet for 7 and 14 weeks and compared these with chow-fed mice sacrificed at 6 weeks old. After 7 weeks HFD we found early atherosclerotic plaques, aortic and liver mitochondrial DNA damage and decreased liver complex I activity¹⁴. We have now examined mice fed HFD for 7 weeks and compared with chow-fed mice sacrificed at 6 weeks old and found no difference in aortic mitochondrial respiration as shown below. This suggests that aortic mitochondrial respiration is still preserved in early atherosclerosis but becomes reduced in advanced atherosclerosis¹⁵.

Figure for reviewer

Oxygen consumption rate (OCR) in aortas from control (ctrl) 6 week old, chow-fed Apolipoprotein E deficient (ApoE^{-/-}) mice vs aortas from ApoE^{-/-} mice fed high fat diet (HFD) for 7 weeks (n=4, unpaired t test). Data are shown as mean ± SEM.

3. OxLDL and NFκB are well established. NFκB has robust effects in atherosclerosis and naturally addresses the inflammatory component of atherosclerosis – why not consider interfering with this well-known pathway in VSMCs? This should be further contextualized in the discussion, particularly since there could be synergistic benefits on endothelial and macrophage dysfunction.

We fully agree that oxLDL and NFκB are well established in atherosclerosis and interfering with NFκB in VSMCs could be a potential therapeutic approach, particularly as it addresses the inflammatory component of atherosclerosis. However, NFκB is also important for VSMC survival and proliferation^{16, 17} and may affect these pathways independently of changes in tafazzin abundance. Indeed we have found that VSMCs treated with Bay 11-7082, which inhibits IκBα phosphorylation, showed significantly decreased proliferation and a trend towards increased apoptosis as shown below.

Figure for reviewer: Percentage of VSMCs positive for EdU (left) or Annexin V (right) in control VSMCs (ctrl) or VSMCs treated with Bay 11-7082 (Bay) or DMSO vehicle control (DMSO). (n=3, 1-way ANOVA, Bonferroni-Holm post hoc)

A targeted approach to selectively regulate NFkB binding to miR-125a-5p and pro-inflammatory genes may therefore be required to have beneficial effects in atherosclerosis. This interesting area is now included in the revised discussion (para 2, page 20).

4. silencing experiments to examine the functional consequences of loss of tafazzin (done with siRNA and lentivirus gene transfer to express WT tafazzin OR a mutant with decreased efficiency) is strong.

a. How much does the latter decrease tafazzin?

There are broadly similar *in vitro* findings between tafazzin-silenced VSMCs and VSMCs expressing mutant tafazzin after lentiviral transduction. Decreased mitochondrial respiration, ATP/ADP and cardiolipin abundance, and increased mitochondrial ROS are observed in both tafazzin-silenced VSMCs and VSMCs expressing mutant tafazzin. With tafazzin silencing we observed a 50% decrease in tafazzin abundance and VSMCs expressing mutant tafazzin showed 50% decrease in total cardiolipin consistent with a 50% decrease in activity.

b. Is it meaningful functionally – the authors suggest it is based upon the improved mitochondrial readouts done in isolated settings. What this means in a plaque milieu might be difficult to ascertain specifically. Can they comment?

VSMCs have a crucial role in determining plaque stability. VSMCs can proliferate and generate extra-cellular matrix to form the stabilizing plaque cap. However,

VSMC apoptosis leads to cell loss and promotes plaque cap thinning⁴. VSMCs cultured from Sm22aTaz mice show decreased apoptosis and increased proliferation (**Figures 7E-F**); similarly, plaques from Sm22a-Taz mice showed a 50% decrease in apoptotic VSMCs (**Figure 7B**) and almost double the percentage of proliferating cells (**Figure 7C**). Given the importance of VSMC apoptosis/proliferation on plaque stability we suggest that these changes are important functionally, and indeed we observed features of increased plaque stability in Sm22a/Taz mice (**Figures 6C-E**).

c. Could increasing tafazzin be a therapeutic strategy? If so, how much is needed to preserve mitochondrial function? Any dose-response understanding of this process?

We thank the reviewer for their comment and believe that increasing/restoring tafazzin could be a therapeutic strategy. In Sm22a-Taz mice tafazzin expression was increased two-fold compared to control, which was sufficient to increase mitochondrial respiration and promote beneficial effects in plaque composition. As yet, we do not have a dose-response understanding of this process but recognise that this would be important when considering translation.

5. Exciting that the Sm22a-Taz/Apoe^{-/-} mice had no change in plaque area after 14wks HFD, but increased fibrous cap and reduced necrotic core – stabilizing plaques (not only preventing or regressing them) is a valuable clinical target. Seeing that mutant tafazzin increased plaque vulnerability with the inverse observations is quite relevant. However, the data are hard to appreciate - particularly the fibrous cap thickness.

a. IN FIGURE 6 – can the authors show regions of fibrous caps that are being evaluated?

The cap was defined as the area rich in VSMCs (red) overlying the cholesterol rich, acellular core as now stated in the main text (para 3, page 12) and methods (para 1, page 25), and shown in **Supplemental Figure 16**. We have also indicated the regions of fibrous cap being evaluated in **Figure 6**.

b. Can a supplemental be provided that shows more histological examples please?

We have included more histological examples in **Supplemental Figure 16**.

6. Does SS-31 stabilize the reacylated form of cardiolipin? Assume this is the goal as it bypasses the tafazzin role required. Please specify in the manuscript.

We thank the reviewer for the comment. From the literature SS-31 would be expected to bind to cardiolipin, accumulate in and stabilise the structure of the mitochondrial inner membrane, thus promoting respiratory efficiency¹⁸. This may occur without a change in the acylated form of cardiolipin. Indeed, there was no significant difference in MLCL/CL ratio in cardiac mitochondria from Taz knockdown mice treated with SS31 vs control¹⁹. However, data has identified that SS31 binds with mitochondrial inner membrane proteins including respiratory complex

subunits²⁰. We have included these points in our revised manuscript (para 3, page 20).

7. Do the authors expect that tafazzin would be low in macrophages and endothelial cells? These are also highly metabolically active cells that require tight control of their mitochondrial function. As a potential therapeutic target, it will be important to understand how any antagomir or small molecule therapeutic will affect these crucial cells that contribute to plaque progression.

Previous studies have shown that oxidized LDL increases miR-125a-5p expression in macrophages and endothelial cells^{21, 22} and we would predict that tafazzin would be lower in macrophages and endothelial cells in atherosclerotic plaques compared with healthy vessels. Therapies targeting tafazzin may therefore be beneficial in these cells as well.

a. Do the authors have any data of their own on the small molecule inhibitor effects on ECs or macrophages?

We do not have any data of our own on the effects of SS31 in ECs or macrophages. Previous studies have shown that SS31 protects retinal endothelial cells against high glucose-induced injury, including decreasing ROS production and mitochondrial cytochrome C release²³. SS31 also decreased ROS production in macrophages²⁴.

8. Looking for upstream regulators – particularly a microRNA – is another strength of this manuscript. miRNAs are microregulators of many disease processes. For a function that requires such tight control, this is an excellent upstream regulator and amenable to therapeutic designs.

We thank the reviewer for their comment and that looking for upstream regulators is another strength of the manuscript.

a. Did the authors try an antagomir strategy in their transgenic mouse model?

We thank the reviewer for the comment; as yet we have not tried an antagomir in our transgenic mouse model and this is a potential area for future study. However, we note that miRNA-125a-5p may target multiple genes therefore interpretation of the findings may prove difficult.

b. Similar to # 9: How would targets get to inner leaflet?

Mitochondria have unique features that can be used for drug targeting. The mitochondrial membrane potential can be targeted by lipophilic cations, such as triphenylphosphonium (TPP), which can then be covalently linked to the drugs of interest²⁵. The mitochondrial inner membrane also has a unique lipid composition with cardiolipin found almost exclusively in the inner membrane. Small peptides, such as SS31, are targeted to the mitochondria through their direct interaction with cardiolipin.

Any drugs already likely to work on this pathway?

Although developing mitochondrial targeted therapies is an active field, as yet there are no drugs that directly target tafazzin.

The authors mention that the mechanism of action is not known for the small molecule SS-31. This will be an issue advancing forward in human studies.

We thank the reviewer for the comment. Whilst the precise mechanisms of SS31 have yet to be fully elucidated it binds to cardiolipin and thereby accumulates in the mitochondrial inner membrane. It is then predicted to stabilise the structure of the mitochondrial inner membrane, thus promoting respiratory efficiency¹⁸. SS31 also binds with mitochondrial inner membrane proteins including respiratory complex subunits²⁰. Importantly SS31 was found to be safe and well-tolerated in humans²⁶.

Are there antagomir strategies?

Antagomirs have been shown to reduce plaque burden in atherosclerosis²⁷. However, intravenous or subcutaneous administration of an antagomir does not allow targeted delivery to vascular cells. Strategies have therefore been developed to achieve cell-specific delivery, including the use of targeted viral vectors, such as adenoviruses or lentiviruses, or targeted nanoparticles²⁸.

c. miRNA often work collectively – were there other miRNA candidates that should be considered? There are groups actively working on multiple miRNA (multiplex) strategies for clinical approaches.

The miRDB database predicts that tafazzin is targeted by 35 miRNAs. Of these, 3 sites have a target prediction score >90: miR-4319, miR-125b-5p and miR-125a-5p. We found no change in miR-4319 expression between healthy aortas and plaques therefore have not pursued this miRNA further. miR-125b-5p has been previously studied in atherosclerosis and its effects are unclear with both pro-atherogenic and protective roles observed²⁹. Increased miR-125a-5p expression is associated with symptomatic carotid plaques and myocardial infarction^{30, 31} therefore we decided to examine miR-125a-5p further.

A multiplex approach is certainly interesting and we note that there are several miRNA sites that have a probability of conserved targeting >80% which could be explored in this context.

9. Cardiolipin is clearly a highly regulated process – in nature this usually signifies importance of tight regulation.

a. Are there other transacylases to consider?

In addition to tafazzin, monolysocardiolipin acyltransferase and acyl-coA:lysocardiolipin acyltransferase-1 can both remodel monolysocardiolipin to cardiolipin. However, as mutations in tafazzin result in Barth's syndrome this would suggest that these other transacylases are not sufficient to compensate for tafazzin function.

b. Are there GWAS data with SNPs in cardiolipin or tafazzin that might help

demonstrate a role in vasculopathy – beyond the Barth’s syndrome with premature death as they noted?

We thank the reviewer for the comment. A variant in *PTPMT1*, which is involved in cardiolipin synthesis, is associated with hypertension^{12 13} suggesting that defects in cardiolipin may have a role in vasculopathy. We have included this point in the revised discussion (para 2, page 19).

10. Mitochondria have been released in EVs and picked up in blood.(Nat Comm 2023). It is known that EVs can be found in plaques (Blaser et al Circulation 2023). Might it be possible that they would serve as long-range dysfunctional signals to VSMCs? OR long-range therapeutics? While not part of the scope of this project, it should be acknowledged.

We thank the reviewer for highlighting this intriguing area. We agree that mitochondria released in EVs could potentially signal danger or have therapeutic potential³² and have included this in the discussion (para 3, page 19; para 1, page 20).

11. Is there a way to harness delivery to the endothelium for release to local plaque environments (examples would include VCAM-1 targeted nanopeptides for regional delivery, PNAS 2021)?

Targeting atherosclerosis treatment is an important and active area of research. Strategies to target the endothelium include VCAM-1 targeting by a short peptide, mAb to ICAM1, and targeting E-selectin with a thioaptamer molecule³³⁻³⁵. Endothelial cell EV release could then be potentially harnessed to release the therapeutic to the local plaque environment³⁶.

12. hVSMCs isolated from tissue were divided into small pieces and cultured in 6 well plates for 1-2wks to allow cells to emerge.

a. Is there any data on how the cells change over time once ex vivo? Given the meticulous controls, it is unlikely an issue.

We thank the reviewer for their comment. hVSMCs cultured ex vivo do undergo replicative senescence but not typically until passages 10-12. In this study we used hVSMCs at passages 2-8 that did not show any alteration in morphology/proliferation.

b. With the VSMC tafazzin and miRNA-stimulation experiments, was there any significant cell death? The MOI was high and presumably these cells are somewhat less hardy after several weeks in a 6 well plate.

Transfection of miR-125a-5p mimic or siRNA targeting tafazzin did not result in massive cell death, with 7-10% annexin V positive cells for these conditions (**Supplemental Figure 8A, Figure 4I**). With lentiviral transduction of EV, Taz, TazH, there was the initial expected cell death after puromycin selection. After selection there was no significant cell death with 10-15% annexin V positive cells (**Figure 4I**). The data were not shown in the manuscript but the % necrotic cells for all the conditions above was minimal at 0.1-0.7%.

MINOR

1. How were the THP-1 macrophages generated? Assume PMA for 24-48%, but the dose and duration should be in the methods section.

The THP-1 macrophages were generated by treatment with PMA at 50 ng/ml for 72 hours, and these details are now included in the methods (para 2, page 28).

2. There are several places in the methods where the authors have indicated “according to the manufacturers instructions”. A brief statement should be included if possible.

For assays where we have followed the manufacturer’s instructions we have also included a brief statement in the methods.

References

1. Miano JM, Cserjesi P, Ligon KL, Periasamy M, Olson EN. Smooth muscle myosin heavy chain exclusively marks the smooth muscle lineage during mouse embryogenesis. *Circ Res* 1994;**75**:803-812.
2. Chappell J, Harman JL, Narasimhan VM, Yu H, Foote K, Simons BD, Bennett MR, Jorgensen HF. Extensive Proliferation of a Subset of Differentiated, yet Plastic, Medial Vascular Smooth Muscle Cells Contributes to Neointimal Formation in Mouse Injury and Atherosclerosis Models. *Circ Res* 2016;**119**:1313-1323.
3. Hsieh CC, Yen MH, Yen CH, Lau YT. Oxidized low density lipoprotein induces apoptosis via generation of reactive oxygen species in vascular smooth muscle cells. *Cardiovasc Res* 2001;**49**:135-145.
4. Clarke MC, Figg N, Maguire JJ, Davenport AP, Goddard M, Littlewood TD, Bennett MR. Apoptosis of vascular smooth muscle cells induces features of plaque vulnerability in atherosclerosis. *Nat Med* 2006;**12**:1075-1080.
5. Nguyen DK, Distèche CM. Dosage compensation of the active X chromosome in mammals. *Nat Genet* 2006;**38**:47-53.
6. Kameoka S, Adachi Y, Okamoto K, Iijima M, Sesaki H. Phosphatidic Acid and Cardiolipin Coordinate Mitochondrial Dynamics. *Trends in cell biology* 2018;**28**:67-76.
7. Acehan D, Vaz F, Houtkooper RH, James J, Moore V, Tokunaga C, Kulik W, Wansapura J, Toth MJ, Strauss A, Khuchua Z. Cardiac and skeletal muscle defects in a mouse model of human Barth syndrome. *J Biol Chem* 2011;**286**:899-908.
8. Soustek MS, Falk DJ, Mah CS, Toth MJ, Schlame M, Lewin AS, Byrne BJ. Characterization of a transgenic short hairpin RNA-induced murine model of Tafazzin deficiency. *Human gene therapy* 2011;**22**:865-871.
9. Wang S, Li Y, Xu Y, Ma Q, Lin Z, Schlame M, Bezzerides VJ, Strathdee D, Pu WT. AAV Gene Therapy Prevents and Reverses Heart Failure in a Murine Knockout Model of Barth Syndrome. *Circ Res* 2020;**126**:1024-1039.

10. Russo S, De Rasmio D, Rossi R, Signorile A, Lobasso S. SS-31 treatment ameliorates cardiac mitochondrial morphology and defective mitophagy in a murine model of Barth syndrome. *Scientific reports* 2024;**14**:13655.
11. Taylor C, Rao ES, Pierre G, Chronopoulou E, Hornby B, Heyman A, Vernon HJ. Clinical presentation and natural history of Barth Syndrome: An overview. *Journal of inherited metabolic disease* 2022;**45**:7-16.
12. Zhang J, Guan Z, Murphy AN, Wiley SE, Perkins GA, Worby CA, Engel JL, Heacock P, Nguyen OK, Wang JH, Raetz CR, Dowhan W, Dixon JE. Mitochondrial phosphatase PTPMT1 is essential for cardiolipin biosynthesis. *Cell Metab* 2011;**13**:690-700.
13. Liu C, Kraja AT, Smith JA, Brody JA, Franceschini N, Bis JC, Rice K, Morrison AC, Lu Y, Weiss S, Guo X, Palmas W, Martin LW, Chen YD, Surendran P, Drenos F, Cook JP, Auer PL, Chu AY, Giri A, Zhao W, Jakobsdottir J, Lin LA, Stafford JM, Amin N, Mei H, Yao J, Voorman A, Consortium CHDE, Exome BPC, Go TDC, Consortium TDG, Larson MG, Grove ML, Smith AV, Hwang SJ, Chen H, Huan T, Kosova G, Stitzel NO, Kathiresan S, Samani N, Schunkert H, Deloukas P, Myocardial Infarction G, Consortia CAE, Li M, Fuchsberger C, Pattaro C, Gorski M, Consortium CK, Kooperberg C, Papanicolaou GJ, Rossouw JE, Faul JD, Kardina SL, Bouchard C, Raffel LJ, Uitterlinden AG, Franco OH, Vasani RS, O'Donnell CJ, Taylor KD, Liu K, Bottinger EP, Gottesman O, Daw EW, Giulianini F, Ganesh S, Salfati E, Harris TB, Launer LJ, Dorr M, Felix SB, Rettig R, Volzke H, Kim E, Lee WJ, Lee IT, Sheu WH, Tsoie KS, Edwards DR, Liu Y, Correa A, Weir DR, Volker U, Ridker PM, Boerwinkle E, Gudnason V, Reiner AP, van Duijn CM, Borecki IB, Edwards TL, Chakravarti A, Rotter JJ, Psaty BM, Loos RJ, Fornage M, Ehret GB, Newton-Cheh C, Levy D, Chasman DI. Meta-analysis identifies common and rare variants influencing blood pressure and overlapping with metabolic trait loci. *Nat Genet* 2016;**48**:1162-1170.
14. Yu E, Calvert PA, Mercer JR, Harrison J, Baker L, Figg NL, Kumar S, Wang JC, Hurst LA, Obaid DR, Logan A, West NE, Clarke MC, Vidal-Puig A, Murphy MP, Bennett MR. Mitochondrial DNA damage can promote atherosclerosis independently of reactive oxygen species through effects on smooth muscle cells and monocytes and correlates with higher-risk plaques in humans. *Circulation* 2013;**128**:702-712.
15. Yu EPK, Reinhold J, Yu H, Starks L, Uryga AK, Foote K, Finigan A, Figg N, Pung YF, Logan A, Murphy MP, Bennett M. Mitochondrial Respiration Is Reduced in Atherosclerosis, Promoting Necrotic Core Formation and Reducing Relative Fibrous Cap Thickness. *Arterioscler Thromb Vasc Biol* 2017.
16. Giordano A, Avellino R, Ferraro P, Romano S, Corcione N, Romano MF. Rapamycin antagonizes NF-kappaB nuclear translocation activated by TNF-alpha in primary vascular smooth muscle cells and enhances apoptosis. *Am J Physiol Heart Circ Physiol* 2006;**290**:H2459-2465.
17. Hoshi S, Goto M, Koyama N, Nomoto K, Tanaka H. Regulation of vascular smooth muscle cell proliferation by nuclear factor-kappaB and its inhibitor, I-kappaB. *J Biol Chem* 2000;**275**:883-889.
18. Szeto HH. First-in-class cardiolipin-protective compound as a therapeutic agent to restore mitochondrial bioenergetics. *British journal of pharmacology* 2014;**171**:2029-2050.

19. Russo S, De Rasmio D, Signorile A, Corcelli A, Lobasso S. Beneficial effects of SS-31 peptide on cardiac mitochondrial dysfunction in tafazzin knockdown mice. *Scientific reports* 2022;**12**:19847.
20. Chavez JD, Tang X, Campbell MD, Reyes G, Kramer PA, Stuppard R, Keller A, Zhang H, Rabinovitch PS, Marcinek DJ, Bruce JE. Mitochondrial protein interaction landscape of SS-31. *Proc Natl Acad Sci U S A* 2020;**117**:15363-15373.
21. Chen T, Huang Z, Wang L, Wang Y, Wu F, Meng S, Wang C. MicroRNA-125a-5p partly regulates the inflammatory response, lipid uptake, and ORP9 expression in oxLDL-stimulated monocyte/macrophages. *Cardiovasc Res* 2009;**83**:131-139.
22. Zhaolin Z, Jiaojiao C, Peng W, Yami L, Tingting Z, Jun T, Shiyuan W, Jinyan X, Dangheng W, Zhisheng J, Zuo W. OxLDL induces vascular endothelial cell pyroptosis through miR-125a-5p/TET2 pathway. *J Cell Physiol* 2019;**234**:7475-7491.
23. Li J, Chen X, Xiao W, Ma W, Li T, Huang J, Liu X, Liang X, Tang S, Luo Y. Mitochondria-targeted antioxidant peptide SS31 attenuates high glucose-induced injury on human retinal endothelial cells. *Biochem Biophys Res Commun* 2011;**404**:349-356.
24. Shang L, Ren H, Wang S, Liu H, Hu A, Gou P, Lin Y, Zhou J, Zhu W, Shi X. SS-31 Protects Liver from Ischemia-Reperfusion Injury via Modulating Macrophage Polarization. *Oxid Med Cell Longev* 2021;**2021**:6662156.
25. Murphy MP, Smith RA. Targeting antioxidants to mitochondria by conjugation to lipophilic cations. *Annu Rev Pharmacol Toxicol* 2007;**47**:629-656.
26. Reid Thompson W, Hornby B, Manuel R, Bradley E, Laux J, Carr J, Vernon HJ. A phase 2/3 randomized clinical trial followed by an open-label extension to evaluate the effectiveness of elamipretide in Barth syndrome, a genetic disorder of mitochondrial cardiolipin metabolism. *Genet Med* 2021;**23**:471-478.
27. Loyer X, Potteaux S, Vion AC, Guerin CL, Boulkroun S, Rautou PE, Ramkhalawon B, Esposito B, Dalloz M, Paul JL, Julia P, Maccario J, Boulanger CM, Mallat Z, Tedgui A. Inhibition of microRNA-92a prevents endothelial dysfunction and atherosclerosis in mice. *Circ Res* 2014;**114**:434-443.
28. Saenz-Pipaon G, Dichek DA. Targeting and delivery of microRNA-targeting antisense oligonucleotides in cardiovascular diseases. *Atherosclerosis* 2023;**374**:44-54.
29. Chao CT, Yeh HY, Yuan TH, Chiang CK, Chen HW. MicroRNA-125b in vascular diseases: An updated systematic review of pathogenetic implications and clinical applications. *J Cell Mol Med* 2019;**23**:5884-5894.
30. Maitrias P, Metzinger-Le Meuth V, Massy ZA, M'Baya-Moutoula E, Reix T, Caus T, Metzinger L. MicroRNA deregulation in symptomatic carotid plaque. *Journal of vascular surgery* 2015;**62**:1245-1250 e1241.
31. Yuan N, Scherzer R, Tanriverdi K, Martin J, Rahalkar S, Hsue P. MicroRNA biomarkers associated with type 1 myocardial infarction in HIV-positive individuals. *Aids* 2019;**33**:2351-2361.
32. Puhm F, Afonyushkin T, Resch U, Obermayer G, Rohde M, Penz T, Schuster M, Wagner G, Rendeiro AF, Melki I, Kaun C, Wojta J, Bock C, Jilma B, Mackman N, Boilard E, Binder CJ. Mitochondria Are a Subset of Extracellular Vesicles Released by Activated Monocytes and Induce Type I IFN and TNF Responses in Endothelial Cells. *Circ Res* 2019;**125**:43-52.

33. Zhou Z, Yeh CF, Mellas M, Oh MJ, Zhu J, Li J, Huang RT, Harrison DL, Shentu TP, Wu D, Lueckheide M, Carver L, Chung EJ, Leon L, Yang KC, Tirrell MV, Fang Y. Targeted polyelectrolyte complex micelles treat vascular complications in vivo. *Proc Natl Acad Sci U S A* 2021;**118**.
34. Serrano D, Bhowmick T, Chadha R, Garnacho C, Muro S. Intercellular adhesion molecule 1 engagement modulates sphingomyelinase and ceramide, supporting uptake of drug carriers by the vascular endothelium. *Arterioscler Thromb Vasc Biol* 2012;**32**:1178-1185.
35. Ma S, Tian XY, Zhang Y, Mu C, Shen H, Bismuth J, Pownall HJ, Huang Y, Wong WT. E-selectin-targeting delivery of microRNAs by microparticles ameliorates endothelial inflammation and atherosclerosis. *Scientific reports* 2016;**6**:22910.
36. Raju S, Botts SR, Blaser MC, Abdul-Samad M, Prajapati K, Khosraviani N, Ho TWW, Breda LCD, Ching C, Galant NJ, Fiddes L, Wu R, Clift CL, Pham T, Lee WL, Singh SA, Aikawa E, Fish JE, Howe KL. Directional Endothelial Communication by Polarized Extracellular Vesicle Release. *Circ Res* 2024;**134**:269-289.

Response to reviewers

We thank the referees for their comments and our point-by-point responses are as below.

REVIEWER COMMENTS

Reviewer #1 (Remarks to the Author):

Comments to the Authors:

The authors performed many of the requested experiments, including in vitro rescue experiments (Fig. 2F-H and Fig. 3B-D), additional in vitro experiments characterizing mitochondrial morphology and gene/protein expression profiles, and mitochondrial ROS measurement, showed additional comparison between experimental groups, included discussion about the endogenous X-chromosome location of TAZ gene vs. transgenic expression of human TAZ gene in the SM22a-Taz/TazH mice, and added more details about methods. However, several critical concerns were left unanswered.

1 – Although the authors replaced IH staining for SMA/TAZ with IF and confocal microscopy for human aortic and carotid plaques, these data only show that medial SMA+ cells (presumably SMC) express TAZ in both aorta and carotid artery with plaque and SMA+ cells in plaque do not express TAZ. Given that SMA is not a specific SMC marker in the lesion, the author can only claim that they see a significant decrease in SMA+TAZ+ cells within the lesion compared to the medial cells. The same is true about miR-125a-5p staining (here they can only discuss media vs. plaque cells). Therefore, all conclusions about TAZ signaling in plaque SMCs are overstated.

As suggested, these data could be supplemented by IF staining on mouse atherosclerotic tissues from SMC-lineage tracing mice. The authors wrote in their rebuttal letter that they had technical issues combining SMC-lineage tracing staining with the miR125a-5p hybridization protocol. Still, it is unclear why they did not perform SMC lineage tracing with TAZ staining to show the SMC-specific TAZ expression within the plaque.

Although the technical difficulties are understandable, the revised version of the manuscript still has the same issues about the origin of the SMA+TAZ+ cells in the plaque as the original one.

We thank the reviewer for their comment. We acknowledge that whilst SMA is a VSMC marker, other cell types within human plaques may express SMA. We have therefore re-written the results section (para 1, page 4) to describe the cells as “SMA-positive cells” rather than VSMCs.

In response to the reviewer’s query, in the initial revision we attempted to combine RNA in situ hybridization for microRNA 125a-5p and tafazzin with imaging for the fluorescence signal from the VSMC-lineage tracing confetti reporter. Unfortunately, the fluorescence signal from the confetti reporter was lost after processing for

miRNAscope. We were therefore unable to demonstrate tafazzin mRNA localization specifically in VSMCs.

However we thank the reviewer for their comment and have now examined tafazzin expression by immunofluorescence in SMC-lineage traced atherosclerotic mice (Myh11-CreERT2/Rosa26-Confetti/ApoE^{-/-}), which stably express red, yellow, green or cyan fluorescent protein in mature VSMCs and their progeny. We show that tafazzin expression is significantly decreased in plaque VSMCs compared with medial VSMCs of mouse atherosclerotic arteries (data described in para 2, page 5, shown in **Supplemental Figure 3**).

2 – The rationale of having publicly available scRNAseq datasets in the manuscript and which important conclusions have been made from these analyses (except that "quantification of lowly expressed genes is problematic") is unclear and confusing.

The authors' relabeling SMCs and fibroblasts to "structural" cells and the rationale for doing this (from the rebuttal letter) is also confusing. Many SMC labs, including Thomas Quertermous' (the scRNAseq dataset that the authors use in this manuscript), Gary Owens', and Muredach Reilly's labs (and many others), performed scRNAseq on SMC lineage tracing atherosclerotic mice and human atherosclerotic arteries and had very well characterized marker panels for phenotypically modulated SMC and fibroblasts.

The difference between TAZ expression levels in SMC vs macrophages and endothelial cells is not supported by any statistical analysis.

As presented, all these scRNAseq data do not support the authors' hypothesis about the role of TAZ in SMC in atherosclerosis and should be removed from the manuscript.

We thank the reviewer for their comment, and have therefore removed the previous scRNA seq analyses from the manuscript. Instead we examined TAZ expression in human plaque cells based on a published meta-analysis¹ of data from carotid^{2, 3} and coronary plaques⁴ (**Supplemental Figure 2**). To avoid confusion about cell type annotation, we have used the clustering and labelling provided in the meta-analysis. This confirms that tafazzin expression is low in VSMCs from plaque samples and finds that expression is significantly lower in VSMCs compared with macrophages (log₂ (fold change) = -0.598, p = 1.15 × 10⁻⁴⁵) and endothelial cells (log₂ (fold change) = 0.769, p = 1.17 × 10⁻²⁰) (data described in para 1, page 4, shown in **Supplemental Figure 2**).

3 – The authors did not answer how plaque SMCs were isolated and characterized and how they were distinguished from fibroblasts. Instead, the authors reference Supplementary Fig. 1, which shows staining for SMA, MYH11, and CD68 without explaining which hVSMC they show (plaque vs. aortic) and how these cells were cultured (e.g., in serum vs. in serum-free media).

We thank the reviewer for their comment. Human plaque VSMCs were cultured from carotid endarterectomy samples. The tissue was divided into small pieces (2-3mm²)

and cultured in 6 well plates for 1-2 weeks to allow cells to emerge. Plaque VSMCs were cultured in Smooth Muscle Cell Growth Medium 2 (Promocell) with 5% Fetal Bovine Serum, supplemented with 100U/ml penicillin, 100 µg/ml streptomycin. The methods have been revised to more clearly describe these details (para 2, page 28).

Plaque VSMCs were identified by immunofluorescence staining for SMA and MYH11 (para 1, page 4, **Supplemental Figure 1A**). The legend has been revised to include culture medium details and to explain which hVSMCs are shown. SMA is a well-recognized SMC marker but can be expressed by other cell types. However, MYH11 is considered to be the most specific VSMC marker and its expression distinguishes the plaque VSMCs from fibroblasts, which only have minimal MYH11 expression^{5,2}. The plaque VSMCs also express TNFRSF11B, a highly specific marker for phenotypically modulated VSMCs² (Figure below). Taken together, the expression of SMA, MYH11 and TNFRSF11B would be consistent with a VSMC cell type undergoing phenotypical modulation.

Figure for reviewer: Isolated plaque human vascular smooth muscle cells were stained for TNFRSF11B; Hoechst 33342 used as nuclear stain

Should the plaque phenotypically modulated SMC have lower (or minimal if completely de-differentiated) levels of the SMC contractile markers?

Contractile markers are progressively downregulated in VSMCs undergoing phenotypical modulation, resulting in a gradient of decreased expression as cells transition from a VSMC to a modulated phenotype^{2, 4}. However, contractile markers are still expressed by phenotypically modulated VSMCs, and indeed MYH11 was identified as one of the defining genes of the phenotypically modulated VSMC cluster². Overall, plaque VSMCs show heterogeneity in their degree of phenotypical modulation and minimal to significant downregulation of VSMC markers may be observed dependent on cell state.

4 – IF staining in Figure 7. There are concerns about this staining. SMA looks overexposed, MAC3 is invisible, and TUNEL staining in the TAZH group looks unspecific (sticky necrotic core unspecificity?). A better representation that includes negative controls would be helpful.

We thank the reviewer for their comments. We acknowledge that the Mac3 staining was of low intensity, which unfortunately did not improve despite varying the

antibody concentration/incubation duration; attempts with CD68 antibody were similarly technically unsuccessful. Given these limitations, in **Figure 7A** we now only present TUNEL co-stained with SMA but this remains well-aligned with the paper which focuses on the effects of tafazzin in VSMCs. For **Figure 7A** a lower exposure for the SMA staining has also been used for all groups, and a better representation of the TUNEL staining has been used for the TazH group. We have also included negative controls in **Supplemental Figure 17A** where TUNEL staining was performed with the terminal deoxynucleotidyl transferase reaction omitted, and immunohistochemistry was performed with mouse IgG control antibody followed by incubation with Alexa Fluor 488-labelled secondary antibody.

5 – The authors partially revised the Discussion section, but it is still very long and repeating results.

We thank the reviewer for their comment. The discussion has been further revised to minimise the repetition of results. During the revision process we have been asked to discuss multiple interesting areas regarding our findings which has contributed to the length of the discussion.

6 – Given the X-chromosome location of the TAZ gene, the authors should indicate the sex of mice/cells/human specimens they used for analyses.

We have indicated the sex of mice/cells/human tissue samples in the text and all figure legends. Data disaggregated by sex is now included in the source data.

Reviewer #2 (Remarks to the Author):

I appreciate the detailed response by the authors. Still, in my opinion, the work merely describes observations that have unproven biological significance and that provide little mechanistic insight.

We thank the reviewer for their comment but must respectfully disagree that the observations have unproven biological significance and provide little mechanistic insight. In terms of biological significance atherosclerosis remains the leading cause of death worldwide despite advances in treatment. There is therefore a continuing need to identify the underlying disease mechanisms and novel therapeutic strategies. Plaque VSMCs show decreased mitochondrial respiration but what disrupts mitochondrial function, and what the consequences are, have not been previously identified. In this study we show for the first time that VSMC tafazzin is decreased in atherosclerosis and identify a potential mechanism for its downregulation. Specifically, microRNA 125a-5p expression is increased in plaques, downregulates tafazzin expression, and is induced by oxidized low-density lipoprotein in a NFkB-dependent manner. Silencing tafazzin or overexpression of mutant tafazzin decreases VSMC cardiolipin content, complex I protein abundance and mitochondrial respiration, and promotes apoptosis and atherosclerosis. Whilst the changes in tafazzin expression in our human data may only be associated with atherosclerosis, the findings of altered plaque composition in Sm22a-Taz/ApoE^{-/-}

mice and increased plaque burden in Sm22a-TazH/ApoE^{-/-} mice indicate that tafazzin levels do indeed have important effects on atherogenesis. Thus, our studies show both biological significance and multiple mechanistic insights, and furthermore identify VSMC tafazzin as a potential new therapeutic target in atherosclerotic disease.

If indeed, TAZ affects the expression of complex I in any specific manner, one would have to reveal the underlying mechanism of transcriptional regulation.

Regarding the effects of tafazzin on complex I expression we have now performed qPCR to assess the expression of multiple complex I subunits, covering the p, q and n modules, and both mitochondrial and nuclear-encoded genes. mRNA expression of these genes was unchanged in VSMCs from Taz and TazH mice consistent with post-transcriptional regulation of protein abundance (described in para 1, page 12, shown in **Supplemental Figure 14C**). Cardiolipin stabilises respiratory supercomplexes⁶ therefore we next examined respiratory complex assembly using blue native polyacrylamide gel electrophoresis and immunoblot analysis. Whilst levels of complex I/III₂/IV supercomplex and individual complex I were increased in Taz VSMCs, lower levels of these complexes were seen in TazH VSMCs. The changes in supercomplex and holoenzyme abundance would explain the changes observed in complex I subunit abundance (data described in para 1, page 12, shown in **Supplemental Figure 14D**, discussed in para 2, page 18). Our data therefore indicate that tafazzin regulates complex I subunit abundance through effects on holoenzyme and supercomplex stability, not at transcriptional level.

Reviewer #3 (Remarks to the Author):

The Authors have satisfied my questions and comments as requested.

We thank the reviewer for their comments.

1. Bleckwehl, T. *et al.* Encompassing view of spatial and single-cell RNA sequencing renews the role of the microvasculature in human atherosclerosis. *Nat Cardiovasc Res* **4**, 26-44 (2025).
2. Wirka, R.C. *et al.* Atheroprotective roles of smooth muscle cell phenotypic modulation and the TCF21 disease gene as revealed by single-cell analysis. *Nat Med* **25**, 1280-1289 (2019).
3. Alsaigh, T., Evans, D., Frankel, D. & Torkamani, A. Decoding the transcriptome of calcified atherosclerotic plaque at single-cell resolution. *Commun Biol* **5**, 1084 (2022).
4. Pan, H. *et al.* Single-Cell Genomics Reveals a Novel Cell State During Smooth Muscle Cell Phenotypic Switching and Potential Therapeutic Targets for Atherosclerosis in Mouse and Human. *Circulation* **142**, 2060-2075 (2020).
5. Miano, J.M., Cserjesi, P., Ligon, K.L., Periasamy, M. & Olson, E.N. Smooth muscle myosin heavy chain exclusively marks the smooth muscle lineage during mouse embryogenesis. *Circ Res* **75**, 803-812 (1994).
6. Wenz, T. *et al.* Role of phospholipids in respiratory cytochrome bc(1) complex catalysis and supercomplex formation. *Biochim Biophys Acta* **1787**, 609-616 (2009).

Response to reviewers

REVIEWERS' COMMENTS

Reviewer #1 (Remarks to the Author):

The authors addressed all my comments and concerns.

We thank the reviewer for their comments.

Reviewer #2 (Remarks to the Author):

I remain unconvinced of the specific connection between tafazzin and atherosclerosis. It is not clear whether microRNA 125a-5p suppresses tafazzin specifically or mitochondrial proteins in general. If the former, I would agree with the authors. If the latter, the work would show that mitochondria are involved in the pathogenesis of atherosclerosis.

We thank the reviewer for their comments. We are also grateful for the editorial comments regarding this point which we have followed. We now acknowledge that “the potential effects of miR-125a-5p on other mitochondrial proteins cannot be completely excluded” in the limitations section of the discussion (para 2, p21). However, the specific tafazzin-targeted experiments have enabled us to identify functional consequences of reduced tafazzin expression or activity in VSMCs.